# A Flexible, Equivariant Framework for Subgraph GNNs via Graph Products and Graph Coarsening

**Guy Bar-Shalom**[*]
Computer Science
Technion - Israel Institute of Technology
`guy.b@campus.technion.ac.il`

**Yam Eitan**[*]
Electrical & Computer Engineering
Technion - Israel Institute of Technology
`yameitan1997@gmail.com`

**Fabrizio Frasca**
Electrical & Computer Engineering
Technion - Israel Institute of Technology
`fabrizio.frasca.effe@gmail.com`

**Haggai Maron**
Electrical & Computer Engineering
Technion - Israel Institute of Technology
NVIDIA Research
`haggaimaron@gmail.com`

## Abstract

Subgraph GNNs enhance message-passing GNNs expressivity by representing graphs as sets of subgraphs, demonstrating impressive performance across various tasks. However, their scalability is hindered by the need to process large numbers of subgraphs. While previous approaches attempted to generate smaller subsets of subgraphs through random or learnable sampling, these methods often yielded suboptimal selections or were limited to small subset sizes, ultimately compromising their effectiveness. This paper introduces a new Subgraph GNN framework to address these issues. Our approach diverges from most previous methods by associating subgraphs with node clusters rather than with individual nodes. We show that the resulting collection of subgraphs can be viewed as the product of coarsened and original graphs, unveiling a new connectivity structure on which we perform generalized message passing.

Crucially, controlling the coarsening function enables meaningful selection of any number of subgraphs. In addition, we reveal novel permutation symmetries in the resulting node feature tensor, characterize associated linear equivariant layers, and integrate them into our Subgraph GNN. We also introduce novel node marking strategies and provide a theoretical analysis of their expressive power and other key aspects of our approach. Extensive experiments on multiple graph learning benchmarks demonstrate that our method is significantly more flexible than previous approaches, as it can seamlessly handle any number of subgraphs, while consistently outperforming baseline approaches. Our code is available at `https://github.com/BarSGuy/Efficient-Subgraph-GNNs`.

## 1 Introduction

Subgraph GNNs [4, 12, 39, 8, 27, 29, 38, 3] have recently emerged as a promising direction in graph neural network research, addressing the expressiveness limitations of Message Passing Neural Networks (MPNNs) [24, 35, 25]. In essence, a Subgraph GNN operates on a graph by transforming it into a collection of subgraphs, generated based on a specific selection policy. Examples of such policies include removing a single node from the original graph or simply marking a node without changing the graph's original connectivity [26]. The model then processes these subgraphs using an equivariant architecture, aggregates the derived representations, and makes graph- or node-level predictions. The growing popularity of Subgraph GNNs stems not only from their enhanced

---

[*]Equal contribution.

38th Conference on Neural Information Processing Systems (NeurIPS 2024).

expressive capabilities over MPNNs but also from their impressive empirical results, as notably demonstrated on well-known molecular benchmarks [38, 12, 3].

Unfortunately, Subgraph GNNs are hindered by substantial computational costs as they necessitate message-passing operations across all subgraphs within the bag. Typically, the number of subgraphs is the number of nodes in the graph, $n$— for bounded degree graphs, this results in a time complexity scaling quadratically ($\mathcal{O}(n^2)$), in contrast to the linear complexity of a standard MPNN. This significant computational burden makes Subgraph GNNs impractical for large graphs, hindering their applicability to important tasks and widely used datasets. To overcome this challenge, various studies have explored methodologies that process only a subset of subgraphs from the bag. These methods range from simple random sampling techniques [8, 4, 40, 3] to more advanced strategies that learn to select the most relevant subset of the bag to process [5, 20, 29]. However, while random sampling of subgraphs yields subpar performance, more sophisticated learnable selection strategies also have significant limitations. Primarily, they rely on *training-time* discrete sampling which complicates the optimization process, as evidenced by the high number of epochs required to train them [20, 5, 29]. As a result, these methods often allow only a very small bag size, yielding only modest performance improvements compared to random sampling and standard MPNNs.

**Our approach.** The goal of this paper is to devise a Subgraph GNN architecture that can flexibly generate and process variable-sized bags, and deliver strong experimental results while sidestepping intricate and lengthy training protocols. Specifically, our approach aims to overcome the common limitation of restricting usage to a very small set of subgraphs.

Our proposed method builds upon and extends an observation made by Bar-Shalom et al. [3], who draw an analogy between using Subgraph GNNs and performing message-passing operations over a larger "product graph". Specifically, it was shown that when considering the maximally expressive (node-based) Subgraph GNN suggested by [38][2], the bag of subgraphs and its update rules can be obtained by transforming a graph

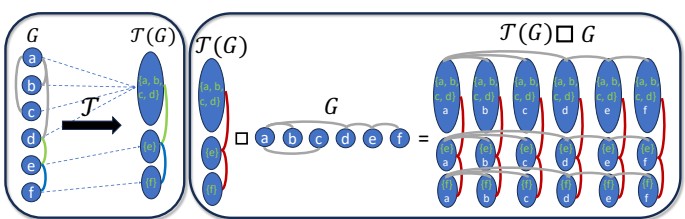

Figure 1: Product graph construction. **Left:** Transforming of the graph into a coarse graph; **Right:** Cartesian product of the coarsened graph with the original graph. The vertical axis corresponds to the subgraph dimension (super-nodes), while the horizontal axis corresponds to the node dimension (nodes).

through the *graph cartesian product* of the original graph with itself, i.e., $G \square G$, and then processing the resulting graph using a standard MPNN. In our approach, we propose to modify the first term of the product and replace it with a *coarsened* version of the original graph, denoted $\mathcal{T}(G)$, obtained by mapping nodes to *super-nodes* (e.g., by applying graph clustering, see Figure 1(left)), making the resulting product graph $\mathcal{T}(G) \square G$ significantly smaller. This construction is illustrated in Figure 1(right). This process effectively associates each subgraph – a row in Figure 1(right) – with a set of nodes produced by the coarsening function $\mathcal{T}$. Different choices of $\mathcal{T}$ allow for both flexible bag sizes and a simple, meaningful selection of the subgraphs.

While performing message passing on $\mathcal{T}(G) \square G$ serves as the core update rule in our architecture, we augment our message passing operations with another set of operations derived from the symmetry structure of the resulting node feature tensor, which we call *symmetry-based updates*. Specifically, our node feature tensor is indexed by pairs $(S, v)$ where $S$ is a super-node and $v$ is an original node. Accordingly, $\mathcal{X}$ is a $T \times n \times d$ tensor, where $d$ is the feature dimension, and $T$ is the number of super-nodes (a constant hyper-parameter). As super-nodes are sets of nodes, $\mathcal{X}$ can also be viewed as a (very) sparse $2^n \times n \times d$ tensor where $2^n$ is the number of all subsets of the vertex set. Since the symmetric group $S_n$ acts naturally on this representation, we use it to develop symmetry based updates.

Interestingly, we find that this node feature tensor, $\mathcal{X}$, adheres to a specific set of symmetries, which, to the best of our knowledge, is yet unstudied in the context of machine learning: applying

---

[2]The architecture suggested in [38] was shown to be at least as expressive as all previously studied node-based Subgraph GNNs

a permutation $\sigma \in S_n$ to the nodes in $S$ and to $v$ results in an equivalent representation of our node feature tensor. We formally define the symmetries of this object and characterize all the affine equivariant operations in this space. We incorporate these operations into our message-passing by encoding the parameter-sharing schemes [30] as additional edge features. These additional update rules significantly improve experimental results. We note that our symmetry analysis may be useful for processing bags derived from other high-order generation policies [29, 20] by treating tuples of nodes as sets.

Inspired by these symmetries and traditional binary-based [4] and shortest path-based [38] *node-marking* strategies, we propose four natural marking strategies for our framework. Interestingly, unlike the full-bag scenario, they vary in expressiveness, with the shortest path-based technique being the most expressive.

The flexibility and effectiveness of our full framework are illustrated in Figure 2, depicting detailed experimental results on the popular ZINC-12K dataset [31]. Our method demonstrates a significant performance boost over baseline models in the *small bag* setting (for which they are designed), while achieving results that compare favourably to state-of-the-art Subgraph GNNs in the *full bag* setting. Additionally, we can obtain results in-between these two regimes.

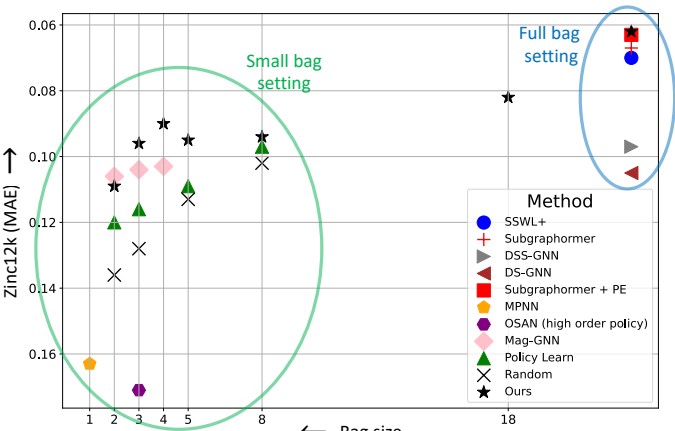

Figure 2: The performance landscape of Subgraph GNNs with varying number of subgraphs: Our method leads in the lower bag-size set, outperforming other approaches in nearly all cases. Additionally, our method matches the performance of state-of-the-art Subgraph GNNs in the full-bag setting. The full mean absolute error (MAE) scores along with standard deviations are available in Table 9 in the appendix.

**Contributions.** The main contributions of this paper are: (1) the development of a novel, flexible Subgraph GNN framework that enables meaningful construction and processing of bags of subgraphs of any size; (2) a characterization of all affine invariant/equivariant layers defined on our node feature tensors; (3) a theoretical analysis of our framework, including the expressivity benefits of our node-marking strategy; and (4) a comprehensive experimental evaluation demonstrating the advantages of the new approach across both small and large bag sizes, achieving state-of-the-art results, often by a significant margin.

## 2 Related work

**Subgraph GNNs.** Subgraph GNNs [39, 8, 27, 4, 40, 26, 12, 29, 17, 38, 3] represent a graph as a collection of subgraphs, obtained by a predefined generation policy. For example, each subgraph can be generated by marking exactly one node in the original graph (see inset [3]) – an approach commonly referred to as *node marking* [26]; this marked node is considered the root node in its subgraph. Several recent papers focused on scaling these methods to larger graphs, starting with basic random selection of subgraphs from the bag, and extending beyond with more sophisticated techniques that aim to learn how to select subgraphs. To elaborate, [5] introduced *Policy-Learn* (PL), an approach based on two models, where the first model predicts a distribution over the nodes of the original graph, and the second model processes bags of subgraphs sampled from this distribution. *MAG-GNN* [20] employs a similar approach utilizing Reinforcement Learning. Similarly to our approach, this method permits high-order policies by associating subgraphs with tuples rather than individual nodes, allowing for the marking of several nodes within a subgraph.

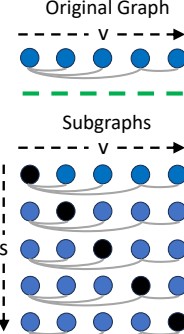

---
[3]The Figure was taken with permission from [3]

However, as mentioned before, these approaches involve discrete sampling while training, making them very hard to train (1000-4000 epochs vs. ∼400 epochs of state-of-the-art methods [3, 38] on the ZINC-12K dataset), and limiting their usage to very small bags. Finally, we mention another high-order method, *OSAN*, introduced by [29], which learns a distribution over tuples that represent subgraphs with multiple node markings. In contrast to these previous approaches, we suggest a simpler and more effective way to select subgraphs and also show how to leverage the resulting symmetry structure to augment our message-passing operations.

**Symmetries in graph learning.** Many previous works have analyzed and utilized the symmetry structure that arises from graph learning setups [22, 23, 18, 2]. Specifically relevant to our paper is the work of [22] that characterized basic equivariant linear layers for graphs, the work of [1] that characterizes equivariant maps for many other types of incidence tensors that arise in graph learning, and the works [4, 12] that leveraged group symmetries for designing Subgraph GNNs in a principled way.

## 3 Preliminaries

**Notation.** Let $\mathcal{G}$ be a family of undirected graphs, and consider a graph $G = (V, E)$ within this family. The adjacency matrix $A \in \mathbb{R}^{n \times n}$ defines the connectivity of the graph[4], while the feature matrix $X \in \mathbb{R}^{n \times d}$ represents the node features. Here, $V$ and $E$ represent the sets of nodes and edges, respectively, with $|V| = n$ indicating the number of nodes. We use the notation $v_1 \sim_A v_2$ to denote that $v_1$ and $v_2$ are neighboring nodes according to the adjacency $A$. Additionally, we define $[n] := \{1, 2, \ldots n\}$, and $\mathcal{P}([n])$ as the power set of $[n]$.

**Subgraph GNNs as graph products.** In a recent work, [3] demonstrated that various types of update rules used by current Subgraph GNNs can be simulated by employing the *Cartesian graph product* between the original graph and another graph, and running standard message passing over that newly constructed product graph. Formally, the cartesian product of two graphs $G_1$ ($n_1$ nodes) and $G_2$ ($n_2$ nodes), denoted by $G_1 \square G_2$, forms a graph with vertex set $V(G_1) \times V(G_2)$. Two vertices $(u_1, u_2)$ and $(v_1, v_2)$ are adjacent if either $u_1 = v_1$ and $u_2$ is adjacent to $v_2$ in $G_2$, or $u_2 = v_2$ and $u_1$ is adjacent to $v_1$ in $G_1$. We denote by $\mathcal{A} \in \mathbb{R}^{n_1 \cdot n_2 \times n_1 \cdot n_2}$ and $\mathcal{X} \in \mathbb{R}^{n_1 \cdot n_2 \times d}$ the adjacency and node feature matrices of the product graph; in general, we use calligraphic letters to denote the adjacency and feature matrices of product graphs, while capital English letters are used for those of the original graphs. In particular, for the graph cartesian product, $G_1 \square G_2$, the following holds:

$$\mathcal{A}_{G_1 \square G_2} = A_1 \otimes I + I \otimes A_2. \tag{1}$$

For a detailed definition of the cartesian product of graphs, please refer to Definition A.1. As a concrete example for the analogy between Subgraph GNNs and the Cartesian product of graphs, we refer to a result by [3], which states that the maximally expressive node-based Subgraph GNN architecture GNN-SSWL+ [38], can be simulated by an MPNN on the Cartesian product of the original graph with itself, denoted as $G \square G$. As we shall see, our framework utilizes a cartesian product of the original graph and a coarsened version of it, as illustrated in Figure 1 (right).

**Equivariance.** A function $L : U \to W$ is called equivariant if it commutes with the group action. More formally, given a group element, $g \in \mathbb{G}$, the function $L$ should satisfy $L(g \cdot v) = g \cdot L(v)$ for all $v \in U$ and $g \in \mathbb{G}$. $L$ is said to be invariant if $L(g \cdot v) = L(v)$.

## 4 Coarsening-based Subgraph GNN

**Overview.** This section introduces the *Coarsening-based Subgraph GNN* (CS-GNN) framework. The main idea is to select and process subgraphs in a principled and flexible manner through the following approach: (1) coarsen the original graph via a coarsening function, $\mathcal{T}$ – see Figure 1(left); (2) Obtain the product graph – Figure 1(right) defined by the combination of two adjacencies, $\mathcal{A}_{\mathcal{T}(G)}$ (red edges), $\mathcal{A}_G$ (grey edges), which arise from the graph Cartesian product operation (details follow); (3) leveraging the symmetry of this product graph to develop *symmetry-based* updates, described by $\mathcal{A}_{\text{Equiv}}$ (this part is not visualized in Figure 1). The general update of our suggested layer takes the following form [4],

---

[4]Edge features are also allowed but are omitted here for simplicity

$$\mathcal{X}^{t+1}(S, v) = f^t \Big( \mathcal{X}(S, v)^t, \tag{2}$$

$$\underbrace{\{\!\!\{\mathcal{X}(S', v')^t\}\!\!\}_{(S', v') \sim \mathcal{A}_G(S, v)}}_{\text{Original connectivity (horizontal)}}, \underbrace{\{\!\!\{\mathcal{X}(S', v')^t\}\!\!\}_{(S', v') \sim \mathcal{A}_{\mathcal{T}(G)}(S, v)}}_{\text{Induced connectivity (vertical)}}, \underbrace{\{\!\!\{\mathcal{X}(S', v')^t\}\!\!\}_{(S', v') \sim \mathcal{A}_{\text{Equiv}}(S, v)}}_{\text{Symmetry-based updates}} \Big),$$

where the superscript $^t$ indicates the layer index. In what follows, we further elaborate on these three steps (in Sections 4.1 to 4.2).

We note that each connectivity in Equation (2) is processed using a distinct MPNN, and after stacking of those layers, we apply a pooling layer[5] to obtain a graph representation; that is, $\rho(\mathcal{X}^{\mathrm{T}}) = \mathtt{MLP}^{\mathrm{T}}\Big( \sum_S \big( \sum_{v=1}^n \mathcal{X}^{\mathrm{T}}(S, v) \big) \Big)$; $\mathrm{T}$ denotes the final layer.

For more specific implementation details, we refer to Appendix F.

## 4.1 Construction of the coarse product graph

As mentioned before, a maximally expressive node-based Subgraph GNN can be realized via the Cartesian product of the original graph with itself $G \square G$. In this work, we extend this concept by allowing the left operand in the product to be the coarsened version of $G$, denoted as $\mathcal{T}(G)$, as defined next. This idea is illustrated in Figure 1.

**Graph coarsening.** Consider a graph $G = (V, E)$ with $n$ nodes and an adjacency matrix $A$. Graph coarsening is defined by the function $\mathcal{T} : \mathcal{G} \to \mathcal{G}$, which maps $G$ to a new graph $\mathcal{T}(G) = (V^{\mathcal{T}}, E^{\mathcal{T}})$ with an adjacency matrix $A^{\mathcal{T}} \in \mathbb{R}^{2^n \times 2^n}$ and a feature matrix $X^{\mathcal{T}} \in \mathbb{R}^{2^n \times d}$. Here, $V^{\mathcal{T}}$, the vertex set of the new graph represents super-nodes – defined as subsets of $[n]$. Additionally, we require that nodes in $V^{\mathcal{T}}$ induce a partition over the nodes of the original graph[6]. The connectivity $E^{\mathcal{T}}$ is extremely sparse and induced from the original graph's connectivity via the following rule:

$$A^{\mathcal{T}}(S_1, S_2) = \begin{cases} 1 & \text{if } \exists v \in S_1, \exists v \in S_2 \text{ s.t. } A(v, u) = 1, \\ 0 & \text{otherwise,} \end{cases} \tag{3}$$

To clarify, in our running example (Figure 1), it holds that $A^{\mathcal{T}}(\{a, b, c, d\}, \{e\}) = 1$, while $A^{\mathcal{T}}(\{e\}, \{f\}) = 0$. For a more formal definition, refer to Definition A.3.

More specifically, our implementation of the graph coarsening function $\mathcal{T}$ employs spectral clustering[7] [33] to partition the graph into $T$ clusters, which in our framework controls the size of the bag. This results in a coarsened graph with fewer nodes and edges than $G$. We highlight and stress that the space complexity of this sparse graph, $\mathcal{T}(G)$, is upper bounded by that of the original graph $G$ (we do not store $2^n$ nodes).

**Defining the (coarse) product graph $\mathcal{T}(G) \square G$.** We define the connectivity of the product graph, see Figure 1(right), by applying the cartesian product between the coarsened graph, $\mathcal{T}(G)$, and the original graph, $G$. The product graph is denoted by $\mathcal{T}(G) \square G$, and is represented by the matrices $\mathcal{A}_{\mathcal{T}(G) \square G} \in \mathbb{R}^{(2^n \times n) \times (2^n \times n)}$ and $\mathcal{X} \in \mathbb{R}^{2^n \times n \times d}$[8], where by recalling Equation (1), we obtain,

$$\mathcal{A}_{\mathcal{T}(G) \square G} = \overbrace{A^{\mathcal{T}} \otimes I}^{\triangleq \mathcal{A}_{\mathcal{T}(G)}} + \overbrace{I \otimes A}^{\triangleq \mathcal{A}_G}. \tag{4}$$

The connectivity in this product graph induces the horizontal ($\mathcal{A}_G$) and vertical updates ($\mathcal{A}_{\mathcal{T}(G)}$) in Equation (2), visualized in Figure 1(right) via grey and red edges, respectively.

---

[5] For some of the theoretical analysis, this pooling operation is expressed as: $\rho(\mathcal{X}^{\mathrm{T}}) = \mathtt{MLP}^{\mathrm{T}} \big( \sum_S \big( \mathtt{MLP}^{\mathrm{T}} \big( \sum_{v=1}^n \mathcal{X}^{\mathrm{T}}(S, v) \big) \big) \big)$

[6] Our method also supports the case of which it is not a partition.

[7] Other graph coarsening or clustering algorithms can be readily used as well.

[8] We note that while the node matrix of the product graph, $\mathcal{X}$, can be initialized in various ways, e.g., deep sets-based architecture [37], in our implementation we simply use the original node features, i.e., $\mathcal{X}(S, v) = X(v)$, given that $X$ is the node feature matrix of the original graph.

## 4.2 Symmetry-based updates

In the previous subsection, we used a combination of a coarsening function and the graph Cartesian product to derive the two induced connectivities $\mathcal{A}_G, \mathcal{A}_{\mathcal{T}(G)}$ of our product graph. We use these connectivities to to perform message-passing on our product graph (see Equation (2)).

Inspired by recent literature on Subgraph GNNs [12, 3, 38], which incorporates and analyzes additional non-local updates arising from various symmetries (e.g., updating a node's representation via all nodes in its subgraphs), this section aims to identify potential new updates that can be utilized over our product graph. To that end, we study the symmetry structure of the node feature tensor in our product graph, $\mathcal{X}(S, v)$.The new updates described below will result in the third term in Equation (2), dubbed *Symmetry-based updates* ($\mathcal{A}_{\text{Equiv}}$). For better clarity in this derivation, we change the notation from nodes ($v$) to indices ($i$).

### 4.2.1 Symmetries of our product graph

Since the order of nodes in the original graph $G$ is arbitrary, each layer in our architecture must exhibit equivariance to any induced changes in the product graph. This requires maintaining equivariance to permutations of nodes in both the original graph and its transformation $\mathcal{T}(G)$. As a result, recalling that $\mathcal{A} \in \mathbb{R}^{(2^n \times n) \times (2^n \times n)}$ and $\mathcal{X} \in \mathbb{R}^{2^n \times n \times d}$ represent the adjacency and feature matrices of the product graph, the symmetries of the product graph are defined by an action of the symmetric group $S_n$. Formally, a permutation $\sigma \in S_n$ acts on the adjacency and feature matrices by:

$$(\sigma \cdot \mathcal{A})\big(S_1, i_1, S_2, i_2\big) = \mathcal{A}\big(\sigma^{-1}(S_1), \sigma^{-1}(i_1), \sigma^{-1}(S_2), \sigma^{-1}(i_2)\big), \tag{5}$$

$$(\sigma \cdot \mathcal{X})(S, i) = \mathcal{X}\big(\sigma^{-1}(S), \sigma^{-1}(i)\big), \tag{6}$$

where we define the action of $\sigma \in S_n$ on a set $S = \{i_1, i_2, \ldots, i_k\}$ of size $k$ as: $\sigma \cdot S := \{\sigma^{-1}(i_1), \sigma^{-1}(i_2), \ldots, \sigma^{-1}(i_k)\} := \sigma^{-1}(S)$.

### 4.2.2 Derivation of linear equivariant layers for the node feature tensor

We now characterize the linear equivariant layers with respect to the symmetry defined above, focusing on Equation (6). We adopt a similar notation to [22], and assume for simplicity that the number of feature channels is $d = 1$ (extension to multiple features is straightforward [22]). In addition, our analysis considers the case where $V^{\mathcal{T}}$ encompasses all potential super-nodes formed by subsets of $[n]$ (i.e we use the sparse coarsened adjacency[9]).

Our main tool is the characterization of linear equivariant layers for permutation symmetries as parameter-sharing schemes [34, 30, 22]. In a nutshell, this characterization states that the parameter vectors of the biases, invariant layers, and equivariant layers can be expressed as a learned weighted sum of basis tensors, where the basis tensors are indicators of the orbits induced by the group action on the respective index spaces. We focus here on presenting the final results and summarize them in Proposition 4.1 at the end of this subsection. Detailed discussion and derivations are available in Appendix E.

**Equivariant bias and invariant layers.** The bias vectors of the linear layers in our space are in $\mathbb{R}^{2^n \times n}$. As shown in Figure 3(right), the set of orbits induced by the action of $S_n$ satisfies:

$$(\mathcal{P}([n]) \times [n])/S_n := \{\gamma^{k^*} : k = 1, \ldots, n; * \in \{+, -\}\}. \tag{7}$$

Here, $\gamma^{k^+}$ corresponds to all pairs $(S, i) \in \mathcal{P}([n]) \times [n]$ with $|S| = k$ and $i \notin S$, and $\gamma^{k^-}$ to all pairs with $|S| = k$ and $i \in S$.

As stated in [34, 30, 22], the tensor set $\{\mathbf{B}_{S,i}^{\gamma}\}_{\gamma \in (\mathcal{P}([n]) \times [n])/S_n}$ where:

$$\mathbf{B}_{S,i}^{\gamma} = \begin{cases} 1, & \text{if } (S, i) \in \gamma; \\ 0, & \text{otherwise.} \end{cases} \tag{8}$$

are a basis of the space of bias vectors of the invariant linear layers induced by the action of $S_n$.

---

[9]This is because the action of $S_n$ is well defined over the index set $P(V[[n]]) \times [n]$ but not over $V \times V^{\mathcal{T}}$

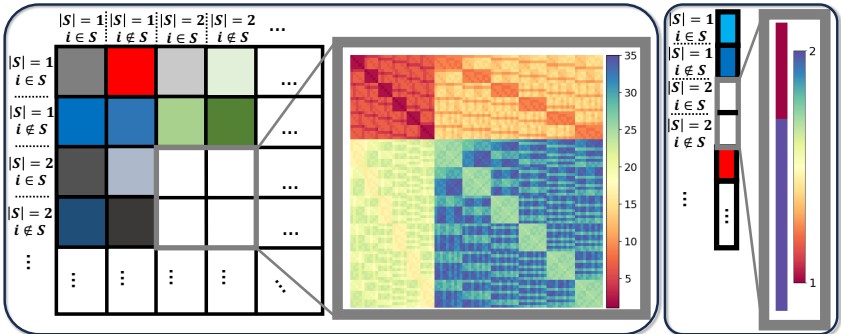

Figure 3: Visualization via heatmaps (different colors correspond to different parameters) of the parameter-sharing scheme determined by symmetries for a graph with $n = 6$ nodes, zooming-in on the block which corresponds to sets of size two. **Left:** Visualization of the weight matrix for the equivariant basis $\mathbf{B}^{\Gamma}_{S_1,i_1;S_2,i_2}$ (a total of 35 parameters in the block). **Right:** Visualization of the bias vector for the invariant basis $\mathbf{B}^{\gamma}_{S,i}$ (a total of 2 parameters in the block). Symmetry-based updates reduce parameters more effectively than previously proposed linear equivariant layers by treating indices as unordered tuples (see Appendix E.3 for a discussion).

**Weight matrices.** Following similar reasoning, consider elements $(S_1, i_1, S_2, i_2) \in (\mathcal{P}([n]) \times [n] \times \mathcal{P}([n]) \times [n])$. In Appendix E we characterize the orbits of $S_n$ in this space as a partition in which each partition set is defined according to six conditions. Some of these conditions include the sizes of $S_1$, $S_2$ and $S_1 \cap S_2$, which remain invariant under permutations. Given an orbit, $\Gamma \in (\mathcal{P}([n]) \times [n] \times \mathcal{P}([n]) \times [n])/S_n$, we define a basis tensor, $\mathbf{B}^{\Gamma} \in \mathbb{R}^{2^n \times n \times 2^n \times n}$ by setting:

$$\mathbf{B}^{\Gamma}_{S_1,i_1;S_2,i_2} = \begin{cases} 1, & \text{if } (S_1, i_1, S_2, i_2) \in \Gamma; \\ 0, & \text{otherwise.} \end{cases} \tag{9}$$

A visualization of the two basis vectors in Equations (8) and (9), is available in Figure 3. The following (informal) proposition summarizes the results in this section (the proof is given in Appendix G),

**Proposition 4.1** (Basis of Invariant (Equivariant) Layers). *The tensors $\mathbf{B}^{\gamma}$ ($\mathbf{B}^{\Gamma}$) in Equation (8) (Equation (9)) form an orthogonal basis (in the standard inner product) of the invariant layers and biases (Equivariant layers – weight matrix).*

### 4.2.3 Incorporating symmetry-based updates in our framework

In the previous subsection, we derived all possible linear invariant and equivariant operations that respect the symmetries of our product graph. We now use this derivation to define the symmetry-based updates in Equation (2), which correspond to the construction of $\mathcal{A}_{\text{Equiv}}$ and the application of an MPNN.

To begin, we note that any linear equivariant layer can be realized through an MPNN [13] applied to a fully connected graph with appropriate edge features. This is formally stated in Lemma F.1, the main idea is to encode the parameters on the edges of this graph (see visualization inset). Thus, the natural

$$x_i = \sum_j \mathbf{W}_{i,j} x_j \iff$$

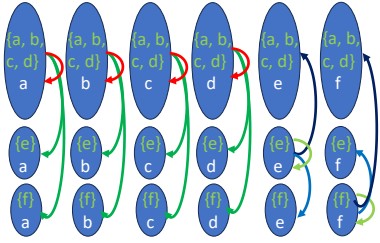

construction of $\mathcal{A}_{\text{Equiv}}$ corresponds to a fully connected graph, with appropriate edge features derived from the parameter-sharing scheme we have developed.

However, one of our main goals and guidelines in developing our flexible framework is to maintain efficiency, and to align with the (node-based) maximally expressive GNN, namely GNN-SSWL+ [38, 3], for the case of a trivial coarsening function, $\mathcal{T}(G) = G$ (which correspond to the full-bag setting). To achieve this, we opt for a sparser choice by using only a subset of the basis vectors (defined in Equation (9)) to construct $\mathcal{A}_{\text{Equiv}}$. Specifically, the matrix $\mathcal{A}_{\text{Equiv}}$ corresponding to the chosen subset of basis vectors is visualized inset – the parameter-sharing scheme is represented by edges with matching

colors. To clarify, the nodes $(S, v)$ that satisfy $v \in S$ "send messages" (i.e., broadcast their representation) to all the nodes $(S', v')$ such that $v = v'$. A more formal discussion regarding our implementation of those symmetry based updates is given in Appendix F.4.

**Maintaining sparsity.** While the updates above are defined over the sparse representation of the coarse product graph, in practice we use its dense representation, treating it as a graph over the set of nodes $V \times V^\mathcal{T}$, which requires space complexity $\mathcal{O}(T \cdot |V|)$. The update rules above are adapted to this representation simply by masking all nodes $(S, v)$ in the sparse representation such that $S \notin V^\mathcal{T}$. We note the models using the resulting update rule remain invariant to the action of $S_n$. See discussion in [1].

### 4.3 Marking Strategies and Theoretical Analysis

One of the key components of subgraph architectures is their marking strategy. Two widely used approaches in node-based subgraph architectures are binary-based node marking [4] and distance-based marking [38], which were proven to be equally expressive in the full-bag setup [38]. Empirically, distance-based marking has been demonstrated to outperform other strategies across several standard benchmarks. In this section, our aim is to develop and theoretically justify an appropriate marking strategy, specifically tailored to the structure of our product graph. We present and discuss here our main results, and refer to Appendix C for a more formal discussion.

Building on existing marking strategies and considering the unique structure of our product graph, we propose two natural extensions to both the binary node marking [4] and distance-based marking strategies [38]. Extending binary node marking, we first suggest **Simple Marking** ($\pi_S$), where an element $(S, v)$ is assigned a binary feature that indicates whether node $v$ belongs to subgraph $S$ ($v \in S$). The second extension, **Node + Size Marking** ($\pi_{SS}$), builds on the *simple marking* by assigning an additional feature that encodes the size of the super-node $S$.

For distance-based strategies, we propose **Minimum Distance** ($\pi_{MD}$), where each element $(S, v)$ is assigned the smallest (minimal) shortest path distance (SPD) from node $v$ to any node $u \in S$. Finally, **Learned Distance Function** ($\pi_{LD}$) extends this further by assigning to each element $(S, v)$ the output of a permutation-invariant learned function, which takes the set of SPDs between node $v$ and the nodes in $S$ as input.

Surprisingly, unlike the node-based full-bag case, we find that these marking strategies are not all equally expressive. We conveniently gather the first three strategies as $\Pi = \{\pi_S, \pi_{SS}, \pi_{MD}\}$ and summarize the relation between all variants as follows:

**Proposition 4.2** (Informal – Expressivity of marking strategies.). *(i) Strategies in $\Pi$ are all equally expressive, independently of the transformation function $\mathcal{T}$. (ii) The strategy $\pi_{LD}$ is at least as expressive as strategies in $\Pi$. Additionally, there exists transformation functions s.t. it is* strictly more expressive *than all of them.*

The above is formally stated in Propositions C.1 and C.2, and more thoroughly discussed in Appendix C. In light of the above proposition, we instatiate the learned distance function $\pi_{LD}$ strategy when implementing our model, as follows,

$$\mathcal{X}_{S,v} \leftarrow \sum_{u \in S} z_{d_G(v,u)} \tag{10}$$

where $d_G(v, u)$ denotes the *shortest path distance* between nodes $v$ and $u$ in the original $G^{10}$.

**Coarsening Function and Expressivity.** We investigate whether our CS-GNN framework offers more expressiveness compared to directly integrating information between the coarsened graph and the original graph.

The two propositions below illustrate that a simple, straight forward integration of the coarsen graph with the original graph (this integration is referred to as the *sum graph* – formally defined in Definition D.2), and further processing it via standard message-passing, results in a less expressive architecture. Furthermore, when certain coarsening functions are employed within the CS-GNN framework, our resulting architecture becomes strictly more expressive than conventional node-based

---

[10]To facilitate this, we maintain a lookup table where each index corresponds to a shortest path distance, assigning a learnable embedding, $z_{d_G(v,u)} \in \mathbb{R}^d$, to each node $(S, v)$.

subgraph GNNs. These results suggest that the interplay between the coarsening function and the subgraph layers we have developed enhances the model's overall performance. We summarize this informally below and provide a more formal discussion in Appendix D.

**Proposition 4.3** (Informal – CS-GNN goes beyond coarsening). *For any transformation function $\mathcal{T}$, CS-GNN can implement message-passing on the sum graph, hence being at least as expressive. Also, there exist transformations $\mathcal{T}$'s s.t. CS-GNN is* strictly more expressive *than that.*

**Proposition 4.4** (Informal – CS-GNN vs node based subgraphs). *There exist transformations $\mathcal{T}$'s s.t. our CS-GNN model using $\mathcal{T}$ as its coarsening function is* strictly more expressive *than GNN-SSWL+.*

# 5 Experiments

We experimented extensively over seven different datasets to answer the following questions: *(Q1) Can* `CS-GNN` *outperform efficient Subgraph GNNs operating on small bags? (Q2) Does the additional symmetry-based updates boost performance? (Q3) Does* `CS-GNN` *offer a good solution in settings where full-bag Subgraph GNNs cannot be applied? (Q4) Does* `CS-GNN` *in the full-bag setting validate its theory and match state-of-the-art full-bag Subgraph GNNs?*

In the following sections, we present our main results and refer to Appendix F for additional experiments and details.

**Baselines.** For each task, we include several baselines. The RANDOM baseline corresponds to random subgraph selection. We report the best performing random baseline from all prior work [5, 20, 29, 3]. The other two (non-random) baselines are: (1) LEARNED [5, 20, 29], which represents methods that learn the specific subgraphs to be used; and (2) FULL [38, 3], which corresponds to full-bag Subgraph GNNs.

**ZINC.** We experimented with both the ZINC-12K and ZINC-FULL datasets [31, 14, 10], adhering to a $500k$ parameter budget as prescribed. As shown in Table 1,

Table 1: Results on ZINC-12K dataset. Top two results are reported as **First** and **Second**.

| Method | Bag size | ZINC (MAE $\downarrow$) |
|---|---|---|
| GIN [35] | $T = 1$ | $0.163 \pm 0.004$ |
| OSAN [29] | $T = 2$ | $0.177 \pm 0.016$ |
| Random [20] | $T = 2$ | $0.131 \pm 0.005$ |
| PL [5] | $T = 2$ | $0.120 \pm 0.003$ |
| Mag-GNN [20] | $T = 2$ | $\mathbf{0.106} \pm 0.014$ |
| Ours | $T = 2$ | $\mathbf{0.109} \pm 0.005$ |
| Random [20] | $T = 3$ | $0.124 \pm N/A$ |
| Mag-GNN [20] | $T = 3$ | $\mathbf{0.104} \pm N/A$ |
| Ours | $T = 3$ | $\mathbf{0.096} \pm 0.005$ |
| Random [20] | $T = 4$ | $0.125 \pm N/A$ |
| Mag-GNN [20] | $T = 4$ | $\mathbf{0.101} \pm N/A$ |
| Ours | $T = 4$ | $\mathbf{0.090} \pm 0.003$ |
| Random [5] | $T = 5$ | $0.113 \pm 0.006$ |
| PL [5] | $T = 5$ | $\mathbf{0.109} \pm 0.005$ |
| Ours | $T = 5$ | $\mathbf{0.095} \pm 0.003$ |
| GNN-SSWL+ [38] | Full | $0.070 \pm 0.005$ |
| Subgraphormer [3] | Full | $0.067 \pm 0.007$ |
| Subgraphormer+PE [3] | Full | $\mathbf{0.063} \pm 0.001$ |
| Ours | Full | $\mathbf{0.062} \pm 0.0007$ |

CS-GNN outperforms all efficient baselines by a significant margin, with at least a $+0.008$ MAE improvement for bag sizes $T \in \{3, 4, 5\}$. Additionally, in the full-bag setting, our method recovers state-of-the-art results. The results for ZINC-FULL are available in Table 8 in the Appendix.

**OGB.** We tested our framework on several datasets from the OGB benchmark collection [16]. Table 4 shows the performance of our method compared to both efficient and full-bag Subgraph GNNs. Our CS-GNN outperforms all baselines across all datasets for bag sizes $T \in \{2, 5\}$, except for the MOLHIV dataset with $T = 2$, where PL achieves the best results and our method ranks second. In the full-bag setting, CS-GNN is slightly outperformed by the top-performing Subgraph GNNs but still offers comparable results.

**Peptides.** We experimented on the PEPTIDES-FUNC and PEPTIDES-STRUCT datasets [9] – which full-bag Subgraph GNNs already struggle to process – evaluating CS-GNN's ability to scale to larger graphs. The results are summarized in Table 2. CS-GNN outperforms all MPNN variants, even when incorporating structural encodings such as GATEDGCN+RWSE. Additionally, our method surpasses the random[11] baseline on both datasets.

Table 2: Results on PEPTIDES dataset.

| Model $\downarrow$ / Dataset $\rightarrow$ | PEPTIDES-FUNC (AP $\uparrow$) | PEPTIDES-STRUCT (MAE $\downarrow$) |
|---|---|---|
| GCN [19] | $0.5930 \pm 0.0023$ | $0.3496 \pm 0.0013$ |
| GIN [35] | $0.5498 \pm 0.0079$ | $0.3547 \pm 0.0045$ |
| GatedGCN [7] | $0.5864 \pm 0.0077$ | $0.3420 \pm 0.0013$ |
| GatedGCN+RWSE [9] | $0.6069 \pm 0.0035$ | $0.3357 \pm 0.0006$ |
| Random [3] | $0.5924 \pm 0.005$ | $0.2594 \pm 0.0021$ |
| Ours | $\mathbf{0.6156} \pm 0.0080$ | $\mathbf{0.2539} \pm 0.0015$ |

---

[11]For the PEPTIDES datasets, we benchmarked our model against the random variant of `Subgraphormer + PE`, which similarly incorporates information from the Laplacian eigenvectors. To ensure a fair comparison,

**Ablation study – symmetry-based updates.** We assessed the impact of the symmetry-based update on the performance of CS-GNN. Specifically, we ask, *do the symmetry-based updates significantly contribute to the performance of CS-GNN?* To evaluate this, we conducted several experiments using the ZINC-12K dataset across various bag sizes, $T \in \{2, 3, 4, 5\}$, comparing CS-GNN with and without the symmetry-based up-

Table 3: Ablation study.

| Bag size | w/ | w/o |
|---|---|---|
| T=2 | **0.109**±0.005 | 0.143±0.003 |
| T=3 | **0.096**±0.005 | 0.101±0.006 |
| T=4 | **0.090**±0.003 | 0.106±0.001 |
| T=5 | **0.095**±0.003 | 0.104±0.005 |

date. The results are summarized in Table 3. It is clear that the symmetry-based updates play a key role in the performance of CS-GNN. For a bag size of $T = 2$, the inclusion of the symmetry-based update improves the MAE by a significant $0.034$. For other bag sizes, the improvements range from $0.005$ to $0.016$, clearly demonstrating the benefits of including the symmetry-based updates.

**Discussion.** In what follows, we address research questions **Q1** to **Q4**. (A1) Tables 1, 2 and 4 clearly demonstrate that we outperform efficient Subgraph GNNs (which operate on a small bag) in 10 out of 12 dataset and bag size combinations. (A2) Our ablation study on the ZINC-12K dataset, as shown in Table 3, clearly demonstrates the benefits of the symmetry-based updates across all the considered bag sizes. (A3) Table 2 demonstrates that CS-GNN provides an effective solution when the full-bag setting cannot be applied, outperforming all baselines. (A4) On the ZINC-12K dataset (see Table 1), CS-GNN achieves state-of-the-art results compared to Subgraph GNNs. On the OGBG datasets (see Table 4), our performance is comparable to these top-performing Subgraph GNNs.

Table 4: Results on OGB datasets. The top two results are reported as **First** and **Second**.

| Model ↓ / Dataset → | Bag size | MOLHIV (ROC-AUC ↑) | MOLBACE (ROC-AUC ↑) | MOLESOL (RMSE ↓) |
|---|---|---|---|---|
| GIN [35] | $T = 1$ | 75.58±1.40 | 72.97±4.00 | 1.173±0.057 |
| Random [5] | $T = 2$ | 77.55±1.24 | 75.36±4.28 | 0.951±0.039 |
| PL [5] | $T = 2$ | **79.13**±0.60 | **78.40**±2.85 | **0.877**±0.029 |
| Mag-GNN [20] | $T = 2$ | 77.12±1.13 | - | - |
| Ours | $T = 2$ | **77.72**±0.76 | **80.58**±1.04 | **0.850**±0.024 |
| OSAN [29] | $T = 3$ | - | - | **0.959**±0.184 |
| OSAN [29] | $T = 5$ | - | 76.30±3.00 | - |
| PL [5] | $T = 5$ | **78.49**±1.01 | **78.39**±2.28 | **0.883**±0.032 |
| Random [5] | $T = 5$ | 77.30±2.56 | 78.14±2.36 | 0.900±0.032 |
| Ours | $T = 5$ | **79.09**±0.90 | **79.64**±1.43 | **0.863**±0.029 |
| GNN-SSWL+ [38] | Full | **79.58**±0.35 | **82.70**±1.80 | 0.837±0.019 |
| Subgraphormer | Full | **80.38**±1.92 | 81.62±3.55 | 0.832±0.043 |
| Subgraphormer + PE | Full | 79.48±1.28 | **84.35**±0.65 | **0.826**±0.010 |
| Ours | Full | 79.44±0.87 | 80.71±1.76 | **0.814**±0.021 |

## 6  Conclusions

In this work, we employed graph coarsenings and leveraged the insightful connection between Subgraph GNNs and the graph Cartesian product to devise CS-GNN, a novel and flexible Subgraph GNN that can effectively generate and process any desired bag size. Several directions for future research remain open. Firstly, we experimented with spectral clustering based coarsening, but other strategies are possible and are interesting to explore. Secondly, in our symmetry-based updates, we have only considered a portion of the whole equivariant basis we derived: evaluating the impact of other basis elements deserve further attention, both theoretically and in practice. Finally, whether Higher-Order Subgraph GNNs can benefit from our developed parameter-sharing scheme remains an intriguing open question.

**Limitations.** Our method operates over a product graph. Although we provide control over the size of this product graph, achieving better performance requires a larger bag size. This can become a complexity bottleneck, particularly when the original graph is large.

## Acknowledgements

The authors are grateful to Beatrice Bevilacqua, for helpful discussions, and constructive conversations about the experiments. HM is the Robert J. Shillman Fellow and is supported by the Israel Science Foundation through a personal grant (ISF 264/23) and an equipment grant (ISF 532/23). FF is funded by the Andrew and Erna Finci Viterbi Post-Doctoral Fellowship; FF partially performed this work while visiting the Machine Learning Research Unit at TU Wien led by Prof. Thomas Gärtner.

---

we used a single vote and the same exact bag size of 35 subgraphs. Additionally, since `Subgraphormer + PE` employs GAT [32] as the underlying MPNN, we also utilized GAT for this specific experiment to maintain consistency and fairness.

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

## Appendix

The appendix is organized as follows:

- In Appendix A, we provide some basic definitions that will be used in later sections of the paper.
- In Appendix B we discuss some theoretical aspects of our model implementation, and its relation to Equation (2).
- In Appendix C we define four natural general node marking policies and analyze their theoretical effects on our model, as well as their relation to some node-based node marking policies. Finally, we provide a principled derivation of one of these policies using the natural symmetry of our base object.
- In Appendix D.1 we compare our model to node-based subgraph GNNs, which are the most widely used variant of subgraph GNNs. Additionally, we demonstrate that different choices of coarsening functions can recover various existing subgraph GNN designs.
- In Appendix D.2 we demonstrate how our model can leverage the information provided by the coarsening function in an effective way, comparing its expressivity to a natural baseline which also leverages the coarsening function. We show that for all coarsening functions, we are at least as expressive as the baseline and that for some coarsening functions, our model is strictly more expressive.
- In Appendix E we delve deeper into the characterization of all linear maps $L : \mathbb{R}^{\mathcal{P}([n]) \times [n]} \to \mathbb{R}^{\mathcal{P}([n]) \times [n]}$ that are equivariant to the action of the symmetric group.
- In Appendix F we provide experimental details to reproduce the results in Section 5, as well as a comprehensive set of ablation studies.
- In Appendix G we provide detailed proofs to all propositions in this paper.

## A  Basic Definitions

We devote this section to formally defining the key concepts of this paper, as well as introducing new useful notation. We start by defining the two principle components of our pipeline, the cartesian product graph and the coarsening function:

**Definition A.1** (Cartesian Product Graph). *Given two graphs $G_1$ and $G_2$, their Cartesian product $G_1 \square G_2$ is defined as:*

- *The vertex set $V(G_1 \square G_2) = V(G_1) \times V(G_2)$.*

- *Vertices $(u_1, u_2)$ and $(v_1, v_2)$ in $G_1 \square G_2$ are adjacent if:*
    - *$u_1 = v_1$ and $u_2$ is adjacent to $v_2$ in $G_2$, or*
    - *$u_2 = v_2$ and $u_1$ is adjacent to $v_1$ in $G_1$.*

**Definition A.2** (Coarsening Function). *A Coarsening function $\mathcal{T}(\cdot)$ is defined as a function that, given a graph $G = (V, E)$ with vertex set $V = [n]$ and adjacency matrix $A \in \mathbb{R}^{n \times n}$, takes $A$ as input and returns a set of "super-nodes" $\mathcal{T}(A) \subseteq \mathcal{P}([n])$. The function $\mathcal{T}(\cdot)$ is considered equivariant if, for any permutation $\sigma \in S_n$, the following condition holds:*

$$\mathcal{T}(\sigma \cdot A) = \sigma \cdot \mathcal{T}(A). \tag{11}$$

*Here, $\sigma \cdot A$, and $\sigma \cdot \mathcal{T}(A)$ represent the group action of the symmetric group $S_n$ on $\mathbb{R}^{n \times n}$, and $\mathcal{P}([n])$ respectively.*

A coarsening function allows us to naturally define a graph structure on the "super-nodes" obtained from a given graph in the following way:

**Definition A.3** (Coarsened Graph). *Given a coarsening function $\mathcal{T}(\cdot)$ and a graph $G = (V, E)$ with vertex set $V = [n]$, adjacency matrix $A \in \mathbb{R}^{n \times n}$, we abuse notation and define the coarsened graph $\mathcal{T}(G) = (V^{\mathcal{T}}, E^{\mathcal{T}})$ as follows:*

- *$V^{\mathcal{T}} = \mathcal{T}(A)$*

- $E^{\mathcal{T}} = \{\{S, S'\} \mid S, S' \in \mathcal{T}(A),\ \exists i \in S, i' \in S'\ s.t.\ A_{i,i'} = 1\}$.

*The adjacency matrix of the coarsened graph can be expressed in two ways. The dense representation* $A^{\mathcal{T}}_{dense} \in \mathbb{R}^{|V^{\mathcal{T}}| \times |V^{\mathcal{T}}|}$ *is defined by:*

$$A^{\mathcal{T}}_{dense}(S, S') = \begin{cases} 1 & \{S, S'\} \in E^{\mathcal{T}} \\ 0 & otherwise. \end{cases} \tag{12}$$

*The sparse representation* $A^{\mathcal{T}}_{sparse} \in \mathbb{R}^{\mathcal{P}([n]) \times \mathcal{P}([n])}$ *is defined by:*

$$A^{\mathcal{T}}_{sparse}(S, S') = \begin{cases} 1 & S, S' \in V^{\mathcal{T}}, \{S, S'\} \in E^{\mathcal{T}} \\ 0 & otherwise. \end{cases} \tag{13}$$

We note that if the coarsened graph $\mathcal{T}(G)$ has a corresponding node feature map $\mathcal{X} : V^{\mathcal{T}} \to \mathbb{R}^d$, it also has sparse and dense vector representations defined similarly. Though the dense representation seems more natural, the sparse representation is also useful, as the symmetric group $S_n$ acts on it by:

$$\sigma \cdot A^{\mathcal{T}}_{sparse}(S, S') = A^{\mathcal{T}}_{sparse}(\sigma^{-1}(S), \sigma^{-1}(S')). \tag{14}$$

When the type of representation is clear from context, we abuse notation and write $A^{\mathcal{T}}$. Note also that in the above discussion, we have used the term "node feature map". Throughout this paper, in order to denote the node features of a graph $G = (V, E)$ with $|V| = n$, we use both the vector representation $X \in \mathbb{R}^{n \times d}$ and the map representation $\mathcal{X} : V \to \mathbb{R}^d$ interchangeably. Now, recalling that our pipeline is defined to create and update a node feature map $\mathcal{X}(S, v)$ supported on the nodes of the product graph $G \square \mathcal{T}(G)$, we define a general node marking policy, the following way:

**Definition A.4** (General Node Marking Policy). *A general node marking policy* $\pi(\cdot, \cdot)$*, is a function which takes as input a graph* $G = (V, E)$*, and a coarsening function* $\mathcal{T}(\cdot)$*, and returns a node feature map* $\mathcal{X} : V^{\mathcal{T}} \times V \to \mathcal{R}^d$.

In Appendix C We provide four different node marking policies, and analyze the effect on our pipeline. We now move on to define the general way in which we update a given node feature map on the product graph.

**Definition A.5** (General CS-GNNLayer Update). *Given a graph* $G = (V, E)$ *and a coarsening function* $\mathcal{T}(\cdot)$*, let* $\mathcal{X}^t(S, v) : V \times V^{\mathcal{T}} \to \mathcal{R}^d$ *denote the node feature map at layer* $t$*. The general CS-GNNlayer update is defined by:*

$$\begin{aligned}
\mathcal{X}^{t+1}(S, v) = f^t \Big( & \mathcal{X}^t(S, v), \\
& agg_1^t \{\!\{ (\mathcal{X}^t(S, v'), e_{v,v'}) \mid v' \sim_G v \}\!\}, \\
& agg_2^t \{\!\{ (\mathcal{X}^t(S', v), \tilde{e}_{S,S'}) \mid S' \sim_{G^{\mathcal{T}}} S \}\!\}, \\
& agg_3^t \{\!\{ (\mathcal{X}^t(S', v), z(S, v, S', v)) \mid S' \in V^{\mathcal{T}} s.t.\ v \in S' \}\!\}, \\
& agg_4^t \{\!\{ (\mathcal{X}^t(S, v'), z(S, v, S, v')) \mid v' \in V s.t.\ v' \in S \}\!\} \Big).
\end{aligned} \tag{15}$$

*Here,* $f^t$ *is an arbitrary (parameterized) continuous function,* $agg_i^t$*,* $i = 1, \ldots 4$ *are learnable permutation invariant aggregation functions,* $e_{v,v'}, \tilde{e}_{S,S'}$ *are the (optional) edge features of* $G$ *and* $\mathcal{T}(G)$ *respectively and the function* $z : \mathcal{P}([n]) \times [n] \times \mathcal{P}([n]) \times [n] \to \mathbb{R}^d$ *maps each tuple of indices* $\mathbf{v} = (S, v, S', v')$ *to a vector uniquely encoding the orbit of* $\mathbf{v}$ *under the action of* $S_n$ *as described in 73.*

We note that for brevity, the notation used in the main body of the paper omits the aggregation functions $agg_1^t, \ldots, agg_4^t$ and the edge features from the formulation of some of the layer updates. However, we explicitly state each component of the update, as we heavily utilize them in later proofs. We also note that this update is different than the general layer update presented in Equation (2), as it doesn't use all global updates characterized in 9. The reason for this is that some of the global updates have an asymptotic runtime of $\tilde{\mathcal{O}}(n^2)$ where $n$ is the number of nodes in the input graph. As our goal was to create models that improve on the scalability of standard subgraph GNNs which have

an asymptotic runtime of $\tilde{\mathcal{O}}(n^2)$, We decided to discard some of the general global updates and keep only the ones that are induced by the last two entries in equation 15 which all have a linear runtime. After a stacking of the layers in Equation (15), we employ the following pooling procedure on the final node feature map $\mathcal{X}^T$:

$$\rho(\mathcal{X}^T) = \texttt{MLP}_2\left(\sum_{S \in V^{\mathcal{T}}}\left(\texttt{MLP}_1\big(\sum_{v \in V}\mathcal{X}^T(S, v)\big)\right)\right). \tag{16}$$

Finally, we define the set of all functions that can be expressed by our model:

**Definition A.6** (Expressivity of Family of Graph Functions). *Let $\mathcal{F}$ be a family of graph functions, we say that $\mathcal{F}$ can express a graph function $g(\cdot)$ if for every finite family of graphs $\mathcal{G}$ there exists a function $f \in \mathcal{F}$ such that:*

$$f(G) = g(G) \quad \forall G \in \mathcal{G}. \tag{17}$$

*Here, $\mathcal{G}$ is a finite family of graphs if all possible values of node/edge features of the graphs in $\mathcal{G}$ form a finite set, and the maximal size of the graphs within $\mathcal{G}$ is bounded.*

**Definition A.7** (Family of Functions Expressed By CS-GNN). *Let $\pi$ be a general node marking policy and $\mathcal{T}$ be a coarsening function. Define $\mathcal{S}(\mathcal{T}, \pi)$ to be the family of graph functions, which when given input graph $G = (V, E)$, first compute $\mathcal{X}^0(S, v)$ using $\pi(G, \mathcal{T})$, then update this node feature map by stacking $T$ layers of the form 15, and finally pooling $\mathcal{X}^0(S, v)$ using equation 16. We define CS-GNN$(\mathcal{T}, \pi)$ to be the set of all functions that can be expressed by $\mathcal{S}(\mathcal{T}, \pi)$.*

# B  Theoretical Validation of Implementation Details

In this section, we provide implementation details of our model and prove that they enable us to recover the conceptual framework of the model discussed thus far. First, we note that in Section 4.2, we characterized all equivariant linear maps $L : \mathbb{R}^{\mathcal{P}([n]) \times [n]} \to \mathbb{R}^{\mathcal{P}([n]) \times [n]}$ in order to incorporate them into our layer update. Given the high dimensionality of the space of all such linear maps, and in order to save parameters, we demonstrate that it is possible to integrate these layers into our layer update by adding edge features to a standard MPNN model. This is formalized in the following proposition:

**Lemma B.1** (Parameter Sharing as MPNN). *Let $B_1, \ldots B_k : \mathbb{R}^{n \times n}$ be orthogonal matrices with entries restricted to 0 or 1, and let $W_1, \ldots W_k \in \mathbb{R}^{d \times d'}$ denote a sequence of weight matrices. Define $B_+ = \sum_{i=1}^k B_i$ and choose $z_1, \ldots z_k \in \mathbb{R}^{d^*}$ to be a set of unique vectors representing an encoding of the index set. The function that represents an update via parameter sharing:*

$$f(X) = \sum_{i=1}^k B_i X W_i, \tag{18}$$

*can be implemented on any finite family of graphs $\mathcal{G}$, by a stack of MPNN layers of the following form [13],*

$$m_v^l = \sum_{u \in N_{B_+}(v)} M^l(X_u^l, e_{u,v}), \tag{19}$$

$$X_v^{l+1} = U^l(X_v^l, m_v^l), \tag{20}$$

*where $U^l, M^l$ are multilayer perceptrons (MLPs). The inputs to this MPNN are the adjacency matrix $B_+$, node feature vector $X$, and edge features – the feature of edge $(u, v)$ is given by:*

$$e_{u,v} = \sum_{i=1}^k z_i \cdot B_i(u, v). \tag{21}$$

*Here, $B_i(u, v)$ denotes the $(u, v)$ entry to matrix $B_i$.*

The proof is given in Appendix G. The analysis in Section 4.2 demonstrates that the basis of the space of all equivariant linear maps $L : \mathbb{R}^{\mathcal{P}([n]) \times [n]} \to \mathbb{R}^{\mathcal{P}([n]) \times [n]}$ satisfies the conditions of Lemma F.1. Additionally, we notice that some of the equivariant linear functions have an asymptotic runtime of

$\tilde{\mathcal{O}}(n^2)$ where $n$ is the number of nodes in the input graph. As our main goal is to construct a more scalable alternative to node-based subgraph GNNs, which also have a runtime of $\tilde{\mathcal{O}}(n^2)$, we limit ourselves to a subset of the basis for which all maps run in linear time. This is implemented by adding edge features to the adjacency matrices $A_{P_1}$ and $A_{P_2}$, defined later in this section.

We now move on to discussing our specific implementation of the general layer update from Definition A.5.

Given a graph $G = (V, E)$ and a coarsening function $\mathcal{T}$, we aim to implement this general layer update by combining several standard message passing updates on the product graph $G \square \mathcal{T}(G)$. In the next two definitions, we define the adjacency matrices supported on the node set $V \times V^{\mathcal{T}}$, which serve as the foundation for these message passing procedures, and formalize the procedures themselves.

**Definition B.1** (Adjacency Matrices on Product Graph). *Let $G = (V, E)$ be a graph with adjacency matrix $A$ and node feature vector $X$, and let $\mathcal{T}(\cdot)$ be a coarsening function. We define the following four adjacency matrices on the vertex set $V^{\mathcal{T}} \times V$:*

$$A_G(S, v, S', v') = \begin{cases} 1 & v \sim_G v', \ S = S' \\ 0 & otherwise. \end{cases} \tag{22}$$

$$A_{\mathcal{T}(G)}(S, v, S', v') = \begin{cases} 1 & S \sim_{\mathcal{T}(G)} S', \ v = v' \\ 0 & otherwise. \end{cases} \tag{23}$$

$$A_{P_1}(S, v, S', v') = \begin{cases} 1 & v \in S', \ v = v' \\ 0 & otherwise. \end{cases} \tag{24}$$

$$A_{P_2}(S, v, S', v') = \begin{cases} 1 & v' \in S, \ S' = S \\ 0 & otherwise. \end{cases} \tag{25}$$

*Given edge features $\{e_{v,v'} \mid v \sim_G v'\}$ and $\{\tilde{e}_{S,S'} \mid s \sim_{\mathcal{T}(G)} s'\}$ corresponding to the graphs $G$ and $\mathcal{T}(G)$, respectively, we can trivially define the edge features corresponding to $A_G$ and $A_{G\mathcal{T}}$ as follows:*

$$e_G(S, v, S', v') = e_{v,v'}, \tag{26}$$
$$e_{\mathcal{T}(G)}(S, v, S', v') = \tilde{e}_{S,S'}. \tag{27}$$

*In addition, for $i = 1, 2$, we define the edge features corresponding to adjacency matrices $A_{P_i}$ as follows:*
$$e_{P_i}(S, v, S', v') = z(S, v, S', v'). \tag{28}$$

*Here, the function $z : \mathcal{P}([n]) \times [n] \times \mathcal{P}([n]) \times [n] \to \mathbb{R}^d$ maps each tuple $\mathbf{v} = (S, v, S', v')$ to a vector uniquely encoding the orbit of $\mathbf{v}$ under the action of $S_n$ as described in Equation 73.*

**Definition B.2** (CS-GNN Update Implementation). *Given a graph $G = (V, E)$, and a coarsening function $\mathcal{T}(\cdot)$, let $A_1 \ldots A_4$ enumerate the set of adjacency matrices $\{A_G, A_{\mathcal{T}(G)}, A_{P_1}, A_{P_2}\}$. We define a CS-GNN layer update in the following way:*

$$\mathcal{X}_i^t(S, v) = U_i^t \left( (1 + \epsilon_i^t) \cdot \mathcal{X}^t(S, v) + \sum_{(S', v') \sim_{A_i}(S, v)} M^t(\mathcal{X}^t(S', v') + e_i(S, v, S', v')) \right). \tag{29}$$

$$\mathcal{X}^{t+1}(S, v) = U_{fin}^t \left( \sum_{i=1}^{4} \mathcal{X}_i^t(S, v) \right). \tag{30}$$

*Here $\mathcal{X}^t(S, v)$ and $\mathcal{X}^{t+1}(S, v)$ denote the node feature maps of the product graph at layers $t$ and $t + 1$, respectively. $e^1(S, v, S', v'), \ldots, e^4(S, v, S', v')$ denote the edge features associated with adjacency matrices $A_1, \ldots, A_4$. $\epsilon_1^t, \ldots, \epsilon_4^t$ represent learnable parameters in $\mathbb{R}$, and $U_1^t, \ldots, U_4^t$, $U_{fin}^t$, $M^t$ all refer to multilayer perceptrons.*

The next proposition states that using the layer update defined in equations 29 and 30 is enough to efficiently recover the general layer update defined in equation 15.

**Proposition B.1** (Equivalence of General Layer and Implemented Layer). *Let $\mathcal{T}(\cdot)$ be a coarsening function, $\pi$ be a generalized node marking policy, and $\mathcal{G}$ be a finite family of graphs. Applying a stack of $t$ general layer updates as defined in Equation 15 to the node feature map $\mathcal{X}(S, v)$ induced by $\pi(G, \mathcal{T})$, can be effectively implemented by applying a stack of $t$ layer updates specified in Equations 29 and 30 to $\mathcal{X}(S, v)$. Additionally, the depths of all MLPs that appear in 29 and 30 can be bounded by 4.*

## C Node Marking Policies – Theoretical Analysis

In this section, we define and analyze various general node marking policies, starting with four natural choices.

**Definition C.1** (Four General Node Marking policies). *Let $G = (V, E)$ be a graph with adjacency matrix $A \in \mathbb{R}^{n \times n}$ and node feature vector $X \in \mathbb{R}^{n \times d}$, and let $\mathcal{T}(\cdot)$ be a coarsening function. All of the following node marking policies take the form:*

$$\pi(G, \mathcal{T}) = \mathcal{X}(S, v) = [X_u, b_\pi(S, v)], \tag{31}$$

*where $[\cdot, \cdot]$ denotes the concatenation operator. We focus on four choices for $b_\pi(S, v)$:*

1. ***Simple Node Marking:***

$$b_\pi(S, v) = \begin{cases} 1 & \text{if } v \in S, \\ 0 & \text{if } v \notin S. \end{cases} \tag{32}$$

   *We denote this node marking policy by $\pi_S$.*

2. ***Node + Size Marking:***

$$b_\pi(S, v) = \begin{cases} (1, |S|) & \text{if } v \in S, \\ (0, |S|) & \text{if } v \notin S. \end{cases} \tag{33}$$

   *We denote this node marking policy by $\pi_{SS}$.*

3. ***Minimum Distance:***

$$b_\pi(S, v) = \min_{v' \in S} d_G(v, v') \tag{34}$$

   *where $d_G(v, v')$ is the shortest path distance between nodes $v$ and $v'$ in the original graph. We denote this node marking policy by $\pi_{MD}$.*

4. ***Learned Distance Function:***

$$b_\pi(S, v) = \phi(\{d_G(v, v') \mid v' \in S\}) \tag{35}$$

   *where $\phi(\cdot)$ is a learned permutation-invariant function. We denote this node marking policy by $\pi_{LD}$.*

We note that when using the identity coarsening function $\mathcal{T}(G) = G$, our general node marking policies output node feature maps supported on the product $V \times V$. Thus, they can be compared to node marking policies used in node-based subgraph GNNs. In fact, in this case, both $\pi_S$ and $\pi_{SS}$ reduce to classical node-based node marking, while $\pi_{MD}$ and $\pi_{LD}$ reduce to distance encoding. The definitions of these can be found in [38]. Interestingly, even though in the case of node-based subgraph GNNSs, both distance encoding and node marking were proven to be maximally expressive [38], in our case for some choices of $\mathcal{T}$, $\pi_{LD}$ is strictly more expressive than the other three choices. The exact effect of each generalized node marking policy on the expressivity of our model is explored in the following two propositions.

**Proposition C.1** (Equal Expressivity of Node Marking Policies). *For any coarsening function $\mathcal{T}(\cdot)$ the following holds:*

$$CS\text{-}GNN(\mathcal{T}, \pi_S) = CS\text{-}GNN(\mathcal{T}, \pi_{SS}) = CS\text{-}GNN(\mathcal{T}, \pi_{MD}). \tag{36}$$

**Proposition C.2** (Expressivity of Learned Distance Policy). *For any coarsening function $\mathcal{T}(\cdot)$ the following holds:*

$$CS\text{-}GNN(\mathcal{T}, \pi_S) \subseteq CS\text{-}GNN(\mathcal{T}, \pi_{LD}). \tag{37}$$

*In addition, for some choices of $\mathcal{T}(\cdot)$ the containment is strict.*

The proofs of both propositions can be found in Appendix G. Finally, we provide a principled approach to deriving a generalized node marking policy based on symmetry invariance, and prove its equivalence to $\pi_{SS}$. Given a graph $G = (V, E)$ with $V = [n]$, adjacency matrix $A$, and node feature vector $X \in \mathbb{R}^{n \times d}$, along with a coarsening function $\mathcal{T}(\cdot)$, We define an action of the symmetric group $S_n$ on the space $\mathbb{R}^{\mathcal{P}([n]) \times [n]}$ as follows:

$$\sigma \cdot \mathcal{X}(S, v) = \mathcal{X}(\sigma^{-1}(S), \sigma^{-1}(v)) \quad \text{for } \sigma \in S_n, \mathcal{X} \in \mathbb{R}^{\mathcal{P}([n]) \times [n]}. \tag{38}$$

Now, for each orbit $\gamma \in (\mathcal{P}([n]) \times [n])/S_n$, we define $\mathbf{1}_\gamma \in \mathbb{R}^{\mathcal{P}([n]) \times [n]}$ as follows:

$$\mathbf{1}_\gamma(S, v) = \begin{cases} 1 & (S, v) \in \gamma, \\ 0 & \text{otherwise.} \end{cases} \tag{39}$$

Choosing some enumeration of the orbit set $(\mathcal{P}([n]) \times [n])/S_n = \{\gamma_1, \ldots, \gamma_k\}$, We now define the invariant generalized node marking policy $\pi_{\text{inv}}$ by first setting:

$$b_{\pi_{\text{inv}}}^{\text{sparse}}(S, v) : \mathcal{P}([n]) \times [n] \to \mathbb{R}^k$$

and

$$b_{\pi_{\text{inv}}} : V^{\mathcal{T}} \times V \to \mathbb{R}^k$$

as follows:

$$b_{\pi_{\text{inv}}}^{\text{sparse}}(S, v) = [\mathbf{1}_{\gamma_1}(S, v), \ldots, \mathbf{1}_{\gamma_k}(S, v)] \qquad S \in \mathcal{P}(V),\ v \in V, \tag{40}$$

$$b_{\pi_{\text{inv}}}(S, v) = b_{\pi_{\text{inv}}}^{\text{sparse}}(S, v) \qquad S \in V^{\mathcal{T}},\ v \in V. \tag{41}$$

Then, we define the node feature map induced by $\pi_{\text{inv}}$ as:

$$\mathcal{X}^{\pi_{\text{inv}}}(S, v) = [X_v, b_{\pi_{\text{inv}}}(S, v)]. \tag{42}$$

Interestingly, $\pi_{\text{inv}}$, derived solely from the group action of $S_n$ on $\mathcal{P}([n]) \times [n]$, is equivalent to the generalized node marking policy $\pi_{SS}$. This is stated more rigorously in the following proposition:

**Proposition C.3** (Node + Size Marking as Invariant Marking). *Given a graph $G = (V, E)$ with node feature vector $X \in \mathbb{R}^{n \times d}$, and a coarsening function $\mathcal{T}(\cdot)$, let $\mathcal{X}^{\pi_{SS}}, \mathcal{X}^{\pi_{inv}}$ be the node feature maps induced by $\pi_{SS}$ and $\pi_{inv}$ respectively. Recall that:*

$$\mathcal{X}^{\pi_{SS}}(S, v) = [X_v, b_{\pi_{SS}}(S, v)], \tag{43}$$

$$\mathcal{X}^{\pi_{inv}}(S, v) = [X_v, b_{\pi_{inv}}(S, v)]. \tag{44}$$

*The following now holds:*

$$b_{\pi_{inv}}(S, v) = OHE(b_{\pi_{SS}}(S, v)) \quad \forall S \in V^{\mathcal{T}}, \forall v \in V. \tag{45}$$

*Here, OHE denotes a one-hot encoder, independent of the choice of both $G$ and $\mathcal{T}$.*

The proof of proposition C.3 can be found in Appendix G.

# D  Expressive Power of CS-GNN

## D.1  Recovering Subgraph GNNs

In this section, we demonstrate that by choosing suitable coarsening functions, our architecture can replicate various previous subgraph GNN designs. We begin by focusing on node-based models, which are the most widely used type. We define a variant of these models which was proven in [38] to be maximally expressive, and show that our approach can recover it.

**Definition D.1** (Maximally Expressive Subgraph GNN). *We define $MSGNN(\pi_{NM})$ as the set of all functions expressible by the following procedure:*

1. ***Node Marking:*** *The representation of tuple* $(u, v) \in V \times V$ *is initially given by:*

$$\mathcal{X}^0(u, v) = \begin{cases} 1 & \text{if } u = v, \\ 0 & \text{if } u \neq v. \end{cases} \tag{46}$$

2. ***Update:*** *The representation of tuple* $(u, v)$ *is updated according to:*

$$\begin{aligned} \mathcal{X}^{t+1}(u, v) = f^t\Big( &\mathcal{X}^t(u, v), \mathcal{X}^t(u, u), \mathcal{X}^t(v, v), \\ &agg_1^t \{\!\!\{ (\mathcal{X}^t(u, v'), e_{v,v'}) \mid v' \sim v \}\!\!\}, \\ &agg_2^t \{\!\!\{ (\mathcal{X}^t(v, u'), e_{u,u'}) \mid u' \sim u \}\!\!\} \Big). \end{aligned} \tag{47}$$

3. ***Pooling:*** *The final node feature vector* $\mathcal{X}^T(u, v)$ *is pooled according to:*

$$MLP_2 \left( \sum_{u \in V} MLP_1 \left( \sum_{v \in V} \mathcal{X}^T(u, v) \right) \right). \tag{48}$$

*Here, for any* $t \in [T]$, $f^t$ *is any continuous (parameterized) functions,* $agg_1^t, , agg_2^t$ *are any continuous (parameterized) permutation-invariant functions and* $MLP_1, MLP_2$ *are multilayer preceptrons.*

**Proposition D.1** (CS-GNN Can Implement MSGNN)**.** *Let* $\mathcal{T}(\cdot)$ *be the identity coarsening function defined by:*

$$\mathcal{T}(G) = \{\{v\} \mid v \in V\} \quad \forall G = (V, E). \tag{49}$$

*The following holds:*

$$\text{CS-GNN}(\mathcal{T}, \pi_S) = \text{MSGNN}(\pi_{NM}). \tag{50}$$

The proof of proposition D.1 can be found in Appendix G. We observe that, similarly, by selecting the coarsening function:

$$\mathcal{T}(G) = E \quad \forall G = (V, E), \tag{51}$$

one can recover edge-based subgraph GNNs. An example of such a model is presented in [4] (DS-GNN), where it was proven capable of distinguishing between two 3-WL indistinguishable graphs, despite having an asymptotic runtime of $\tilde{\mathcal{O}}(m^2)$, where $m$ is the number of edges in the input graph. This demonstrates our model's ability to achieve expressivity improvements while maintaining a (relatively) low asymptotic runtime by exploiting the graph's sparsity through the coarsening function. Finally, we note that by selecting the coarsening function:

$$\mathcal{T}(G) = \{S \in \mathcal{P}(V) \mid |S| = k\} \quad G = (V, E), \tag{52}$$

We can recover an unordered variant of the $k$-OSAN model presented in [29].

## D.2 Comparison to Natural Baselines

In this section, we demonstrate how our model can leverage the information provided by the coarsening function $\mathcal{T}(\cdot)$ in an effective way. First, we define a baseline model that incorporates $\mathcal{T}$ in a straightforward manner. We then prove that, for any $\mathcal{T}(\cdot)$, our model is at least as expressive as this baseline. Additionally, we show that for certain choices of $\mathcal{T}(\cdot)$, our model exhibits strictly greater expressivity. To construct the baseline model, we first provide the following definition:

**Definition D.2** (Coarsened Sum Graph)**.** *Given a graph* $G = (V, E)$ *and a coarsening function* $\mathcal{T}(\cdot)$, *we define the coarsened sum graph* $G_+^{\mathcal{T}} = (V_+^{\mathcal{T}}, E_+^{\mathcal{T}})$ *by:*

- $V_+^{\mathcal{T}} = V \cup V^{\mathcal{T}}$.

- $E_+^{\mathcal{T}} = E \cup E^{\mathcal{T}} \cup \{\{S, v\} \mid S \in V^{\mathcal{T}}, v \in V \; v \in S\}$.

*If graph* $G$ *had a node feature vector* $X \in \mathbb{R}^{n \times d}$, *we define the node feature vector of* $G_+^{\mathcal{T}}$ *as:*

$$X_v = \begin{cases} [X_v, 1] & v \in V \\ 0_{d+1} & v \in V^{\mathcal{T}} \end{cases}. \tag{53}$$

*Here we concatenated a 1 to the end of node features of* $V$ *to distinguish them from the nodes of* $V^{\mathcal{T}}$.

The connectivity of the sum graph (for our running example Figure 1) is visualized inset.

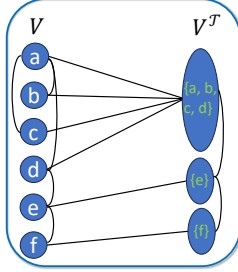

We now define our baseline model:

**Definition D.3** (Coarse MPNN). *Let $\mathcal{T}(\cdot)$ be a coarsening function. Define $MPNN_+(\mathcal{T})$ as the set of all functions which can be expressed by the following procedure:*

1. **Preprocessing:** *We first construct the sum graph $G_+^{\mathcal{T}}$ of the input graph G, along with a node feature map $\mathcal{X}^0 : V_+^{\mathcal{T}} \to \mathbb{R}^d$ defined according to equation 53.*

2. **Update:** *The representation of node $v \in V_+^{\mathcal{T}}$ is updated according to:*

$$
\begin{aligned}
\text{For } v \in V : \quad & \mathcal{X}^{t+1}(v) = f_V^t \left( \mathcal{X}^t(v), agg_1^t \{\!\{ (\mathcal{X}^t(u), e_{u,v}) \mid u \sim_G v \}\!\}, \right. \\
& \left. agg_2^t \{\!\{ \mathcal{X}^t(S) \mid S \in V^T, v \in S \}\!\} \right), \\
\text{For } S \in V^{\mathcal{T}} : \quad & \mathcal{X}^{t+1}(S) = f_{V^{\mathcal{T}}}^t \left( \mathcal{X}^t(S), agg_1^t \{\!\{ (\mathcal{X}^t(S'), e_{S,S'}) \mid S' \sim_{\mathcal{T}(G)} S \}\!\}, \right. \\
& \left. agg_2^t \{\!\{ \mathcal{X}^t(v) \mid v \in V, v \in S \}\!\} \right).
\end{aligned}
$$
(54)

3. **Pooling:** *The final node feature vector $\mathcal{X}^T(\cdot)$ is pooled according to:*

$$
MLP \left( \sum_{v \in V_+^{\mathcal{T}}} \mathcal{X}^T(v) \right).
$$
(55)

*Here, for $t \in [T]$, $f_V^t$, $f_{V^{\mathcal{T}}}^t$ are continuous (parameterized) functions and , $agg_1^t, agg_2^t T$ are continuous (parameterized) permutation invariant functions. Finally, we notice that for the trivial coarsening function defined by*

$$
\mathcal{T}_\emptyset(G) = \emptyset,
$$
(56)

*the update in Equation (54) devolves into a standard MPNN update, as defined in [13] and so we define:*

$$
MPNN = MPNN_+(\mathcal{T}_\emptyset).
$$
(57)

In essence, given an input graph $G = (V, E)$, the $MPNN_+(\mathcal{T})$ pipeline first constructs the coarsened graph $\mathcal{T}(G)$. It then adds edges between each super-node $S \in V^{\mathcal{T}}$ and the nodes it is comprised of (i.e., any $v \in S$). This is followed by a standard message passing procedure on the graph. The following two propositions suggest that this simple approach to incorporating $\mathcal{T}$ into a GNN pipeline is less powerful than our model.

**Proposition D.2** (CS-GNN Is at Least as Expressive as Coarse MPNN ). *For any coarsening function $\mathcal{T}(\cdot)$ the following holds:*

$$
MPNN \subseteq MPNN_+(\mathcal{T}) \subseteq CS\text{-}GNN(\mathcal{T}, \pi_S)
$$
(58)

**Proposition D.3** (CS-GNN Can Be More Expressive Than MPNN+). *Let $\mathcal{T}(\cdot)$ be the identity coarsening function defined by:*

$$
\mathcal{T}(G) = \{\{v\} \mid v \in V\} \quad G = (V, E).
$$
(59)

*The following holds:*

$$
MPNN = MPNN_+(\mathcal{T}).
$$
(60)

*Thus:*

$$
MPNN_+(\mathcal{T}) \subset CS\text{-}GNN(\mathcal{T}, \pi_S),
$$
(61)

*where this containment is strict.*

The proofs to the last two propositions can be found in Appendix G. Proposition D.3 demonstrates that CS-GNNis strictly more expressive than $\text{MPNN}_+$ when using the identity coarsening function. However, this result extends to more complex coarsening functions as well. We briefly discuss one such example. Let $\mathcal{T}(\cdot)$ be the coarsening function defined by:

$$\mathcal{T}_\triangle(G) = \{v_1, v_2, v_3 \mid G[v_1, v_2, v_3] \cong \triangle\}, \tag{62}$$

i.e. for an input graph $G$, the set of super-nodes is composed of all triplets of nodes whose induced subgraph is isomorphic to a triangle. To see that CS-GNN is strictly more expressive then $\text{MPNN}_+$ when using $\mathcal{T}_\triangle(\cdot)$, we look at the two graphs $G$ and $H$ depicted in Figure 4. In the figure, we see the two original graphs, $G$ and $H$, their corresponding sum graphs $G_+^{\mathcal{T}_\triangle}$ and $H_+^{\mathcal{T}_\triangle}$, and a subgraph of their corresponging product graphs $G \square \mathcal{T}_\triangle(G)$ and $H \square \mathcal{T}_\triangle(H)$ induced by the sets $\{(S_0, v) \mid v \in V_G\}$ and $\{(S_0, v) \mid v \in V_H\}$ respectively (this can be thought of as looking at a single subgraph from the bag of subgraphs induced by CS-GNN). One can clearly see that both the original graphs and their respective sum graphs are 1-WL indistinguishable. On the other hand, the subgraphs induced by our method are 1-WL distinguishable. Since for both $G$ and $H$ the "bag of subgraphs" induced by CS-GNN is composed of 6 isomorphic copies of the same graph, this would imply that our method can distinguish between $G$ and $H$, making it strictly mor expressive then $\text{MPNN}_+$.

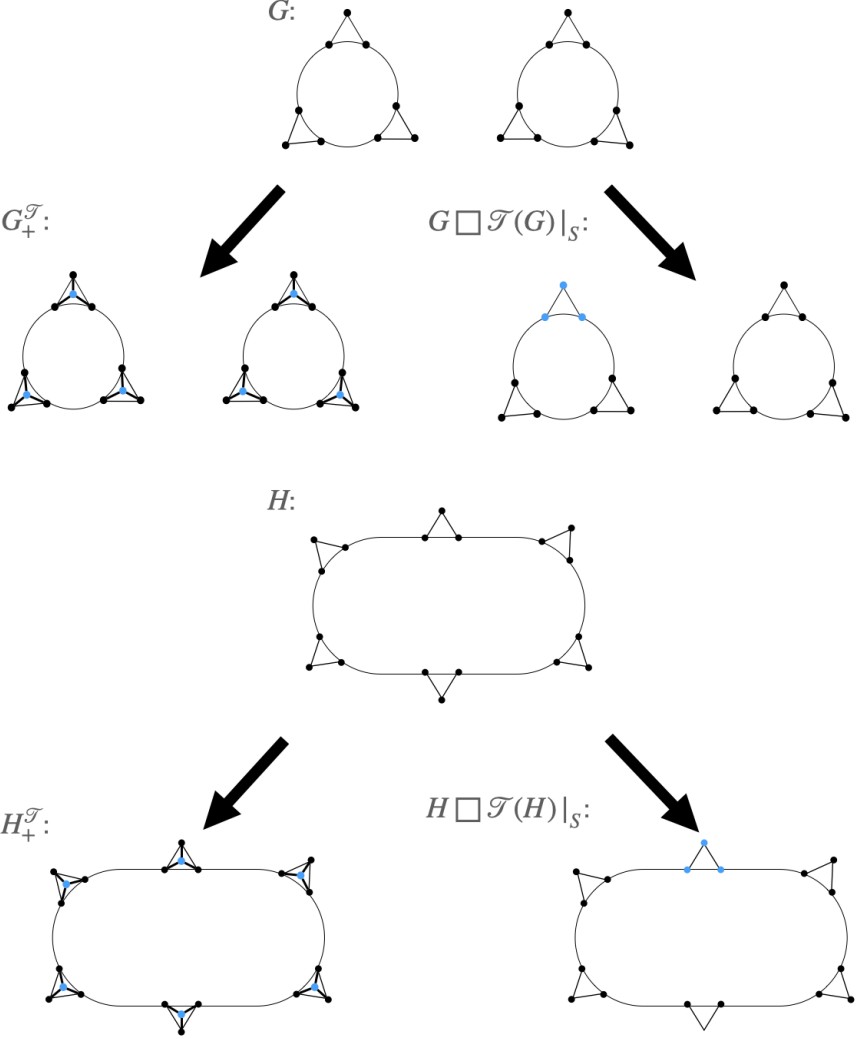

Figure 4: Rows 1 and 3 depict two 1-WL indistinguishable graphs> Rows 2 and 4 depict the sum graph of each of these graphs, as well as one subgraph of their product graphs induced by all node, super-node tuples whose super-node is fixed.

We conclude this section with the following proposition, showing there exists coarsening functions which, when combined with CS-GNN, results in an architecture that is strictly more expressive then node-based subgraph GNNs.

**Proposition D.4** (CS-GNN can be strictly more expressive then node-based subgraph GNNs). *Let $\mathcal{T}$ be the coarsening function defined by:*

$$\mathcal{T}(G) = \{\{v\} \mid v \in V\} \cup E \quad G = (V, E). \tag{63}$$

*The following holds:*

1. *Let $G_1, G_2$ be a pair of graphs such that there exists a node-based subgraph GNN model $M$ where $M(G_1) \neq M(G_2)$. There exists a CS-GNNmodel $M'$ which uses $\mathcal{T}$ such that $M'(G_1) \neq M'(G_2)$.*

2. *There exists a pair of graphs $G_1, G_2$ such that for any subgraph GNN model $M$ it holds that $M(G_1) = M(G_2)$, but there exists a CS-GNNmodel $M'$ which uses $\mathcal{T}$ such that $M'(G_1) \neq M'(G_2)$.*

This proposition is proved in Appendix G.

# E   Linear Invariant (Equivariant) Layer – Extended Section

We introduce some key notation. In the matrix $\mathcal{X}$, the $i$-th row corresponds to the $i$-th subset $S$ arranged in the lexicographic order of all subsets of $[n]$, namely, $[\{0\}, \{0, 1\}, \{0, 2\}, \ldots, \{0, 1, 2, \ldots, n\}]$. Each $i$-th position in this sequence aligns with the $i$-th row index in $\mathcal{X}$. It follows, that the standard basis for such matrices in $\mathbb{R}^{2^n \times n}$ is expressed as $\mathbf{e}^{(S)} \cdot \mathbf{e}^{(i)^T}$, where $\mathbf{e}^{(S)}$ is a 1-hot vector, with the value 1 positioned according to $S$ in the lexicographic order. For a matrix $X \in \mathbb{R}^{a \times b}$, the operation of vectorization, denoted by $\text{vec}(X)$, transforms $X$ into a single column vector in $\mathbb{R}^{ab \times 1}$ by sequentially stacking its columns; in the context of $\mathcal{X}$, the basis vectors of those vectors are $\mathbf{e}^{(i)} \otimes \mathbf{e}^{(S)}$. The inverse process, reshaping a vectorized matrix back to its original format, is denoted as $[\text{vec}(X)] = X$. We also denote an arbitrary permutation by $\sigma \in S_n$. The actions of permutations on vectors, whether indexed by sets or individual indices, are represented by $\mathbf{P}_{\mathcal{S}} \in \text{GL}(2^n)$ and $\mathbf{P}_{\mathcal{I}} \in \text{GL}(n)$, respectively. This framework acknowledges $S_n$ as a subgroup of the larger permutation group $S_{2^n}$, which permutes all $2^n$ positions in a given vector $\mathbf{v}_S \in \mathbb{R}^{2^n}$.

Let $\mathbf{L} \in \mathbb{R}^{1 \times 2^n \cdot n}$ be the matrix representation of a general linear operator $\mathcal{L} : \mathbb{R}^{2^n \times n} \to \mathbb{R}$ in the standard basis. The operator $\mathcal{L}$ is order-invariant iff

$$\mathbf{L} \, \text{vec}(\mathbf{P}_{\mathcal{S}}^T \mathcal{X} \mathbf{P}_{\mathcal{I}}) = \mathbf{L} \, \text{vec}(\mathcal{X}). \tag{64}$$

Similarly, let $\mathbf{L} \in \mathbb{R}^{2^n \cdot n \times 2^n \cdot n}$ denote the matrix for $\mathcal{L} : \mathbb{R}^{2^n \times n} \to \mathbb{R}^{2^n \times n}$. The operator $\mathcal{L}$ is order-equivariant if and only if

$$[\mathbf{L} \, \text{vec}(\mathbf{P}_{\mathcal{S}}^T \mathcal{X} \mathbf{P}_{\mathcal{I}})] = \mathbf{P}_{\mathcal{S}}^T [\mathbf{L} \, \text{vec}(\mathcal{X})] \mathbf{P}_{\mathcal{I}}. \tag{65}$$

Using properties of the Kronecker product (see Appendices E.1 and E.2 for details), we derive the following conditions for invariant and equivariant linear layers:

$$\text{Invariant } \mathbf{L}: \quad \mathbf{P}_{\mathcal{I}} \otimes \mathbf{P}_{\mathcal{S}} \, \text{vec}(\mathbf{L}) = \text{vec}(\mathbf{L}), \tag{66}$$

$$\text{Equivariant } \mathbf{L}: \quad \mathbf{P}_{\mathcal{I}} \otimes \mathbf{P}_{\mathcal{S}} \otimes \mathbf{P}_{\mathcal{I}} \otimes \mathbf{P}_{\mathcal{S}} \, \text{vec}(\mathbf{L}) = \text{vec}(\mathbf{L}). \tag{67}$$

**Solving Equations (66) and (67)**. Let $\sigma \in S_n$ denote a permutation corresponding to the permutation matrix $\mathbf{P}$. Let $\mathbf{P} \star \mathbf{L}$ denote the tensor that results from expressing $\mathbf{L}$ after renumbering the nodes in $V^{\mathcal{T}}, V$ according to the permutation $\sigma$. Explicitly, for $\mathbf{L} \in \mathbb{R}^{2^n \times n}$, the $(\sigma(S), \sigma(i))$-entry of $\mathbf{P} \star \mathbf{L}$ equals to the $(S, i)$-entry of $\mathbf{L}$. The matrix that corresponds to the operator $\mathbf{P} \star$ in the standard basis, $\mathbf{e}^{(i)} \otimes \mathbf{e}^{(S)}$ is the kronecker product $\mathbf{P}_{\mathcal{I}} \otimes \mathbf{P}_{\mathcal{S}}$. Since $\text{vec}(\mathbf{L})$ is exactly the coordinate vector of the tensor $\mathbf{L}$ in the standard basis we have,

$$\text{vec}(\mathbf{P} \star \mathbf{L}) = \mathbf{P}_{\mathcal{I}} \otimes \mathbf{P}_{\mathcal{S}} \, \text{vec}(\mathbf{L}), \tag{68}$$

following the same logic, the following holds for the equivariant case, where $\mathbf{L} \in \mathbb{R}^{2^n \cdot n \times 2^n \cdot n}$,

$$\text{vec}(\mathbf{P} \star \mathbf{L}) = \mathbf{P}_{\mathcal{I}} \otimes \mathbf{P}_{\mathcal{S}} \otimes \mathbf{P}_{\mathcal{I}} \otimes \mathbf{P}_{\mathcal{S}} \, \text{vec}(\mathbf{L}). \tag{69}$$

Given Equations (66) and (68) and Equations (67) and (69), it holds that we should focus on solving,
$$\mathbf{P} \star \mathbf{L} = \mathbf{L}, \quad \forall \mathbf{P} \text{ permutation matrices,} \tag{70}$$
for both cases where $\mathbf{L} \in \mathbb{R}^{2^n \times n}$ and $\mathbf{L} \in \mathbb{R}^{2^n \times n \times 2^n \times n}$, corresponding to the bias term, and linear term.

**Bias.** To this end, let us define an equivalence relation in the index space of a tensor in $\mathbb{R}^{2^n \times n}$. Given a pair $(S, i) \in \mathcal{P}([n]) \times [n]$, we define $\gamma^{k^+}$ to correspond to all pairs $(S, i)$ such that $|S| = k$ and $i \notin S$. Similarly, $\gamma^{k^-}$ corresponds to all pairs $(S, i)$ such that $|S| = k$ and $i \in S$. We denote this equivalence relation as follows:
$$(\mathcal{P}([n]) \times [n])/_\sim \triangleq \{\gamma^{k^*} : k = 1, \ldots, n; * \in \{+, -\}\}. \tag{71}$$

For each set-equivalence class $\gamma \in (\mathcal{P}([n]) \times [n])_\sim$, we define a basis tensor, $\mathbf{B}^\gamma \in \mathbb{R}^{2^n \times n}$ by setting:
$$\mathbf{B}^\gamma_{S,i} = \begin{cases} 1, & \text{if } (S, i) \in \gamma; \\ 0, & \text{otherwise.} \end{cases} \tag{72}$$

Following similar reasoning, consider elements $(S_1, i_1, S_2, i_2) \in (\mathcal{P}([n]) \times [n] \times \mathcal{P}([n]) \times [n])$. We define a partition according to six conditions: the relationship between $i_1$ and $i_2$, denoted as $i_1 \leftrightarrow i_2$, which is determines by the condition: $i_1 = i_2$ or $i_1 \neq i_2$; the cardinalities of $S_1$ and $S_2$, denoted as $k_1$ and $k_2$, respectively; the size of the intersection $S_1 \cap S_2$, denoted as $k^\cap$; the membership of $i_l$ in $S_l$ for $l \in \{1, 2\}$, denoted as $\delta_{\text{same}} \in \{1, 2, 3, 4\}$; and the membership of $i_{l_1}$ in $S_{l_2}$ for distinct $l_1, l_2 \in \{1, 2\}$, denoted as $\delta_{\text{diff}} \in \{1, 2, 3, 4\}$. The equivalence relation thus defined can be represented as:
$$(\mathcal{P}([n]) \times [n] \times \mathcal{P}([n]) \times [n])/_\sim \triangleq \{\Gamma^{\leftrightarrow;k_1;k_2;k^\cap;\delta_{\text{same}};\delta_{\text{diff}}}\}. \tag{73}$$

For each set-equivalence class $\Gamma \in (\mathcal{P}([n]) \times [n] \times \mathcal{P}([n]) \times [n])/_\sim$, we define a basis tensor, $\mathbf{B}^\Gamma \in \mathbb{R}^{2^n \times n \times 2^n \times n}$ by setting:
$$\mathbf{B}^\Gamma_{S_1,i_1;S_2,i_2} = \begin{cases} 1, & \text{if } (S_1, i_1, S_2, i_2) \in \Gamma; \\ 0, & \text{otherwise.} \end{cases} \tag{74}$$

The following two proposition summarizes the results in this section,

**Lemma E.1** ($\gamma$ ($\Gamma$) are orbits). *The sets $\{\gamma^{k^*} : k = 1, \ldots, n; * \in \{+, -\}\}$ and $\{\Gamma^{\leftrightarrow;k_1;k_2;k^\cap;\delta_{\text{same}};\delta_{\text{diff}}}\}$ are the orbits of $S_n$ on the index space $(\mathcal{P}([n]) \times [n])$ and $(\mathcal{P}([n]) \times [n] \times (\mathcal{P}([n]) \times [n])$, respectively.*

**Proposition E.1** (Basis of Invariant (Equivariant) Layer). *The tensors $\mathbf{B}^\gamma$ ($\mathbf{B}^\Gamma$) in Equation* (72) *(Equation* (74)*) form an orthogonal basis (in the standard inner product) to the solution of Equation* (66) *(Equation* (67)*).*

The proofs are given in Appendix G.

### E.1 Full Derivation of Equation (66).

Our goal is to transition from the equation,
$$\mathbf{L} \operatorname{vec}(\mathbf{P}_\mathcal{S}^T \mathcal{X} \mathbf{P}_\mathcal{I}) = \mathbf{L} \operatorname{vec}(\mathcal{X}) \tag{64}$$
to the form,
$$\mathbf{P}_\mathcal{I} \otimes \mathbf{P}_\mathcal{S} \operatorname{vec}(\mathbf{L}) = \operatorname{vec}(\mathbf{L}) \tag{66}$$
We introduce the following property of the Kronecker product,
$$\operatorname{vec}(\mathbf{ABC}) = (\mathbf{C}^T \otimes \mathbf{A})\operatorname{vec}(\mathbf{B}). \tag{75}$$
Using Equation (75) on the left side of Equation (64), we obtain
$$\mathbf{L}\mathbf{P}_\mathcal{I}^T \otimes \mathbf{P}_\mathcal{S}^T \operatorname{vec}(\mathcal{X}) = \mathbf{L} \operatorname{vec}(\mathcal{X}), \tag{76}$$
since this should be true for any $\mathcal{X} \in \mathbb{R}^{2^n \times n}$, we derive
$$\mathbf{L}\mathbf{P}_\mathcal{I}^T \otimes \mathbf{P}_\mathcal{S}^T = \mathbf{L}. \tag{77}$$
Applying the transpose operation on both sides, and noting that $(\mathbf{P}_\mathcal{I}^T \otimes \mathbf{P}_\mathcal{S}^T)^T = \mathbf{P}_\mathcal{I} \otimes \mathbf{P}_\mathcal{S}$, we obtain
$$\mathbf{P}_\mathcal{I} \otimes \mathbf{P}_\mathcal{S}\mathbf{L}^T = \mathbf{L}^T. \tag{78}$$
Recalling that $\mathbf{L} \in \mathbb{R}^{1 \times 2^n \cdot n}$, and thus $\mathbf{L}^T \in \mathbb{R}^{2^n \cdot n \times 1}$, we find that $\mathbf{L}^T = \operatorname{vec}(\mathbf{L})$. Substituting this back into the previous equation we achieve Equation (66).

## E.2 Full Derivation of Equation (67).

Our goal is to transition from the equation,

$$[\mathbf{L} \operatorname{vec}(\mathbf{P}_{\mathcal{S}}^T \mathcal{X} \mathbf{P}_{\mathcal{I}})] = \mathbf{P}_{\mathcal{S}}^T [\mathbf{L} \operatorname{vec}(\mathcal{X})] \mathbf{P}_{\mathcal{I}} \tag{65}$$

to the form,

$$\mathbf{P}_{\mathcal{I}} \otimes \mathbf{P}_{\mathcal{S}} \otimes \mathbf{P}_{\mathcal{I}} \otimes \mathbf{P}_{\mathcal{S}} \operatorname{vec}(\mathbf{L}) = \operatorname{vec}(\mathbf{L}). \tag{67}$$

Applying the property in Equation (75), after the reverse operation of the vectorization, namely,

$$[\operatorname{vec}(\mathbf{ABC})] = [(\mathbf{C}^T \otimes \mathbf{A}) \operatorname{vec}(\mathbf{B})] \tag{79}$$

on the right hand side of Equation (65), for

$$\mathbf{A} \triangleq \mathbf{P}_{\mathcal{S}}^T; \tag{80}$$
$$\mathbf{B} \triangleq [\mathbf{L} \operatorname{vec}(\mathcal{X})]; \tag{81}$$
$$\mathbf{C} \triangleq \mathbf{P}_{\mathcal{I}}, \tag{82}$$

we obtain,

$$[\mathbf{L} \operatorname{vec}(\mathbf{P}_{\mathcal{S}}^T \mathcal{X} \mathbf{P}_{\mathcal{I}})] = [\mathbf{P}_{\mathcal{I}}^T \otimes \mathbf{P}_{\mathcal{S}}^T \mathbf{L} \operatorname{vec}(\mathcal{X})]. \tag{83}$$

Thus, by omitting the revere-vectorization operation,

$$\mathbf{L} \operatorname{vec}(\mathbf{P}_{\mathcal{S}}^T \mathcal{X} \mathbf{P}_{\mathcal{I}}) = \mathbf{P}_{\mathcal{I}}^T \otimes \mathbf{P}_{\mathcal{S}}^T \mathbf{L} \operatorname{vec}(\mathcal{X}). \tag{84}$$

Noting that $(\mathbf{P}_{\mathcal{I}}^T \otimes \mathbf{P}_{\mathcal{S}}^T)^{-1} = \mathbf{P}_{\mathcal{I}} \otimes \mathbf{P}_{\mathcal{S}}$, and multiplying by this inverse both sides (from the left), we obtain,

$$\mathbf{P}_{\mathcal{I}} \otimes \mathbf{P}_{\mathcal{S}} \mathbf{L} \operatorname{vec}(\mathbf{P}_{\mathcal{S}}^T \mathcal{X} \mathbf{P}_{\mathcal{I}}) = \mathbf{L} \operatorname{vec}(\mathcal{X}). \tag{85}$$

Applying, again, the property in Equation (75), we obtain,

$$\mathbf{P}_{\mathcal{I}} \otimes \mathbf{P}_{\mathcal{S}} \mathbf{L} \mathbf{P}_{\mathcal{I}}^T \otimes \mathbf{P}_{\mathcal{S}}^T \operatorname{vec}(\mathcal{X}) = \mathbf{L} \operatorname{vec}(\mathcal{X}). \tag{86}$$

Since this should be true for any $\mathcal{X} \in \mathbb{R}^{2^n \times n}$, we derive,

$$\mathbf{P}_{\mathcal{I}} \otimes \mathbf{P}_{\mathcal{S}} \mathbf{L} \mathbf{P}_{\mathcal{I}}^T \otimes \mathbf{P}_{\mathcal{S}}^T = \operatorname{vec}(\mathbf{L}). \tag{87}$$

Again, applying Equation (75) on the left side, where,

$$\mathbf{A} \triangleq \mathbf{P}_{\mathcal{I}} \otimes \mathbf{P}_{\mathcal{S}}; \tag{88}$$
$$\mathbf{B} \triangleq \mathbf{L}; \tag{89}$$
$$\mathbf{C} \triangleq \mathbf{P}_{\mathcal{I}}^T \otimes \mathbf{P}_{\mathcal{S}}^T, \tag{90}$$

we get the following equality,

$$\mathbf{P}_{\mathcal{I}} \otimes \mathbf{P}_{\mathcal{S}} \mathbf{L} \mathbf{P}_{\mathcal{I}}^T \otimes \mathbf{P}_{\mathcal{S}}^T = \mathbf{P}_{\mathcal{I}} \otimes \mathbf{P}_{\mathcal{S}} \otimes \mathbf{P}_{\mathcal{I}} \otimes \mathbf{P}_{\mathcal{S}} \operatorname{vec}(\mathbf{L}). \tag{91}$$

By substituting this to the left side of Equation (87) we obtain Equation (67).

## E.3 Comparative Parameter Reduction in Linear Equivariant Layers

To demonstrate the effectiveness of our parameter-sharing scheme, which results from considering unordered tuples rather than ordered tuples, we present the following comparison. 3-IGNs [22] are structurally similar to our approach, with the main difference being that they consider indices as ordered tuples, while we consider them as sets. Both approaches use a total of six indices, as shown in the visualized block in Figure 3, making 3-IGNs a natural comparator. By leveraging our scheme, we reduce the number of parameters from 203 (the number of parameters in 3-IGNs) to just 35!

Table 5: Overview of the graph learning datasets.

| Dataset | # Graphs | Avg. # nodes | Avg. # edges | Directed | Prediction task | Metric |
|---|---|---|---|---|---|---|
| ZINC-12K [31] | 12,000 | 23.2 | 24.9 | No | Regression | Mean Abs. Error |
| ZINC-FULL [31] | 249,456 | 23.2 | 49.8 | No | Regression | Mean Abs. Error |
| OGBG-MOLHIV [16] | 41,127 | 25.5 | 27.5 | No | Binary Classification | AUROC |
| OGBG-MOLBACE [16] | 1513 | 34.1 | 36.9 | No | Binary Classification | AUROC |
| OGBG-MOLESOL [16] | 1,128 | 13.3 | 13.7 | No | Regression | Root Mean Squ. Error |
| PEPTIDES-FUNC [9] | 15,535 | 150.9 | 307.3 | No | 10-task Classification | Avg. Precision |
| PEPTIDES-STRUCT [9] | 15,535 | 150.9 | 307.3 | No | 11-task Regression | Mean Abs. Error |

# F  Extended Experimental Section

## F.1  Dataset Description

In this section we overview the eight different datasets considered; this is summarized in Table 5.

**ZINC-12K and ZINC-FULL Datasets [31, 14, 10].** The ZINC-12K dataset includes 12,000 molecular graphs sourced from the ZINC database, a compilation of commercially available chemical compounds. These molecular graphs vary in size, ranging from 9 to 37 nodes, where each node represents a heavy atom, covering 28 different atom types. Edges represent chemical bonds and there are three types of bonds. The main goal when using this dataset is to perform regression analysis on the constrained solubility (logP) of the molecules. The dataset is divided into training, validation, and test sets with 10,000, 1,000, and 1,000 molecular graphs respectively. The full version, ZINC-FULL, comprises approximately 250,000 molecular graphs, ranging from 9 to 37 nodes and 16 to 84 edges per graph. These graphs also represent heavy atoms, with 28 distinct atom types, and the edges indicate bonds between these atoms, with four types of bonds present.

**OGBG-MOLHIV, OGBG-MOLBACE, OGBG-MOLESOL Datasets [16].** These datasets are used for molecular property prediction and have been adopted by the Open Graph Benchmark (OGB, MIT License) from MoleculeNet. They use a standardized featurization for nodes (atoms) and edges (bonds), capturing various chemophysical properties.

**PEPTIDES-FUNC and PEPTIDES-STRUCT Datasets [9].** The PEPTIDES-FUNC and PEPTIDES-STRUCT datasets consist of atomic graphs representing peptides released with the Long Range Graph Benchmark (LRGB, MIT License). In PEPTIDES-FUNC, the task is to perform multi-label graph classification into ten non-exclusive peptide functional classes. Conversely, PEPTIDES-STRUCT is focused on graph regression to predict eleven three-dimensional structural properties of the peptides.

We note that for all datasets, we used the random splits provided by the public benchmarks.

## F.2  Experimental Details

**Implementation Details.** Our implementation of Equation (2) is given by:

$$\mathcal{X}^{(l+1)} = \text{MLP}\left(\sum_{i=1}^{3} \text{MPNN}^{(l+1,i)}\left(\mathcal{X}, \mathcal{A}_i\right)\right), \tag{92}$$

where $\mathcal{A}_1 = \mathcal{A}_G$, $\mathcal{A}_2 = \mathcal{A}_{\mathcal{T}(G)}$, and $\mathcal{A}_3 = \mathcal{A}_{\text{Equiv}}$.

For all considered datasets, namely, ZINC-12K, ZINC-FULL, OGBG-MOLHIV, OGBG-MOLBACE, and OGBG-MOLESOL, except for the PEPTIDES-FUNC and PEPTIDES-STRUC datasets, we use a GINE [15] base encoder. Given an adjacency matrix $\mathcal{A}$, and defining $e_{(S',v'),(S,v)}$ to denote the edge features from node $(S',v')$ to node $(S,v)$, it takes the following form:

$$\mathcal{X}(S,v) = \text{MLP}\left((1+\epsilon) \cdot \mathcal{X}(S,v) + \sum_{(S',v')\sim_{\mathcal{A}}(S,v)} \text{ReLU}\left(\mathcal{X}(S',v') + e_{(S',v'),(S,v)}\right)\right). \tag{93}$$

We note that for the symmetry-based updates, we switch the ReLU to an MLP[12] to align with the theoretical analyses[13] (Appendix B), stating that we can implement the equivariant update developed

---

[12]With the exception of the OGB datasets, to avoid overfitting.

[13]The theoretical analysis assumes the usage of an MLP for all three considered updates.

in Section 4.2. A more thorough discussion regarding the implementation of the symmetry-based updates is given in Appendix F.4.

When experimenting with the PEPTIDES-FUNC and PEPTIDES-STRUC datasets, we employ GAT [32] as our underlying MPNN to ensure a fair comparison with the random baseline—the random variant of `Subgraphormer + PE` [3]. To clarify, we consider the random variant of `Subgraphormer + PE` as a natural random baseline since it incorporates the information in the eigenvectors of the Laplacian (which we also do via the coarsening function). To maintain a fair comparison, we use a single vote for this random baseline, and maintained the same hyperparameters.

Our experiments were conducted using the PyTorch [28] and PyTorch Geometric [11] frameworks (resp. BSD and MIT Licenses), using a single NVIDIA L40 GPU, and for every considered experiment, we show the mean $\pm$ std. of 3 runs with different random seeds. Hyperparameter tuning was performed utilizing the Weight and Biases framework [6] – see Appendix F.3. All our MLPs feature a single hidden layer equipped with a ReLU non-linearity function. For the encoding of atom numbers and bonds, we utilized learnable embeddings indexed by their respective numbers.

In the case of the OGBG-MOLHIV, OGBG-MOLESOL, OGBG-MOLBACE datasets, we follow Frasca et al. [12], therefore adding a residual connection between different layers. Additionally, for those datasets (except OGBG-MOLHIV), we used linear layers instead of MLPs inside the GIN layers. Moreover, for these four datasets, and for the PEPTIDES datasets, the following pooling mechanism was employed

$$\rho(\mathcal{X}) = \texttt{MLP}\left(\sum_S \left(\frac{1}{n}\sum_{v=1}^{n}\mathcal{X}(s,v)\right)\right). \tag{94}$$

For the PEPTIDES datasets, we also used a residual connection between layers.

## F.3 HyperParameters

In this section, we detail the hyperparameter search conducted for our experiments. Besides standard hyperparameters such as learning rate and dropout, our specific hyperparameters are:

1. **Laplacian Dimension:** This refers to the number of columns used in the matrix $U$, where $L = U^T\lambda U$, for the spectral clustering in the coarsening function.
2. **SPD Dimension:** This represents the number of indices used in the node marking equation. To clarify, since $|S|$ might be large, we opt for using the first $k$ indices that satisfy $i \in S$, sorted according to the SPD distance.

**SPD Dimension.** For the Laplacian dimension, we chose a fixed value of 10 for all bag sizes for both ZINC-12K and ZINC-FULL datasets. For OGBG-MOLHIV, we used a fixed value of 1, since the value 10 did not perform well. For the PEPTIDES datasets, we also used the value 1. For the OGBG-MOLESOL and OGBG-MOLBACE datasets, we searched over the two values $\{1, 2\}$.

**Laplacian Dimension.** For the Laplacian dimension, we searched over the values $\{1, 2\}$ for all datasets.

**Standard Hyperparameters.** For ZINC-12K, we used a weight decay of 0.0003 for all bag sizes, except for the full bag size, for which we used 0.0001.

All of the hyperparameter search configurations are presented in Table 6, and the selected hyperparameters are presented in Table 7.

Table 6: Hyperparameters search for CS-GNN.

| Dataset | Bag size | Num. layers | Learning rate | Embedding size | Epochs | Batch size | Dropout | Laplacian dimension | SPD dimension |
|---|---|---|---|---|---|---|---|---|---|
| ZINC-12K | $T = 2$ | 6 | 0.0005 | 96 | 400 | 128 | 0 | $\{1, 2\}$ | 10 |
| ZINC-12K | $T \in \{3, 4, 5, 8, 18\}$ | 6 | 0.0007 | 96 | 400 | 128 | 0 | $\{1, 2\}$ | 10 |
| ZINC-12K | $T = $ "full" | 6 | 0.0007 | 96 | 500 | 128 | 0 | $\{1, 2\}$ | 10 |
| ZINC-FULL | $T = 4$ | 6 | 0.0007 | 96 | 400 | 128 | 0 | $\{1, 2\}$ | 10 |
| ZINC-FULL | $T = $ "full" | 6 | $\{0.001, 0.0005\}$ | 96 | 500 | 128 | 0 | $\{1, 2\}$ | 10 |
| OGBG-MOLHIV | $T \in \{2, 5, \text{"full"}\}$ | 2 | 0.01 | 60 | 100 | 32 | 0.5 | $\{1, 2\}$ | 1 |
| OGBG-MOLESOL | $T \in \{2, 5, \text{"full"}\}$ | 3 | 0.001 | 60 | 100 | 32 | 0.3 | $\{1, 2\}$ | $\{1, 2\}$ |
| OGBG-MOLBACE | $T \in \{2, 5, \text{"full"}\}$ | $\{2, 3\}$ | 0.01 | 60 | 100 | 32 | 0.3 | $\{1, 2\}$ | $\{1, 2\}$ |
| PEPTIDES-FUNC | $T = 30$ | 5 | $\{0.01, 0.005\}$ | 96 | 200 | 128 | 0 | $\{1, 2\}$ | 1 |
| PEPTIDES-STRUC | $T = 30$ | 4 | $\{0.01, 0.005\}$ | 96 | 200 | 128 | 0 | $\{1, 2\}$ | 1 |

Table 7: Chosen Hyperparameters for CS-GNN.

| Dataset | Bag size | Num. layers | Learning rate | Embedding size | Epochs | Batch size | Dropout | Laplacian dimension | SPD dimension |
|---|---|---|---|---|---|---|---|---|---|
| ZINC-12K | $T = 2$ | 6 | 0.0005 | 96 | 400 | 128 | 0 | 1 | 10 |
| ZINC-12K | $T = 3$ | 6 | 0.0007 | 96 | 400 | 128 | 0 | 2 | 10 |
| ZINC-12K | $T = 4$ | 6 | 0.0007 | 96 | 400 | 128 | 0 | 1 | 10 |
| ZINC-12K | $T = 5$ | 6 | 0.0007 | 96 | 400 | 128 | 0 | 1 | 10 |
| ZINC-12K | $T = 8$ | 6 | 0.0007 | 96 | 400 | 128 | 0 | 1 | 10 |
| ZINC-12K | $T = 18$ | 6 | 0.0007 | 96 | 400 | 128 | 0 | 1 | 10 |
| ZINC-12K | $T = $ "full" | 6 | 0.0007 | 96 | 500 | 128 | 0 | N/A | 10 |
| ZINC-FULL | $T = 4$ | 6 | 0.0007 | 96 | 400 | 128 | 0 | 1 | 10 |
| ZINC-FULL | $T = $ "full" | 6 | 0.0005 | 96 | 500 | 128 | 0 | N/A | N/A |
| OGBG-MOLHIV | $T = 2$} | 2 | 0.01 | 60 | 100 | 32 | 0.5 | 1 | 1 |
| OGBG-MOLHIV | $T = 5$ | 2 | 0.01 | 60 | 100 | 32 | 0.5 | 1 | 1 |
| OGBG-MOLHIV | $T = $ "full" | 2 | 0.01 | 60 | 100 | 32 | 0.5 | N/A | N/A |
| OGBG-MOLESOL | $T = 2$ | 3 | 0.001 | 60 | 100 | 32 | 0.3 | 1 | 2 |
| OGBG-MOLESOL | $T = 5$ | 3 | 0.001 | 60 | 100 | 32 | 0.3 | 1 | 2 |
| OGBG-MOLESOL | $T = $ "full" | 3 | 0.001 | 60 | 100 | 32 | 0.3 | N/A | N/A |
| OGBG-MOLBACE | $T = 2$ | 3 | 0.01 | 60 | 100 | 32 | 0.3 | 1 | 1 |
| OGBG-MOLBACE | $T = 5$ | 3 | 0.01 | 60 | 100 | 32 | 0.3 | 1 | 2 |
| OGBG-MOLBACE | $T = $ "full" | 3 | 0.01 | 60 | 100 | 32 | 0.3 | N/A | N/A |
| PEPTIDES-FUNC | $T = 30$ | 5 | 0.005 | 96 | 200 | 128 | 0 | 1 | 1 |
| PEPTIDES-STRUC | $T = 30$ | 4 | 0.01 | 96 | 200 | 128 | 0 | 1 | 1 |

**Optimizers and Schedulers.** For the ZINC-12K and ZINC-FULL datasets, we employ the Adam optimizer paired with a ReduceLROnPlateau scheduler,factor set to 0.5, patience at 40[14], and a minimum learning rate of 0. For the OGBG-MOLHIV dataset, we utilized the ASAM optimizer [21] without a scheduler. For both OGBG-MOLESOL and OGBG-MOLBACE, we employed a constant learning rate without any scheduler. Lastly, for the PEPTIDES-FUNC and PEPTIDES-STRUCT datasets, the AdamW optimizer was chosen in conjunction with a cosine annealing scheduler, incorporating 10 warmup epochs.

### F.4 Implementation of Linear Equivariant and Invariant layers – Extended Section

In this section, in a more formal discussion, we specify how to integrate those invariant and equivariant layers to our proposed architecture. We start by drawing an analogy between parameter sharing in linear layers and the operation of an MPNN on a fully connected graph with edge features in the following lemma,

**Lemma F.1** (Parameter Sharing as MPNN). *Let $B_1, \ldots B_k : \mathbb{R}^{n \times n}$ be orthogonal matrices with entries restricted to 0 or 1, and let $W_1, \ldots W_k \in \mathbb{R}^{d \times d'}$ denote a sequence of weight matrices. Define $B_+ = \sum_{i=1}^{k} B_i$ and choose $z_1, \ldots z_k \in \mathbb{R}^{d^*}$ to be a set of unique vectors representing an encoding of the index set. The function, which represents an update via parameter sharing:*

$$f(X) = \sum_{i=1}^{k} B_i X W_i, \tag{95}$$

*can be implemented by a stack of MPNN layers of the following form [13],*

$$m_u^l = \sum_{v \in N_{B_+}(u)} M^l(X_v^l, e_{u,v}),, \tag{96}$$

$$X_u^{l+1} = U^l(X_v^l, m_v^l), \tag{97}$$

*where $U^l, M^l$ are multilayer preceptrons (MLPs). The inputs to this MPNN are the adjacency matrix $B_+$, node feature vector $X$, and edge features – the feature of edge $(u, v)$ is given by:*

$$e_{u,v} = \sum_{i=1}^{k} z_i \cdot B_i(u, v). \tag{98}$$

*Here, $B_i(u, v)$ denotes the $(u, v)$ entry to matrix $B_i$.*

The proof is given in Appendix G.

---

[14]For ZINC-12K, $T \in \{2, $ "full"$\}$, we used a patience of 50.

Table 8: Comparison over the ZINC-FULL molecular dataset under $500k$ parameter budget. The best performing method is highlighted in **blue**, while the second best is highlighted in **red**.

| Model ↓ / Dataset → | ZINC-FULL (MAE ↓) |
|---|---|
| MAG-GNN [20] ($T = 4$) | **0.030**±0.002 |
| Ours ($T = 4$) | **0.027**±0.002 |
| GNN-SSWL [38] ($T =$ "full") | 0.026±0.001 |
| GNN-SSWL+ [38] ($T =$ "full") | 0.022±0.001 |
| Subgraphormer [3]($T =$ "full") | **0.020**±0.002 |
| Subgraphormer + PE [3] ($T =$ "full") | 0.023±0.001 |
| Ours ($T =$ "full") | **0.021**±0.001 |

Thus, our implementation for the global update is as follows,

$$\mathcal{X}(S, i) = \texttt{MLP}\left((1 + \epsilon) \cdot \mathcal{X}(S, i) + \sum_{(S', i') \sim \mathcal{A}_{\text{Equiv}}(S, i)} \texttt{MLP}\left(\mathcal{X}(S', i') + e_{(S', i'), (S, i)}\right)\right), \quad (99)$$

where $e_{(S', i'), (S, i)} = \sum_\Gamma z_\Gamma \cdot \mathbf{B}^\Gamma_{S, i; S', i'}$ and $z_\Gamma$ are orthogonal 1-hot vectors for different $\Gamma$'s. The connectivity $\mathcal{A}_{Equiv}$ is such that $\mathcal{A}_{Equiv}(S, v, S', v')$ contains the value one iff $v \in S, v = v'$. This corresponds to choosing only several $\Gamma$'s in the partition, and since each $\Gamma$ is invariant to the permutation, this choice still maintains equivariance.

### F.5 Additional Results

**ZINC-FULL.** Below, we present our results on the ZINC-FULL dataset for a bag size of $T = 4$ and the full-bag. For the bag size $T = 4$, we benchmark against MAG-GNN [20], which in their experiments used the best out of the bag sizes $T \in \{2, 3, 4\}$; however, they did not specify which one performed the best. The results are summarized in Table 8.

**ZINC-12K – additional results.** We present all the results from Figure 2, along with some additional ones, in Table 9.

**Runtime comparison.** We compare the training time and prediction performance on the ZINC-12K dataset. For all methods, we report the training and inference times on the entire training and test sets, respectively, using a batch size of 128. Our experiments were conducted using an NVIDIA L40 GPU, while for the baselines, we used the timing reported in [5], which utilized an RTX A6000 GPU. The runtime comparison is presented in Table 10.

### F.6 ZINC12K Product Graph Visualization

In this subsection, we visualize the product graph derived from the first graph in the ZINC12K dataset. Specifically, we present the right part of Figure 1, for the case of the real-world graphs in the ZINC12K dataset. We perform this visualization for different cluster sizes, $T \in \{2, 3, 4, 5, 8, 12\}$, which also define the bag size, hence the notation $T$. The nodes in the product graph, $\mathcal{T}(G)\square G$, are $(S, v)$, where $S$ is the coarsened graph node (again a tuple), and $v$ is the node index (of a node from the original graph). For better clarity, we color the nodes $(S, v)$ with $v \in S$ using different colors, while reserving the gray color exclusively for nodes $(S, v)$ where $v \notin S$. The product graphs are visualized in Figures 5 to 10 below.

Table 9: Test results on the ZINC-12K molecular dataset under $500k$ parameter budget. The top two results are reported as **First** and **Second**.

| Method | Bag size | ZINC (MAE $\downarrow$) |
|---|---|---|
| GCN [19] | $T = 1$ | $0.321 \pm 0.009$ |
| GIN [35] | $T = 1$ | $0.163 \pm 0.004$ |
| OSAN [29] | $T = 2$ | $0.177 \pm 0.016$ |
| Random [20] | $T = 2$ | $0.131 \pm 0.005$ |
| PL [5] | $T = 2$ | $0.120 \pm 0.003$ |
| Mag-GNN [20] | $T = 2$ | $\mathbf{0.106} \pm 0.014$ |
| Ours | $T = 2$ | $\mathbf{0.109} \pm 0.005$ |
| Random [20] | $T = 3$ | $0.124 \pm$ N/A |
| Mag-GNN [20] | $T = 3$ | $\mathbf{0.104} \pm$ N/A |
| Ours | $T = 3$ | $\mathbf{0.096} \pm 0.005$ |
| Random [20] | $T = 4$ | $0.125 \pm$ N/A |
| Mag-GNN [20] | $T = 4$ | $\mathbf{0.101} \pm$ N/A |
| Ours | $T = 4$ | $\mathbf{0.090} \pm 0.003$ |
| Random [5] | $T = 5$ | $0.113 \pm 0.006$ |
| PL [5] | $T = 5$ | $\mathbf{0.109} \pm 0.005$ |
| Ours | $T = 5$ | $\mathbf{0.095} \pm 0.003$ |
| Random [5] | $T = 8$ | $0.102 \pm 0.003$ |
| PL [5] | $T = 8$ | $\mathbf{0.097} \pm 0.005$ |
| Ours | $T = 8$ | $\mathbf{0.094} \pm 0.006$ |
| Ours | $T = 18$ | $\mathbf{0.082} \pm 0.003$ |
| NGNN [39] | Full | $0.111 \pm 0.003$ |
| DS-GNN [4] | Full | $0.116 \pm 0.009$ |
| DSS-GNN [4] | Full | $0.102 \pm 0.003$ |
| GNN-AK [40] | Full | $0.105 \pm 0.010$ |
| GNN-AK+ [40] | Full | $0.091 \pm 0.002$ |
| SUN [12] | Full | $0.083 \pm 0.003$ |
| OSAN [29] | Full | $0.154 \pm 0.008$ |
| GNN-SSWL+ [38] | Full | $0.070 \pm 0.005$ |
| Subgraphormer [3] | Full | $0.067 \pm 0.007$ |
| Subgraphormer+PE [3] | Full | $\mathbf{0.063} \pm 0.001$ |
| Ours | Full | $\mathbf{0.062} \pm 0.0007$ |

Table 10: Run time comparison over the ZINC-12K dataset. Time taken at train for one epoch and at inference on the test set. All values are in milliseconds.

| Method | Train time (for a single epoch; ms) | Test time (ms) | MAE $\downarrow$ |
|---|---|---|---|
| GIN [35] | $1370.10 \pm 10.97$ | $84.81 \pm 0.26$ | $0.163 \pm 0.004$ |
| OSAN [29] ($T = 2$) | $2964.46 \pm 30.36$ | $227.93 \pm 0.21$ | $0.177 \pm 0.016$ |
| PL [5] ($T = 2$) | $2489.25 \pm 9.42$ | $150.38 \pm 0.33$ | $0.120 \pm 0.003$ |
| Ours ($T = 2$) | $2764.60 \pm 234$ | $383.14 \pm 15.74$ | $0.109 \pm 0.005$ |

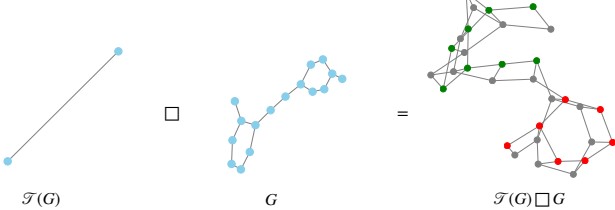

Figure 5: $T = 2$.

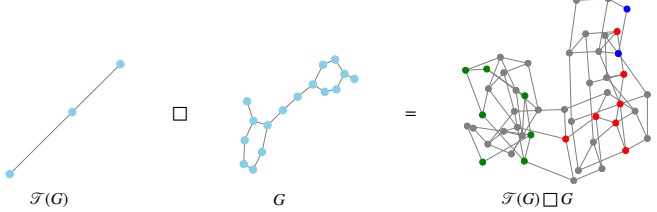

Figure 6: $T = 3$.

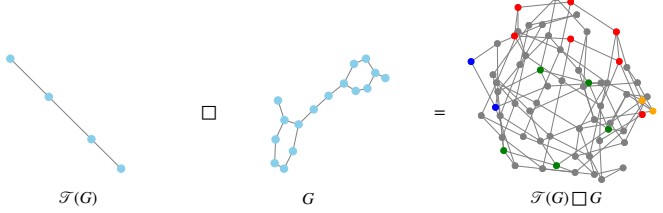

Figure 7: $T = 4$.

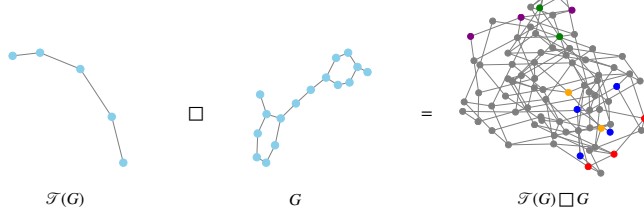

Figure 8: $T = 5$.

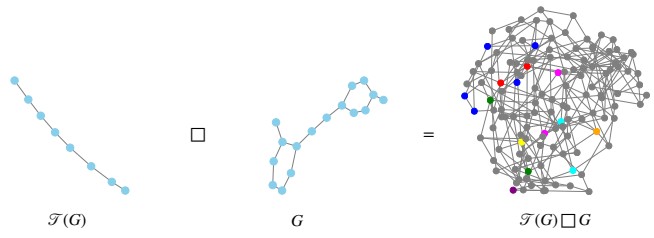

Figure 9: $T = 8$.

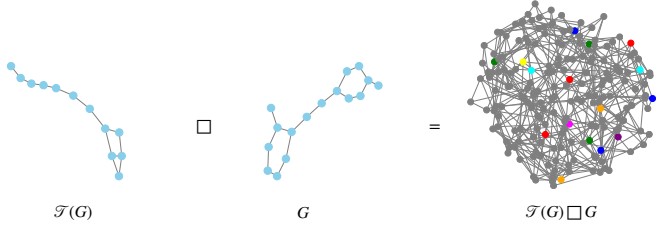

Figure 10: $T = 12$.

# G Proofs

## G.1 Proofs of Appendix B

We first state the memorization theorem, proven in [36] , which will be heavily used in a lot of the proofs in this section.

**Theorem G.1** (Memorization Theorem). *Consider a dataset $\{x_j, y_j\}_{j=1}^N \in \mathbb{R}^d \times \mathbb{R}^{d_y}$ , with each $x_j$ being distinct and every $y_j \in \{0,1\}^{d_y}$. There exists a 4-layer fully connected ReLU neural network $f_\theta : \mathbb{R}^d \to \mathbb{R}^{d_y}$ that perfectly maps each $x_j$ to its corresponding $y_j$, i.e., $f_\theta(x_j) = y_j$ for all j.*

We now restate and prove the propositions and lemmas of Appendix B.

**Lemma B.1** (Parameter Sharing as MPNN). *Let $B_1, \ldots B_k : \mathbb{R}^{n \times n}$ be orthogonal matrices with entries restricted to 0 or 1, and let $W_1, \ldots W_k \in \mathbb{R}^{d \times d'}$ denote a sequence of weight matrices. Define $B_+ = \sum_{i=1}^k B_i$ and choose $z_1, \ldots z_k \in \mathbb{R}^{d^*}$ to be a set of unique vectors representing an encoding of the index set. The function that represents an update via parameter sharing:*

$$f(X) = \sum_{i=1}^k B_i X W_i, \tag{18}$$

*can be implemented on any finite family of graphs $\mathcal{G}$, by a stack of MPNN layers of the following form [13],*

$$m_v^l = \sum_{u \in N_{B_+}(v)} M^l(X_u^l, e_{u,v}), \tag{19}$$

$$X_v^{l+1} = U^l(X_v^l, m_v^l), \tag{20}$$

*where $U^l$, $M^l$ are multilayer perceptrons (MLPs). The inputs to this MPNN are the adjacency matrix $B_+$, node feature vector $X$, and edge features – the feature of edge $(u,v)$ is given by:*

$$e_{u,v} = \sum_{i=1}^k z_i \cdot B_i(u,v). \tag{21}$$

*Here, $B_i(u,v)$ denotes the $(u,v)$ entry to matrix $B_i$.*

*Proof.* Since we are concerned only with input graphs $G$ from a finite family of graphs (where "finite" means that the maximal graph size is bounded and all possible node and edge feature values come from a finite set), we assume that for any $v \in [n]$, $i \in [k]$, both the input feature vectors $X_v \in \mathbb{R}^d$ and the encoding vectors $z_i \in \mathbb{R}^{d^*}$ are one-hot encoded. We aim to show that under these assumptions, any function $f(\cdot)$ of the form 95 can be realized through a single-layer update detailed in Equations 97 , 96, where $M$ is a 4 layer MLP , and $U$ is a single linear layer. The proof involves the following steps:

1. Compute $[B_1 X, \ldots, B_k X]$ using the message function $M$.

2. Compute $f(X)$ using the update function $U$.

**Step 1:** We notice that for every $i \in [k]$, $v \in [n]$ we have:

$$(B_i X)_v = \sum_{B_i(v,u)=1} X_u = \sum_{u \in N_{B_+}(v)} X_u \cdot \mathbf{1}_{z_i}(e_{u,v}). \tag{100}$$

Here $\mathbf{1}_{z_i}$ is the indicator function of the set $\{z_i\}$. We notice that since $X_u$ and $z_i$ are one-hot encoded, there is a finite set of possible values for the pair $(X_u, e_{u,v})$. In addition, the function:

$$enc(X_u, e_{u,v}) = [X_u \cdot \mathbf{1}_{z_1}(e_{u,v}), \ldots, X_u \cdot \mathbf{1}_{z_k}(e_{u,v})] \tag{101}$$

outputs vectors in the set $\{0,1\}^{d \times k}$. Thus, employing the memorization theorem G.1, we define a dataset $\{x_j, y_j\}_{j=1}^N$ by taking the $x_j$s to be all possible (distinct) values of $(X_u, e_{u,v})$ with each

corresponding $y_i$ being the output $enc(x_i)$. We note that there are finitely many such values as both $X_u$ and $e_{u,v}$ are one-hot encoded. The theorem now tells us that there exists a a 4-layer fully connected ReLU neural network $M$ such that:

$$M(X_u, e_{u,v}) = enc(X_u, e_{u,v}). \tag{102}$$

and so, equation 100 implies:

$$m_v = \sum_{u \in N_{B_+}(v)} M(X_u, e_{u,v}) = [(B_1 X)_v, \dots, (B_k X)_v]. \tag{103}$$

**Step 2:** Define $P_i : \mathbb{R}^{k \times d} \to \mathbb{R}^d$ as the projection operator, extracting coordinates $d \cdot i + 1$ through $d \cdot (i+1)$ from its input vector:

$$P_i(V) = V|_{d \cdot i + 1 : d \cdot (i+1)}. \tag{104}$$

We define the update function to be the following linear map:

$$U(X_v, m_v) = \sum_{i=1}^{k} P_i(m_v) W_i. \tag{105}$$

Combining equations 103 and 105 we get:

$$\tilde{X}_v = U(X_v, m_v) = \sum_{i=1}^{k} (B_i X)_v \cdot W_i = f(X)_v. \tag{106}$$

$\square$

**Proposition B.1** (Equivalence of General Layer and Implemented Layer). *Let $\mathcal{T}(\cdot)$ be a coarsening function, $\pi$ be a generalized node marking policy, and $\mathcal{G}$ be a finite family of graphs. Applying a stack of $t$ general layer updates as defined in Equation 15 to the node feature map $\mathcal{X}(S, v)$ induced by $\pi(G, \mathcal{T})$, can be effectively implemented by applying a stack of $t$ layer updates specified in Equations 29 and 30 to $\mathcal{X}(S, v)$. Additionally, the depths of all MLPs that appear in 29 and 30 can be bounded by 4.*

*Proof.* For convenience, let us first restate the general layer update:

$$
\begin{aligned}
\mathcal{X}^{t+1}(S, v) = f^t \Big( & \mathcal{X}^t(S, v), \\
& \text{agg}_1^t \{\!\{ (\mathcal{X}^t(S, v'), e_{v,v'}) \mid v' \sim_G v \}\!\}, \\
& \text{agg}_2^t \{\!\{ (\mathcal{X}^t(S', v), \tilde{e}_{S,S'}) \mid S' \sim_{G^\mathcal{T}} S \}\!\}, \\
& \text{agg}_3^t \{\!\{ (\mathcal{X}^t(S', v), z(S, v, S', v)) \mid S' \in V^\mathcal{T} \text{s.t. } v \in S' \}\!\}, \\
& \text{agg}_4^t \{\!\{ (\mathcal{X}^t(S, v'), z(S, v, S, v')) \mid v' \in V \text{s.t. } v' \in S \}\!\} \Big),
\end{aligned}
\tag{15}
$$

as well as the two step implemented layer update:

$$\mathcal{X}_i^t(S, v) = U_i^t \left( (1 + \epsilon_i^t) \cdot \mathcal{X}^t(S, v) + \sum_{(S', v') \sim_{A_i}(S, v)} M^t(\mathcal{X}^t(S', v') + e_i(S, v, S', v')) \right). \tag{29}$$

$$\mathcal{X}^{t+1}(S, v) = U_{\text{fin}}^t \left( \sum_{i=1}^{4} \mathcal{X}_i^t(S, v) \right). \tag{30}$$

We aim to demonstrate that any general layer, which updates the node feature map $\mathcal{X}^t(S, v)$ at layer $t$ to node feature map $\mathcal{X}^{t+1}(S, v)$ at layer $t+1$ as described in equation 15, can be effectively implemented using the layer update processes outlined in equations 29 and 30.

As we are concerned only with input graphs belonging to the finite graph family $\mathcal{G}$ (where "finite" indicates that the maximal graph size is bounded and all node and edge features have a finite set of possible values), we assume that the values of the node feature map $\mathcal{X}^t(S, v)$ and the edge feature vectors $e_i(S, v, S', v')$ are represented as one-hot vectors in $\mathbb{R}^k$. We also assume that the parameterized functions $f^t$ and $\mathrm{agg}_1^t, \ldots \mathrm{agg}_4^t$, which are applied in Equation 15 outputs one-hot vectors. Finally, we assume that there exists integers $d, d^*$, such that the node feature map values are supported on coordinates $1, \ldots d$, the edge feature vectors are supported on coordinates $d+1, \ldots d + d^*$, and coordinates $d + d^* + 1, \ldots k$ are used as extra memory space, with:

$$k > d \times d^* + d + d^*. \tag{107}$$

We note that the last assumption can be easily achieved using padding. The proof involves the following steps:

1. For $i = 1, \ldots, 4$, Use the term:

$$m_i^t = \sum_{(S', v') \sim A_i(S, v)} M^t(\mathcal{X}^t(S', v') + e_i(S, v, S', v')) \tag{108}$$

   to uniquely encode:

$$\{\!\!\{ (\mathcal{X}^t(S', v'), e_i(S, v, S', v')) \mid (S', v') \sim_{A_i} (S, v) \}\!\!\}. \tag{109}$$

2. Use the term:

$$\mathcal{X}_*^t = \sum_{i=1}^4 \mathcal{X}_i^t(S, v) \tag{110}$$

   to uniquely encode the input of $f^t$ as a whole.

3. Implement the parameterized function $f^t$.

**Step 1:** Since we assume that node feature map values and edge feature vectors are supported on orthogonal sub-spaces of $\mathbb{R}^k$, the term:

$$\mathcal{X}^t(S, v) + e_i(S, v, S', v') \tag{111}$$

uniquely encodes the value of the tuple:

$$(\mathcal{X}^t(S, v), e_i(S, v, S', v')). \tag{112}$$

Since $\mathcal{X}^t(S, v)$ is a one-hot encoded vector with $d$ possible values, while $e_i(S, v, S', v')$ is a one-hot encoded vector with $d^*$ possible values, their sum has $d \cdot d^*$ possible values. Thus there exists a function:

$$\mathrm{enc} : \mathbb{R}^k \to \mathbb{R}^k$$

which encodes each such possible value as a one-hot vector in $\mathbb{R}^k$ supported on the last $k - d - d^*$ coordinates (this is possible because of equation 107). Now, employing theorem G.1, we define the $x_j$s as all possible (distinct) values of 111, with each corresponding $y_j$ being the output $\mathrm{enc}(x_j)$. The theorem now tells us that there exists a 4-layer fully connected ReLU neural network capable of implementing the function $\mathrm{enc}(\cdot)$. We choose $M^t$ to be this network. Now since $m_i^t$, defined in equation 108 is a sum of one-hot encoded vectors, it effectively counts the number of each possible value in the set 109. This proves step 1.

**Step 2:** First, we note that:

$$
\begin{aligned}
&\{\!\!\{ (\mathcal{X}^t(S, v'), e_{v, v'}) \mid v' \sim_G v \}\!\!\} \\
&= \{\!\!\{ (\mathcal{X}^t(S', v'), e_G(S, v, S', v')) \mid (S, v) \sim_{A_G} (S', v') \}\!\!\}
\end{aligned}
\tag{113}
$$

$$
\begin{aligned}
&\{\!\!\{ (\mathcal{X}^t(S', v), e_{s', s}) \mid S' \sim_{\mathcal{T}(G)} S \}\!\!\} \\
&= \{\!\!\{ (\mathcal{X}^t(S', v'), e_{\mathcal{T}(G)}(S, v, S', v')) \mid (S, v) \sim_{A_{\mathcal{T}(G)}} (S', v') \}\!\!\}
\end{aligned}
\tag{114}
$$

$$
\begin{aligned}
&\{\!\!\{ (\mathcal{X}^t(S', v), z(S, v, S', v')) \mid v \in S' \}\!\!\} \\
&= \{\!\!\{ (\mathcal{X}^t(S', v'), e_{P_1}(S, v, S', v')) \mid (S, v) \sim_{A_{P_1}} (S', v') \}\!\!\}
\end{aligned}
\tag{115}
$$

$$\{\!\{(\mathcal{X}^t(S, v'), z(S, v, S', v')) \mid v' \in S\}\!\}$$
$$= \{\!\{(\mathcal{X}^t(S', v'), e_{P_2}(S, v, S', v')) \mid (S, v) \sim_{A_{P_2}} (S', v')\}\!\} \tag{116}$$

Now, since $m_i^t$ and $\mathcal{X}^t(S, v)$ are supported on orthogonal sub-spaces of $\mathbb{R}^k$, the sum $\mathcal{X}^t(S, v) + m_i^t$ uniquely encodes the value of:

$$\left(\mathcal{X}^t(S, v), \{\!\{(\mathcal{X}^t(s, v), e_i(S, v, S', v')) \mid (S, v) \sim_{A_i} (S,' v')\}\!\}\right). \tag{117}$$

Thus, we choose $\epsilon_1^t, \ldots, epsilon_4^t$ to be all zeroes. To compute the aggregation functions $\mathrm{agg}_1^t, \ldots, \mathrm{agg}_4^t$ using these unique encodings, and to avoid repetition of the value $\mathcal{X}^t(S, v)$, we define auxiliary functions $\tilde{\mathrm{agg}}_i^t : \mathbb{R}^k \to \mathbb{R}^{k_i}$ for $i = 1, \ldots, 4$ as follows:

$$\tilde{\mathrm{agg}}_1^t(\mathcal{X}^t(S, v) + m_1^t) = \left(\mathcal{X}^t(S, v), \mathrm{agg}_1^t\{\!\{(\mathcal{X}^t(S, v'), e_1(S, v, S', v')) \mid (S, v) \sim_{A_1} (S,' v')\}\!\}\right) \tag{118}$$

and for $i > 1$:

$$\tilde{\mathrm{agg}}_i^t(\mathcal{X}^t(S, v) + m_i^t) = \mathrm{agg}_i^l\{\!\{(\mathcal{X}^t(s, v), e_i(S, v, S', v')) \mid (S, v) \sim_{A_i} (S,' v')\}\!\}. \tag{119}$$

Here, since we avoided repeating the value of $\mathcal{X}^t(S, v)$ by only adding it to the output of $\tilde{\mathrm{agg}}_1^t(\cdot)$, the expression:

$$\left(\tilde{\mathrm{agg}}_1^t(\mathcal{X}^t(S, v) + m_1^t), \ldots, \tilde{\mathrm{agg}}_4^t(\mathcal{X}^t(S, v) + m_4^t)\right) \tag{120}$$

is exactly equal to the input of $f^t$. In addition, since the function $\mathrm{agg}_i^t$ outputs one-hot encoded vectors, and the vector $\mathcal{X}^t(S, v)$ is one-hot encoded, the output of $\tilde{\mathrm{agg}}_i^t$ is always within the set $\{0, 1\}^{k_i}$. Now for any input vector $X \in \mathbb{R}^k$ define:

$$V_1^t(X) = (\tilde{\mathrm{agg}}_1^t(X), \ 0_{k_2}, \ 0_{k_3}, \ 0_{k_4}). \tag{121}$$

$$V_2^t(X) = (0_{k_1}, \ \tilde{\mathrm{agg}}_2^t(X), \ 0_{k_3}, \ 0_{k_4}). \tag{122}$$

$$V_3^t(X) = (0_{k_1}, \ 0_{k_2}, \ \tilde{\mathrm{agg}}_3^t(X), \ 0_{k_4}). \tag{123}$$

$$V_4^t(X) = (0_{k_1}, \ 0_{k_2}, \ 0_{k_3}, \tilde{\mathrm{agg}}_4^t(X)). \tag{124}$$

We note that since the output of $\mathrm{agg}_i^t$ is always within the set $\{0, 1\}^{k_i}$, the outputs of $V_i^t$ is always within $\{0, 1\}^{k_1 + \cdots + k_4}$. Now for $i = 1, \ldots 4$, employing theorem G.1 we define a dataset $\{x_j, y_j\}_{j=1}^N$ by taking the $x_j$s as all possible (distinct) values of $\mathcal{X}^t(S, v) + m_i^t$, with each corresponding $y_j$ being the output $V_i^t(x_j)$. We note that there are finitely many such values as both $\mathcal{X}^t(S, v)$ and $m_i^t$ are one-hot encoded vectors. The theorem now tells us that there exists a a 4-layer fully connected ReLU neural network capable of implementing the function $V_i^t(\cdot)$. We choose $U_i^t$ to be this network. Equations 121 - 124 now give us:

$$\sum_{i=1}^4 \mathcal{X}_i^t(S, v) = \left(\tilde{\mathrm{agg}}_1^t(\mathcal{X}^t(S, v) + m_1^t), \ldots, \tilde{\mathrm{agg}}_4^t(\mathcal{X}^t(S, v) + m_4^t)\right). \tag{125}$$

which as stated before, is exactly the input to $f^t$. This proves step 2.

**Step 3:** We employ theorem G.1 for one final time, defining a dataset $\{x_j, y_j\}_{j=1}^N$ by taking the $x_j$s as all possible(distinct) values of:

$$\sum_{i=1}^4 \mathcal{X}_i^t(S, v)$$

(which we showed is a unique encoding to the input of $f^t(\cdot)$), with each corresponding $y_j$ being the output $f^t(x_j)$. We note that Given the finite nature of our graph set, there are finitely many such values. Recalling that $f^t(\cdot)$ outputs one-hot encoded vectors, The theorem now tells us that there exists a a 4-layer fully connected ReLU neural network capable of implementing the function $f^t(\cdot)$. We choose $U_{\mathrm{fin}}^t$ to be this network. This completes the proof. $\qquad\square$

## G.2 Proofs of Appendix C

**Proposition C.1** (Equal Expressivity of Node Marking Policies). *For any coarsening function $\mathcal{T}(\cdot)$ the following holds:*

$$CS\text{-}GNN(\mathcal{T}, \pi_S) = CS\text{-}GNN(\mathcal{T}, \pi_{SS}) = CS\text{-}GNN(\mathcal{T}, \pi_{MD}). \tag{36}$$

*Proof.* Let $\Pi = \{\pi_S, \pi_{SS}, \pi_{MD}\}$ be the set of all relevant node initialization policies, and assume for simplicity that our input graphs have no node features (the proof can be easily adjusted to account for the general case). For each $\pi \in \Pi$, let $\mathcal{X}^\pi(S, v)$ denote the node feature map induced by general node marking policy $\pi$, as per Definition C.1. We notice it is enough to prove for each $\pi_1, \pi_2 \in \Pi$ that $\mathcal{X}^{\pi_1}(S, v)$ can be implemented by updating $\mathcal{X}^{\pi_2}(S, v)$ using a stack of $T$ layers of type 54. Thus, we prove the following four cases:

- Node + Size Marking $\Rightarrow$ Simple Node Marking.

- Minimum Distance $\Rightarrow$ Simple Node Marking.

- Simple Node Marking $\Rightarrow$ Node + Size Marking.

- Simple Node Marking $\Rightarrow$ Minimum Distance.

**Node + Size Marking $\Rightarrow$ Simple Node Marking**:

In this case, we aim to update the node feature map:

$$\mathcal{X}^0(S, v) = \mathcal{X}^{\pi_{SS}}(S, v) = \begin{cases} (1, |S|) & v \in S \\ (0, |S|) & v \notin S. \end{cases} \tag{126}$$

We notice that:

$$\mathcal{X}^0(S, v) = \langle (1, 0), \ \mathcal{X}^{\pi_S}(S, v) \rangle, \tag{127}$$

where $\langle \cdot, \cdot \rangle$ denotes the standard inner product in $\mathbb{R}^2$. Using a CS-GNN update as per equation 15, with the update function:

$$f^1(\mathcal{X}^0(S, v), \cdot, \cdot, \cdot, \cdot) = \langle (1, 0), \mathcal{X}^0(S, v) \rangle, \tag{128}$$

where $f(a, \cdot, \cdot, \cdot, \cdot)$ indicates that the function $f$ depends solely on the parameter $a$, we obtain:

$$\mathcal{X}^1(S, v) = f^1(\mathcal{X}^0(S, v), \cdot, \cdot, \cdot, \cdot) = \mathcal{X}^{\pi_S}(S, v). \tag{129}$$

This implies that for any coarsening function $\mathcal{T}(\cdot)$, the following holds:

$$CS\text{-}GNN(\mathcal{T}, \pi_S) \subseteq CS\text{-}GNN(\mathcal{T}, \pi_{SS}). \tag{130}$$

**Minimum Distance $\Rightarrow$ Simple Node Marking**:

In this case, we aim to update the node feature map:

$$\mathcal{X}^0(S, v) = \mathcal{X}^{\pi_{MD}}(S, v) = \min_{v \in s} d_G(u, v) \tag{131}$$

We notice that:

$$\mathcal{X}^S(S, v) = g(\mathcal{X}^0(S, v)) \tag{132}$$

where $g : \mathbb{R} \to \mathbb{R}$ is any continuous function such that:

1. $g(x) = 1 \ \forall x > \frac{1}{2}$,

2. $g(x) = 0 \ \forall x < \frac{1}{4}$.

Using a CS-GNN update as per equation 15, with the update function:

$$f^1(\mathcal{X}^0(S, v), \cdot, \cdot, \cdot, \cdot) = g(\mathcal{X}^0(S, v)), \tag{133}$$

we obtain:

$$\mathcal{X}^1(S, v) = f^1(\mathcal{X}^0(S, v), \cdot, \cdot, \cdot, \cdot) = \mathcal{X}^{\pi_S}(S, v). \tag{134}$$

This implies that for any coarsening function $\mathcal{T}(\cdot)$ the following holds:

$$\text{CS-GNN}(\mathcal{T}, \pi_\text{S}) \subseteq \text{CS-GNN}(\mathcal{T}, \pi_\text{MD}). \tag{135}$$

**Simple Node Marking $\Rightarrow$ Node + Size Marking**: In this case, we aim to update the node feature map:

$$\mathcal{X}^0(S, v) = \mathcal{X}^{\pi_S}(S, v) = \begin{cases} 1 & v \in S \\ 0 & v \notin S. \end{cases} \tag{136}$$

We notice that:

$$\sum_{v' \in S} \mathcal{X}^0(S, v') = |S|. \tag{137}$$

Using a CS-GNN update as per Equation (15), with aggregation function:

$$\text{agg}_4^l \{\!\!\{ (\mathcal{X}^0(S, v'), z(S, v, S, v')) \mid v' \in S \}\!\!\} = \sum_{v' \in S} \mathcal{X}^0(S, v'), \tag{138}$$

and update function:

$$f^1 \left( \mathcal{X}^0(S, v), \cdot, \cdot, \cdot, \sum_{v' \in S} \mathcal{X}^0(S, v') \right) = \left( \mathcal{X}^0(S, v), \sum_{v' \in S} \mathcal{X}^0(S, v') \right), \tag{139}$$

we obtain:

$$\mathcal{X}^1(S, v) = f^1 \left( \mathcal{X}^0(S, v), \cdot, \cdot, \cdot, \sum_{v' \in S} \mathcal{X}^0(S, v') \right) = \mathcal{X}^{\pi_{SS}}(S, v). \tag{140}$$

This implies that for any coarsening function $\mathcal{T}(\cdot)$ the following holds:

$$\text{CS-GNN}(\mathcal{T}, \pi_\text{SS}) \subseteq \text{CS-GNN}(\mathcal{T}, \pi_\text{S}). \tag{141}$$

**Simple Node Marking $\Rightarrow$ Minimum Distance**:

In this case, we aim to update the node feature map:

$$\mathcal{X}^0(S, v) = \mathcal{X}^{\pi_S}(S, v) = \begin{cases} 1 & v \in S \\ 0 & v \notin S. \end{cases} \tag{142}$$

We shall prove that $\mathcal{X}^{\pi_\text{MD}}$ can be expressed by updating $\mathcal{X}^0(S, v)$ with a stack of CS-GNN layers. We do this by inductively showing that this procedure can express the following auxiliary node feature maps:

$$\mathcal{X}_*^t(S, v) = \begin{cases} \min_{v' \in S} d_G(v, v') + 1 & \min_{v' \in S} d_G(v, v') \le t \\ 0 & \text{otherwise.} \end{cases} \tag{143}$$

We notice first that:

$$\mathcal{X}^0(S, v) = \mathcal{X}_*^0(S, v). \tag{144}$$

Now for the induction step, assume that there exists a stack of $t$ CS-GNN layers such that:

$$\mathcal{X}^t(S, v) = \mathcal{X}_*^t(S, v). \tag{145}$$

We observe that equation:

$$\min_{v' \in S} d_G(v, v') = t + 1 \tag{146}$$

holds if and only if the following two conditions are met:

$$\min_{v' \in S} d_G(v, v') > t \tag{147}$$

$$\exists u \in N_G(v) \text{ s.t. } \min_{u' \in S} d_G(u, u') = t. \tag{148}$$

Equations 143 imply:

$$\min_{v' \in S} d_G(v, v') > t \Leftrightarrow \mathcal{X}^t(S, v) = 0. \tag{149}$$

In addition, since the node feature map $\mathcal{X}^t = \mathcal{X}_*^t$ is bounded by $t+1$, Equation (143) implies:

$$\exists u \in N_G(v) \text{ s.t. } \min_{u' \in S} d_G(u, u') = t \Leftrightarrow \max\{\mathcal{X}^t(s, u) \mid v \sim_G u\} = t + 1. \tag{150}$$

Now, let $g_t : \mathbb{R}^2 \to \mathbb{R}$ be any continuous function such that for every pair of natural numbers $a, b \in \mathbb{N}$:

1. $g_t(a, b) = t + 2$    if $a = 0, b = t + 1$,

2. $g_t(a, b) = a$    otherwise.

Equations 146 - 150 imply:

$$\mathcal{X}_*^{t+1}(S, v) = g_t(\mathcal{X}^t(S, v), \max\{\mathcal{X}^t(s, u) \mid v \sim_G u\}). \tag{151}$$

Using a CS-GNN update as per Equation (15), with aggregation function:

$$\text{agg}_1^t \{\!\!\{(\mathcal{X}^t(S, v'), e_{v,v'}) \mid v' \sim_G v\}\!\!\} = \max_{v' \sim_G v} \mathcal{X}^t(S, v'). \tag{152}$$

and update function:

$$f^t(\mathcal{X}^t(S, v), \max_{v' \sim_G v} \mathcal{X}^t(S, v'), \cdot, \cdot, \cdot) = g_t(\mathcal{X}^t(S, v), \max_{v' \sim_G v} \mathcal{X}^t(S, v')) \tag{153}$$

we obtain:

$$\mathcal{X}^{t+1}(S, v) = f^t(\mathcal{X}^t(S, v), \max_{v' \sim_G v} \mathcal{X}^t(S, v'), \cdot, \cdot, \cdot) = \mathcal{X}_*^{t+1}(S, v). \tag{154}$$

This completes the induction step. Now, let $\mathcal{G}$ be a finite family of graphs, whose maximal vertex size is $n$. We notice that:

$$\mathcal{X}^{\pi_{\text{MD}}}(S, v) = \mathcal{X}_*^n(S, v) - 1, \tag{155}$$

Which implies that there exists a stack of $n$ CS-GNN layers such that:

$$\mathcal{X}^0(S, v) = \mathcal{X}^{\pi_S}(S, v) \quad \text{and} \quad \mathcal{X}^n(S, v) = \mathcal{X}^{\pi_{\text{MD}}}(S, v). \tag{156}$$

This implies:

$$\text{CS-GNN}(\mathcal{T}, \pi_{\text{MD}}) \subseteq \text{CS-GNN}(\mathcal{T}, \pi_S). \tag{157}$$

This concludes the proof. $\qquad \square$

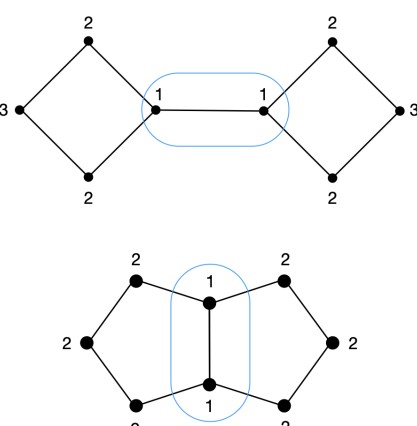

Figure 11: Graphs G and H defined in the proof of Proposition C.2. In each graph, the circle marks the single super-node induced by $\mathcal{T}$, while the number next to each node $u$ is the maximal SPD between $u$ and the nodes that compose the super-node.

**Proposition C.2** (Expressivity of Learned Distance Policy). *For any coarsening function $\mathcal{T}(\cdot)$ the following holds:*

$$CS\text{-}GNN(\mathcal{T}, \pi_S) \subseteq CS\text{-}GNN(\mathcal{T}, \pi_{LD}). \tag{37}$$

*In addition, for some choices of $\mathcal{T}(\cdot)$ the containment is strict.*

*Proof.* First, since we are concerned with input graphs belonging to a finite graph family $\mathcal{G}$, the learned function $\phi(\cdot)$ implemented by an MLP can express any continuous function on $\mathcal{G}$. This follows from Theorem G.1 (see the proof of Proposition B.1 for details). By choosing $\phi = \min(\cdot)$ in equation 35, it is clear that for any coarsening function $\mathcal{T}(\cdot)$ we have:

$$\text{CS-GNN}(\mathcal{T}, \pi_S) = \text{CS-GNN}(\mathcal{T}, \pi_{MD}) \subseteq \text{CS-GNN}(\mathcal{T}, \pi_{LD}). \tag{158}$$

We now construct a coarsening function $\mathcal{T}(\cdot)$ along with two graphs, $G$ and $H$, and demonstrate that there exists a function in CS-GNN$(\mathcal{T}, \pi_{LD})$ that can separate $G$ and $H$. However, every function in CS-GNN$(\mathcal{T}, \pi_S)$ cannot separate the two.

For an input graph $G = (V, E)$ define:

$$\mathcal{T}(G) = \{\{u \in V \mid \deg_G(u) = 3\}\}. \tag{159}$$

i.e., $\mathcal{T}(\cdot)$ returns a single super-node composed of all nodes with degree 3. Now, define $G = (V_G, E_G)$ as the graph obtained by connecting two cycles of size four by adding an edge between a single node from each cycle. Additionally, define $H = (V_H, E_H)$ as the graph formed by joining two cycles of size five along one of their edges. See Figure 11 for an illustration of the two graphs. By choosing $\phi = \max(\cdot)$ in equation 35 a quick calculation shows that:

$$\sum_{S \in V_{\mathcal{T}(G)}} \sum_{v \in V_G} \mathcal{X}^{\pi_{LD}}(S, v) = 16, \tag{160}$$

while:

$$\sum_{S \in V_{\mathcal{T}(H)}} \sum_{v \in V_H} \mathcal{X}^{\pi_{LD}}(S, v) = 14. \tag{161}$$

Refer to Figure 11 for more details. Observe that:

$$f(G) = \sum_{s \in V_{\mathcal{T}(H)}} \sum_{u \in V_H} \mathcal{X}^{\pi_{LD}}(S, v) \in \text{CS-GNN}(\mathcal{T}, \pi_{LD}) \tag{162}$$

Thus it is enough to show that:

$$f(G) = f(H), \quad \forall f \in \text{CS-GNN}(\mathcal{T}, \pi_S). \tag{163}$$

To achieve this, we use the layer update as per Definition B.2, which was demonstrated in Proposition B.1 to be equivalent to the general equivariant message passing update in Definition A.5. First, we observe that the graphs $G$ and $H$ are WL-indistinguishable. We then observe that since $|V^{\mathcal{T}}| = 1$, the graphs induced by the adjacency matrices $A_G$ and $A_H$ in Definition B.1 are isomorphic to the original graphs $G$ and $H$, respectively, and therefore they are also WL-indistinguishable. Additionally, we notice that the graphs induced by the adjacency matrices $A_{\mathcal{T}(G)}$ and $A_{\mathcal{T}(H)}$ in Definition B.1 are both isomorphic to the fully disconnected graph with 8 nodes, making them WL-indistinguishable as well. We also observe that there exists a bijection $\sigma : V_G \to V_H$ that maps all nodes of degree 3 in $G$ to all nodes of degree 3 in $H$. The definition of $\mathcal{T}(\cdot)$ implies that $\sigma$ is an isomorphism between the adjacency matrices $A_{P_i}$ corresponding to $G$ and $H$, where $i = 1, 2$. Finally, we notice that for both $G$, and $H$, the node feature map induced by $\pi_S$ satisfies:

$$\mathcal{X}^{\pi_S}(S, v) = \deg(v) - 2. \tag{164}$$

This node feature map can be easily implemented by the layer update in definition B.2 and so it can be ignored. Since all four graphs corresponding to $G$ that are induced by the adjacency matrices in Definition B.1, are WL-indistinguishable from their counterpart corresponding to $H$, and equation 29 in definition B.2 is an MPNN update, which is incapable of distinguishing graphs that are WL-indistinguishable, we see that equation 163 holds, concluding the proof. □

**Proposition C.3** (Node + Size Marking as Invariant Marking). *Given a graph $G = (V, E)$ with node feature vector $X \in \mathbb{R}^{n \times d}$, and a coarsening function $\mathcal{T}(\cdot)$, let $\mathcal{X}^{\pi_{SS}}, \mathcal{X}^{\pi_{inv}}$ be the node feature maps induced by $\pi_{SS}$ and $\pi_{inv}$ respectively. Recall that:*

$$\mathcal{X}^{\pi_{SS}}(S, v) = [X_v, b_{\pi_{SS}}(S, v)], \tag{43}$$

$$\mathcal{X}^{\pi_{inv}}(S, v) = [X_v, b_{\pi_{inv}}(S, v)]. \tag{44}$$

*The following now holds:*

$$b_{\pi_{inv}}(S, v) = OHE(b_{\pi_{SS}}(S, v)) \quad \forall S \in V^{\mathcal{T}}, \forall v \in V. \tag{45}$$

*Here, OHE denotes a one-hot encoder, independent of the choice of both $G$ and $\mathcal{T}$.*

*Proof.* Let $G = (V, E)$ be a graph with $V = [n]$, and let $\mathcal{T}(\cdot)$ be a coarsening function. Recall that the maps $b_{\pi_{SS}}(\cdot, \cdot)$ and $b_{\pi_{inv}}(\cdot, \cdot)$ are both independent of the connectivity of $G$ and are defined as follows:

$$b_{\pi_{SS}}(S, v) = \begin{cases} (1, |S|) & v \in S, \\ (0, |S|) & v \notin S. \end{cases} \tag{165}$$

$$b_{\pi_{inv}}(S, v) = [\mathbf{1}_{\gamma_1}(S, v), \dots, \mathbf{1}_{\gamma_k}(S, v)]. \tag{166}$$

Here, $v \in [n]$, $S \in \mathcal{T}([n]) \subseteq \mathcal{P}([n])$, $\gamma_1, \dots, \gamma_k$ is any enumeration of the set of all orbits $(\mathcal{P}([n]) \times [n])/S_n$, and $\mathbf{1}_{\gamma_i}$ denotes the indicator function of orbit $\gamma_i$. Since any tuple $(S, v) \in \mathcal{P}([n]) \times [n]$ belongs to exactly one orbit $\gamma_i$, we note that the right hand side of Equation (166) is a one-hot encoded vector. Thus, it suffices to show that for every $v, v' \in [n]$ and $S, S' \in \mathcal{P}([n])$, we have:

$$b_{\pi_{SS}}(S, v) = b_{\pi_{SS}}(S,' v') \Leftrightarrow b_{\pi_{inv}}(S, v) = b_{\pi_{inv}}(S,' v'). \tag{167}$$

This is equivalent to:

$$(\mathcal{P}([n]) \times [n])/S_n = \{\{(S, v) \mid |S| = i, \mathbf{1}_S(v) = j\} \mid i \in [n], j \in \{0, 1\}\}. \tag{168}$$

Essentially, this means that each orbit corresponds to a choice of the size of $s$ and whether $v \in S$ or not. To conclude the proof, it remains to show that for any two pairs $(S, v), (S,' v') \in \mathcal{P}([n]) \times [n]$, there exists a permutation $\sigma \in S_n$ such that:

$$\sigma \cdot (S, v) = (S,' v') \tag{169}$$

if and only if

$$|S| = |S'| \text{ and } \mathbf{1}_S(v) = \mathbf{1}_{S'}(v'). \tag{170}$$

Assume first that $\sigma \cdot (S, v) = (S', v')$, then $\sigma^{-1}(S) = S'$ and since $\sigma$ is a bijection, $|S| = |S'|$. In addition $\sigma^{-1}(v) = v'$ thus:

$$v \in S \Leftrightarrow v' = \sigma^{-1}(v) \in \sigma^{-1}(S) = S'. \tag{171}$$

Assume now that:

$$|S| = |S'| \tag{172}$$

$$\mathbf{1}_S(v) = \mathbf{1}_{S'}(v') \tag{173}$$

It follows that for some $r, m \in [n]$:

$$|S \setminus \{v\}| = |S' \setminus \{v'\}| = r \quad \text{and} \quad |[n] \setminus (S \cup \{v\})| = |[n] \setminus (S' \cup \{v'\})| = m \tag{174}$$

Write:

$$S \setminus \{v\} = \{i_1, \dots, i_r\}, \quad S' \setminus \{v'\} = \{i'_1, \dots, i'_r\},$$
$$[n] \setminus (S \cup \{v\}) = \{j_1, \dots j_m\}, \quad [n] \setminus (S' \cup \{v'\}) = \{j'_1, \dots j'_m\}$$

and define:

$$\sigma(x) = \begin{cases} v' & x = v \\ i'_l & x = i_l, l \in [r] \\ j'_l & x = j_l, l \in [m] \end{cases} \tag{175}$$

We now have:

$$\sigma \cdot (S, v) = (S', v'). \tag{176}$$

This concludes the proof. $\qquad \square$

## G.3 Proofs of Appendix D.1

**Proposition D.1** (CS-GNN Can Implement MSGNN). *Let $\mathcal{T}(\cdot)$ be the identity coarsening function defined by:*

$$\mathcal{T}(G) = \{\{v\} \mid v \in V\} \quad \forall G = (V, E). \tag{49}$$

*The following holds:*

$$\text{CS-GNN}(\mathcal{T}, \pi_S) = \text{MSGNN}(\pi_{NM}). \tag{50}$$

*Proof.* Abusing notation, for a given graph $G = (V, E)$ we write $\mathcal{T}(G) = G$, $V^{\mathcal{T}} = V$. First, we observe that:

$$v \in \{u\} \Leftrightarrow u = v, \tag{177}$$

This implies that the initial node feature map $\mathcal{X}^0(u, v)$ induced by $\pi_S$ is equivalent to the standard node marking described in equation 46. Additionally, we note that the pooling procedures for both models, as described in equations 16 and 55, are identical. Therefore, it is sufficient to show that the CS-GNN and MSGNN layer updates described in equations 15 and 47 respectively are also identical. For this purpose, let $\mathcal{X}^t(v, u)$ be a node feature map supported on the set $V \times V$. The inputs to the MSGNN layer are the following:

1. $\mathcal{X}^t(u, v)$.

2. $\mathcal{X}^t(u, u)$.

3. $\mathcal{X}^t(v, v)$.

4. $\text{agg}_1^t \{\!\{ (\mathcal{X}^t(u, v'), e_{v,v'}) \mid v' \sim v \}\!\}$.

5. $\text{agg}_2^t \{\!\{ (\mathcal{X}^t(u', v), e_{u,u'}) \mid u' \sim u \}\!\}$.

The inputs to the CS-GNN layer are the following:

1. $\mathcal{X}^t(S, v) \Rightarrow \mathcal{X}^t(u, v)$.

2. $\text{agg}_1^t \{\!\{ (\mathcal{X}^t(S, v'), e_{v,v'}) \mid v' \sim_G v \}\!\} \Rightarrow \text{agg}_1^t \{\!\{ (\mathcal{X}^t(u, v'), e_{v,v'}) \mid v' \sim v \}\!\}$.

3. $\text{agg}_2^t \{\!\{ (\mathcal{X}^t(S', v), \tilde{e}_{S,S'}) \mid S' \sim_{G^{\mathcal{T}}} S \}\!\} \Rightarrow \text{agg}_2^t \{\!\{ (\mathcal{X}^t(u, u'), e_{u,v'}) \mid v' \sim v \}\!\}$.

4. $\text{agg}_3^t \{\!\{ (\mathcal{X}^t(S', v), z(S, v, S', v)) \mid \forall s' \in V^{\mathcal{T}} \text{ s.t. } v \in S' \}\!\} \Rightarrow \{\!\{ (X^t(v, v), z(u, v, v, v)) \}\!\}$.

5. $\text{agg}_4^t \{\!\{ (\mathcal{X}^t(S, v'), z(S, v, S, v')) \mid \forall u' \in V \text{ s.t. } v' \in S \}\!\} \Rightarrow \{\!\{ (X^t(u, u), z(u, v, u, u)) \}\!\}$.

The terms $z(u, v, v, v)$ and $z(u, v, u, u)$ appearing in the last two input terms of the CS-GNN layer uniquely encode the orbit tuples $(u, v, v, v)$ and $(u, v, u, u)$ belong to respectively. Since these orbits depend solely on whether $u = v$, these values are equivalent to the node marking feature map $\mathcal{X}^0(u, v)$. Therefore, these terms can be ignored. Observing the two lists above, we see that the inputs to both update layers are identical (ignoring the $z(\cdot)$ terms), Thus, as both updates act on these inputs in the same way, the updates themselves are identical. and so

$$\text{MSGNN}(\pi_{\text{NM}}) = \text{CS-GNN}(\mathcal{T}, \pi_S). \tag{178}$$

$\square$

## G.4 Proofs of Appendix D.2

**Proposition D.2** (CS-GNN Is at Least as Expressive as Coarse MPNN ). *For any coarsening function $\mathcal{T}(\cdot)$ the following holds:*

$$\text{MPNN} \subseteq \text{MPNN}_+(\mathcal{T}) \subseteq \text{CS-GNN}(\mathcal{T}, \pi_S) \tag{58}$$

*Proof.* For convenience, let us first restate the CS-GNN layer update:

$$\mathcal{X}^{t+1}(S,v) = f^t\Bigg(\mathcal{X}^t(S,v),$$
$$\text{agg}_1^t\{\!\{(\mathcal{X}^t(S,v'),e_{v,v'}) \mid v' \sim_G v\}\!\},$$
$$\text{agg}_2^t\{\!\{(\mathcal{X}^t(S',v),\tilde{e}_{S,S'}) \mid S' \sim_{G^\mathcal{T}} S\}\!\},$$
$$\text{agg}_3^t\{\!\{(\mathcal{X}^t(S',v),z(S,v,S',v)) \mid s' \in V^\mathcal{T}\text{s.t. } v \in S'\}\!\},$$
$$\text{agg}_4^t\{\!\{(\mathcal{X}^t(S,v'),z(S,v,S,v')) \mid u' \in V\text{s.t. } v' \in S\}\!\}\Bigg), \tag{15}$$

as well as the MPNN$_+$ layer update:

$$\text{For } v \in V: \quad \mathcal{X}^{t+1}(v) = f_V^t\left(\mathcal{X}^t(v),\text{agg}_1^t\{\!\{(\mathcal{X}^t(v'),e_{v,v'}) \mid v \sim_G v'\}\!\},\right.$$
$$\left.\text{agg}_2^t\{\!\{\mathcal{X}^t(S) \mid S \in V^T, v \in S\}\!\}\right),$$
$$\text{For } S \in V^\mathcal{T}: \quad \mathcal{X}^{t+1}(S) = f_{V^\mathcal{T}}^t\left(\mathcal{X}^t(S),\text{agg}_1^t\{\!\{(\mathcal{X}^t(S'),e_{S,S'}) \mid S \sim_{G^\mathcal{T}} S'\}\!\},\right.$$
$$\left.\text{agg}_2^t\{\!\{\mathcal{X}^t(v) \mid v \in V, v \in S\}\!\}\right). \tag{54}$$

We note that by setting $f_{V^\mathcal{T}}^t$ to be a constant zero and choosing $f_V^t$ to be any continuous function that depends only on its first two arguments, the update in equation 54 becomes a standard MPNN layer. This proves:

$$\text{MPNN} \subseteq \text{MPNN}_+(T). \tag{179}$$

Next, we prove the following 2 Lemmas:

**Lemma G.1.** *Given a graph $G = (V,E)$ such that $V = [n]$ with node feature vector $X \in \mathbb{R}^{n \times d}$, and a coarsening function $\mathcal{T}(\cdot)$, there exists a CS-GNN$(\mathcal{T},\pi_S)$ layer such that:*

$$\mathcal{X}^1(S,v) = [0_{d+1}, X_v, 1] = [\tilde{\mathcal{X}}^0(S), \tilde{\mathcal{X}}^0(v)]. \tag{180}$$

*Here $[\cdot,\cdot]$ denotes concatenation and $\tilde{\mathcal{X}}^0(\cdot)$ denotes the initial node feature map of the coarsened sum graph $G_+^\mathcal{T}$.*

**Lemma G.2.** *Let $\tilde{\mathcal{X}}^t(\cdot)$ denote the node feature maps of $G_+^T$ at layers $t$ of a stack of MPNN$_+(\mathcal{T})$ layers. There exists a stack of $t+1$ CS-GNN$(\mathcal{T},\pi_S)$ layers such that:*

$$\mathcal{X}^{t+1}(S,v) = [\tilde{\mathcal{X}}^t(S), \tilde{\mathcal{X}}^t(v)]. \tag{181}$$

*proof of Lemma G.1.* Recall that the initial node feature map of CS-GNN$(\mathcal{T},\pi_S)$ is given by:

$$\mathcal{X}^0(S,v) = \begin{cases} [X_v, 1] & v \in S \\ [X_v, 0] & v \notin S. \end{cases} \tag{182}$$

In addition, the initial node feature map of MPNN$_+(\mathcal{T})$ is given by:

$$\tilde{X}^0(v) = \begin{cases} [X_v, 1] & v \in V \\ 0_{d+1} & v \in V^\mathcal{T}. \end{cases} \tag{183}$$

Thus, we choose a layer update as described in equation 15 with:

$$\mathcal{X}^1(S,v) = f^0(\mathcal{X}^0(S,v),\cdot,\cdot,\cdot,\cdot) = [0_{d+1}, \mathcal{X}^0(S,v)_{1:d}, 1] \tag{184}$$

Here, $f(a,\cdot,\cdot,\cdot)$ denotes that the function depends only on the parameter $a$, and $X_{a:b}$ indicates that only the coordinates $a$ through $b$ of the vector $X$ are taken. This gives us:

$$\mathcal{X}^1(S,v) = [\tilde{\mathcal{X}}^0(S), \tilde{\mathcal{X}}^0(v)]. \tag{185}$$

$\square$

*proof of Lemma G.2.* We prove this Lemma by induction on $t$. We note that Lemma G.1 provides the base case $t = 0$. Assume now that for a given stack of $t + 1$ MPNN$_+(\mathcal{T})$ layer updates, with corresponding node feature maps:

$$\tilde{\mathcal{X}}^i : V_+^{\mathcal{T}} \to \mathbb{R}^{d_i} \quad i = 1\ldots, t+1, \tag{186}$$

there exists a stack of $t + 1$ CS-GNN$(\mathcal{T}, \pi_S)$ layers with node feature maps:

$$\mathcal{X}^i : V^{\mathcal{T}} \times V \to \mathbb{R}^{2d_i} \quad i = 1, \ldots, t+1, \tag{187}$$

such that:

$$\mathcal{X}^{t+1}(S, v) = [\tilde{\mathcal{X}}^t(S), \tilde{\mathcal{X}}^t(v)]. \tag{188}$$

We shall show that there exists a single additional CS-GNN$(\mathcal{T}, \pi_S)$ layer update such that:

$$\mathcal{X}^{t+2}(S, v) = [\tilde{\mathcal{X}}^{t+1}(S), \tilde{X}^{t+1}(v)]. \tag{189}$$

For that purpose we define the following CS-GNN$(\mathcal{T}, \pi_S)$ update (abusing notation, the left hand side refers to components of the CS-GNN$(\mathcal{T}, \pi_S)$ update at layer $t + 1$, while the right hand side refers to components of the MPNN$_+(\mathcal{T})$ update at layer $t$):

$$\begin{aligned}
\text{agg}_1^{t+1} &= \text{agg}_1^t|_{1:d_t}, \\
\text{agg}_2^{t+1} &= \text{agg}_1^t|_{d_t+1:2d_t}, \\
\text{agg}_3^{t+1} &= \text{agg}_2^t|_{1:d_t}, \\
\text{agg}_4^{t+1} &= \text{agg}_2^t|_{d_t+1:2d_t},
\end{aligned} \tag{190}$$

$$f^{t+1}(a, b, c, d, e) = [f_V^t(a_{1:d_t}, b, d), f_{V\mathcal{T}}^t(a_{d_t+1:2d_t}, c, e)]. \tag{191}$$

Here the operation $\text{agg}|_{a:b}$ initially projects all vectors in the input multi-set onto coordinates $a$ through $b$, and subsequently passes them to the function agg. equations 190 , 191 guarantee that:

$$\begin{aligned}
\mathcal{X}^{t+2}(S, v)_{1:d_{t+1}} &= f_V^t\big(\mathcal{X}^t(S, v)_{1:d_t}, \\
&\quad \text{agg}_1^t\{\!\{(S, v')_{1:d_t} \mid v \sim_G v'\}\!\}, \\
&\quad \text{agg}_2^t\{\!\{(S', v)_{1:d_t} \mid v \in S'\}\!\}\big) \\
&= \tilde{X}^{t+1}(v), \\
\mathcal{X}^{t+2}(S, v)_{d_{t+1}+1:2d_{t+1}} &= f_{V\mathcal{T}}^t\big(\mathcal{X}^t(S, v)_{d_t+1:2d_t}, \\
&\quad \text{agg}_1^t\{\!\{(S', v)_{d_t+1:2d_t} \mid S' \sim_{\mathcal{T}(G)} S\}\!\}, \\
&\quad \text{agg}_2^t\{\!\{(S, v')_{d_t+1:2d_t} \mid v' \in S\}\!\}\big) \\
&= \tilde{X}^{t+1}(S).
\end{aligned} \tag{192}$$

This proves the Lemma. $\qquad\square$

Now, for a given finite family of graphs $\mathcal{G}$ and a function $f \in \text{MPNN}_+(\mathcal{T})$, there exists a stack of $T$ MPNN$_+(\mathcal{T})$ layers such that:

$$f(G) = U\left(\sum_{v \in V_+^{\mathcal{T}}} \tilde{\mathcal{X}}^T(v)\right) \quad \forall G \in \mathcal{G}. \tag{193}$$

Here, $\tilde{\mathcal{X}}^T : V_+^{\mathcal{T}} \to \mathbb{R}^{d_T}$ denotes the final node feature map, and $U$ is an MLP. Lemma G.2 now tells us that there exists a stack of $T + 1$ CS-GNN$(\mathcal{T}, \pi_S)$ layers such that:

$$\mathcal{X}^{T+1}(S, v) = [\tilde{\mathcal{X}}^T(S), \tilde{\mathcal{X}}^T(v)]. \tag{194}$$

Similarly to Lemma G.1, we use one additional layer to pad $\mathcal{X}^{T+1}(S, v)$ as follows:

$$\mathcal{X}^{T+2}(S, v) = [\tilde{\mathcal{X}}^T(S), \tilde{\mathcal{X}}^T(v), 1]. \tag{195}$$

We notice that:

$$\sum_{s \in V^{\mathcal{T}}} \mathcal{X}^{T+2}(S, v) = \left[ \sum_{S \in V^{\mathcal{T}}} \tilde{X}^T(S), \sum_{S \in V^{\mathcal{T}}} \tilde{X}^T(v), \sum_{S \in V^{\mathcal{T}}} 1 \right]$$

$$= \left[ \sum_{S \in V^{\mathcal{T}}} \tilde{X}^T(S), |V^{\mathcal{T}}| \cdot \tilde{X}^T(v), |V^{\mathcal{T}}| \right]. \tag{196}$$

Thus, in order to get rid of the $|V^{\mathcal{T}}|$ term, We define:

$$\mathrm{MLP}_1(a, b, c) = [a, \frac{1}{c} \cdot b, 1], \quad a, b \in \mathbb{R}^{d_L}, c > 0. \tag{197}$$

We note that since we are restricted to a finite family of input graphs, the use of an MLP in equation 200 can be justified using Theorem G.1 (see the proof of Proposition B.1 for a detailed explanation).

Equations 196 and 200 imply:

$$\mathrm{MLP}_1 \left( \sum_{s \in V^{\mathcal{T}}} \mathcal{X}^{T+2}(S, v) \right) = \left[ \sum_{S \in V^{\mathcal{T}}} \tilde{X}^T(S), \tilde{X}^T(v), 1 \right] \tag{198}$$

Thus, similarly to equation 196:

$$\sum_{v \in V} \mathrm{MLP}_1 \left( \sum_{S \in V^{\mathcal{T}}} \mathcal{X}^{T+2}(S, v) \right) = \left[ |V| \cdot \sum_{S \in V^{\mathcal{T}}} \tilde{X}^T(S), \sum_{v \in V} \tilde{X}^T(v), |V| \right] \tag{199}$$

And so, in order to get rid of the $|V|$ term, We define:

$$\mathrm{MLP}_2(a, b, c) = U(a \cdot \frac{1}{c} + b, 1), \quad a, b \in \mathbb{R}^{d_T}, c > 0. \tag{200}$$

Thus for all $G \in \mathcal{G}$:

$$\mathrm{MLP}_2 \left( \sum_{v \in V} \mathrm{MLP}_1 \left( \sum_{S \in V^{\mathcal{T}}} \mathcal{X}^{T+2}(S, v) \right) \right)$$

$$= \mathrm{MLP}_2 \left( \left[ |V| \cdot \sum_{S \in V^{\mathcal{T}}} \tilde{X}^T(S), \sum_{v \in V} \tilde{X}^T(v), |V| \right] \right) \tag{201}$$

$$= U \left( \sum_{v \in V_+^{\mathcal{T}}} \tilde{X}^T(v) \right)$$

$$= f(G).$$

and so $f \in \mathrm{CS\text{-}GNN}(\mathcal{T}, \pi_S)$. This proves:

$$\mathrm{MPNN}_+(T) \subseteq \mathrm{CS\text{-}GNN}(\mathcal{T}, \pi_S). \tag{202}$$

$\square$

**Proposition D.3** (CS-GNN Can Be More Expressive Than MPNN+). *Let $\mathcal{T}(\cdot)$ be the identity coarsening function defined by:*

$$\mathcal{T}(G) = \{\{v\} \mid v \in V\} \quad G = (V, E). \tag{59}$$

*The following holds:*

$$MPNN = MPNN_+(\mathcal{T}). \tag{60}$$

*Thus:*

$$MPNN_+(\mathcal{T}) \subset CS\text{-}GNN(\mathcal{T}, \pi_S), \tag{61}$$

*where this containment is strict.*

*Proof.* First, using the notation $\tilde{v}$ to mark the single element set $\{v\} \in V^{\mathcal{T}}$, We notice that the MPNN$_+(\mathcal{T})$ layer update described in equation 54, becomes:

$$\text{For } v \in V : \quad \mathcal{X}^{t+1}(v) = f_V^t\left(\mathcal{X}^t(v), \mathcal{X}^t(\tilde{v}), \text{agg}^t\{\!\!\{(\mathcal{X}^t(v'), e_{v,v'}) \mid v' \sim_G v\}\!\!\},\right),$$

$$\text{For } \tilde{v} \in V^{\mathcal{T}} : \quad \mathcal{X}^{t+1}(\tilde{v}) = f_{V^{\mathcal{T}}}^t\left(\mathcal{X}^t(\tilde{v}), \mathcal{X}^t(v), \text{agg}^t\{\!\!\{(\mathcal{X}^t(\tilde{v}'), e_{\tilde{v},\tilde{v}'}) \mid v \sim_G v'\}\!\!\}\right). \tag{203}$$

Now, for a given finite family of graphs $\mathcal{G}$ and a function $f \in \text{MPNN}_+(\mathcal{T})$, there exists a stack of $T$ MPNN$_+(\mathcal{T})$ layers such that:

$$f(G) = U\left(\sum_{v \in V_+^{\mathcal{T}}} \mathcal{X}^T(v)\right) \quad \forall G \in \mathcal{G}. \tag{204}$$

Here, $\mathcal{X}^T : V_+^{\mathcal{T}} \to \mathbb{R}^d$ denotes the final node feature map, and $U$ is an MPL. We now prove by induction on $t$ that there exists a stack of $t$ standard MPNN layers, with corresponding node feature map $X^t : V \to \mathbb{R}^{2d_t}$ such that :

$$X^t(v) = [\mathcal{X}^t(v), \mathcal{X}^t(\tilde{v})]. \tag{205}$$

Here, $[\cdot, \cdot]$ stands for concatenation. We assume for simplicity that the input graph $G$ does not have node features, though the proof can be easily adapted for the more general case. We notice that for the base case $t = 0$, equation 53 in definition D.2 implies:

$$\mathcal{X}^0(v) = \begin{cases} 1 & v \in V, \\ 0 & v \in V^{\mathcal{T}}. \end{cases} \tag{206}$$

Thus, we define:

$$X^0(v) = (1, 0). \tag{207}$$

This satisfies Equation (205), establishing the base case of the induction. Assume now that Equation (205) holds for some $t \in [T]$. Let $\text{agg}^t, f_V^t, f_{V^{\mathcal{T}}}^t$ be the components of layer $t$, as in equation 203. We define:

$$\tilde{\text{agg}}^t = [\text{agg}^t|_{1:d_t}, \text{agg}^t|_{d_t+1:2d_t}]. \tag{208}$$

Here the operation $\text{agg}|_{a:b}$ initially projects all vectors in the input multi-set onto coordinates $a$ through $b$, and subsequently passes them to the function agg.

Additionally, let $d^*$ denote the dimension of the output of the function $\text{agg}^t$. We define:

$$\tilde{f}^t(a, b) = \left[f_V^t\left(a|_{1:d_t}, a|_{d_t+1:2d_t}, b|_{1:d^*}\right), f_V^t\left(a|_{d_t+1:2d_t}, a|_{1:d_t}, b|_{d^*+1:2d^*}\right)\right]. \tag{209}$$

Finally, we update our node feature map $X^t$ using a standard MPNN update according to:

$$X^{t+1}(v) = \tilde{f}^l\left(X^t(v), \{\!\!\{(X^t(v'), e_{v,v'}) \mid v' \sim_G v\}\!\!\}\right). \tag{210}$$

equations 203, 205 and 210 now guarantee that:

$$X^{t+1}(v) = [\mathcal{X}^t(v), \mathcal{X}^{t+1}(\tilde{v})]. \tag{211}$$

This concludes the inductive proof. We now define:

$$\text{MLP}(x) = U(x|_{1:d_T}) + U(x|_{d_T+1:2d_T}). \tag{212}$$

This gives us:

$$U\left(\sum_{v \in V_+^{\mathcal{T}}} \mathcal{X}^T(v)\right) = \text{MLP}\left(\sum_{v \in V} X^T(v)\right) = f(G). \tag{213}$$

We have thus proven that $f \in \text{MPNN}$ and so:

$$\text{MPNN}_+(\mathcal{N}) \subseteq \text{MPNN}. \tag{214}$$

Combining this result with Proposition D.2, we obtain:

$$\text{MPNN} = \text{MPNN}_+(\mathcal{T}). \tag{215}$$

Finally, since Proposition D.1 tells us that CS-GNN$(\mathcal{T}, \pi_S)$ has the same implementation power as the maximally expressive node policy subgraph architecture MSGNN, which is proven to be strictly more expressive than the standard MPNN, we have:

$$\text{MPNN}_+(\mathcal{T}) \subset \text{CS-GNN}(\mathcal{T}, \pi_S). \tag{216}$$

$\square$

**Proposition D.4** (CS-GNN can be strictly more expressive then node-based subgraph GNNs). *Let* $\mathcal{T}$ *be the coarsening function defined by:*

$$\mathcal{T}(G) = \{\{v\} \mid v \in V\} \cup E \quad G = (V, E). \tag{63}$$

*The following holds:*

1. *Let* $G_1, G_2$ *be a pair of graphs such that there exists a node-based subgraph GNN model* $M$ *where* $M(G_1) \neq M(G_2)$. *There exists a CS-GNNmodel* $M'$ *which uses* $\mathcal{T}$ *such that* $M'(G_1) \neq M'(G_2)$.

2. *There exists a pair of graphs* $G_1, G_2$ *such that for any subgraph GNN model* $M$ *it holds that* $M(G_1) = M(G_2)$, *but there exists a CS-GNNmodel* $M'$ *which uses* $\mathcal{T}$ *such that* $M'(G_1) \neq M'(G_2)$.

*Proof.* First, notice that the super-nodes produced by $\mathcal{T}$ are either of size 1, in which case they correspond to nodes, or they are of size two, in which case they correspond to edges. Since an CS-GNNmodel processes feature maps $\mathcal{X}^t(S, v)$ where in the initial layer the sset size of $S$ is encoded in $\mathcal{X}^t(S, v)$, we can easily use the CS-GNNupdate in Definition A.5 to ignore all values of $\mathcal{X}^t(S, v)$ were $|S| = 2$ (This can be done by using $f^t, \text{agg}_1^t, \ldots \text{agg}_1^t$ in Definition A.5 to zero out these values at each update). This means CS-GNNusing $\mathcal{T}$ is able to simulate an CS-GNNupdate with the identity coarsening function, which was shown in Proposition D.1 to be as expressive as GNN-SSWL+ (Definition D.1) which is a maximally expressive node-based subgraph GNN, thus proving part (1) of the proposition. To prove part (2), notice that using the same reasoning as before, an CS-GNNmodel using $\mathcal{T}$ as a coarsening function cal implement an CS-GNNmodel using the edge coarsening function:

$$\mathcal{T}'(G) = E \quad G = (V, E). \tag{217}$$

An CS-GNNmodel with the identity coarsening function can be interpreted as a GNN-SSWL+ model. Similarly, an CS-GNNmodel using the edge coarsening function $\mathcal{T}'$ generalizes the GNN-SSWL+ framework by extending it from node-based subgraph GNNs to edge-based subgraph GNNs. In fact, the same proof in [38, 12] showing that GNN-SSWL+ is at least as expressive as a DSS subgraph GNN using the node deletion policy (see [4] for a definition of the DSS subgraph GNN), can be used to show that CS-GNNusing the edge coarsening function $\mathcal{T}'$ is at least as expressive as a DSS subgraph GNN with an edge deletion policy. The latter model was shown in [4] to be able to separate a pair of 3-WL indistinguishable graphs. In contrast, node-based subgraph GNNs were shown in [12] to not be able to separate any pair of 3-WL indistinguishable graphs. Thus, there exists a pair of graphs which CS-GNNusing $\mathcal{T}$ can separate while node-based subgraph GNNs cant, proving part (2) of the proposition.

$\square$

## G.5  Proofs of Appendix E

**Lemma E.1** ($\gamma (\Gamma)$ are orbits). *The sets* $\{\gamma^{k^*} : k = 1, \ldots, n; * \in \{+, -\}\}$ *and* $\{\Gamma^{\leftrightarrow; k_1; k_2; k^\cap; \delta_{same}; \delta_{diff}}\}$ *are the orbits of* $S_n$ *on the index space* $(\mathcal{P}([n]) \times [n])$ *and* $(\mathcal{P}([n]) \times [n] \times (\mathcal{P}([n]) \times [n])$, *respectively.*

*Proof.* We will prove this lemma for $\gamma$. The proof for $\Gamma$ follows similar reasoning; we also refer the reader to [22] for a general proof.

We will prove this lemma through the following three steps.

**(1).** Given indices $(S, i) \in \mathcal{P}([n]) \times [n]$, there exists $\gamma \in (\mathcal{P}([n]) \times [n])_\sim$ such that $(S, i) \in \gamma$.

**(2).** Given indices $(S, i) \in \gamma$, for any $\sigma \in S_n$, it holds that $(\sigma^{-1}(S), \sigma^{-1}(i)) \in \gamma$.

**(3).** Given $(S, i) \in \gamma$ and $(S', i') \in \gamma$ (the same $\gamma$), it holds that there exists a $\sigma \in S_n$ such that $\sigma \cdot (S, i) = (S', i')$.

We prove in what follows.

**(1).** Given indices $(S, i) \in \mathcal{P}([n]) \times [n]$, w.l.o.g. we assume that $|S| = k$, thus if $i \in S$ ($i \notin S$) it holds that $(S, i) \in \gamma^{k^-}$ $\left((S, i) \in \gamma^{k^+}\right)$, recall Equation (71).

**(2).** Given indices $(S, i) \in \gamma$, note that any permutation $\sigma \in S_n$ does not change the cardinality of $S$ nor the inclusion (or exclusion) of $i$ in $S$. Recalling Equation (71), we complete this step.

**(3).** Given that $(S, i) \in \gamma$ and $(S', i') \in \gamma$, and recalling Equation (71), we note that $|S| = |S'|$ and that either both $i \in S$ and $i' \in S'$, or both $i \notin S$ and $i' \notin S'$.

**(3.1).** In **(3.1)** we focus on the case where $i \notin S$ and $i' \notin S'$. Let $S = \{i_1, \ldots, i_k\}$ and $S' = \{i_1', \ldots, i_k'\}$. Then, we have $(\{i_1, \ldots, i_k\}, j)$ and $(\{i_1', \ldots, i_k'\}, j')$. Define $\sigma \in S_n$ such that $\sigma(i_l) = i_l'$ for $l \in [k]$, and $\sigma(j) = j'$. Since $(\{i_1, \ldots, i_k\}, j)$ consists of $k + 1$ distinct indices and $(\{i_1', \ldots, i_k'\}, j')$ also consists of $k + 1$ distinct indices, this is a valid $\sigma \in S_n$.

**(3.2).** Here, we focus on the case where $i \in S$ and $i' \in S'$. This proof is similar to **(3.1)**, but without considering the indices $j$ and $j'$, as they are included in $S$ and $S'$, respectively.

$\square$

**Proposition E.1** (Basis of Invariant (Equivariant) Layer). *The tensors $\mathbf{B}^\gamma$ ($\mathbf{B}^\Gamma$) in Equation (72) (Equation (74)) form an orthogonal basis (in the standard inner product) to the solution of Equation (66) (Equation (67)).*

*Proof.* We prove this proposition for the invariant case. The equivariant case is proved similarly – we also refer the reader for [22] for a general proof. We will prove this in three steps,

**(1).** For any $\gamma \in (\mathcal{P}([n]) \times [n])_\sim$ it holds that $\mathbf{B}_{S,i}^\gamma$ solves Equation (66).

**(2).** Given a solution $\mathbf{L}$ to Equation (66), it is a linear combination of the basis elements.

**(3).** We show that the basis vectors are orthogonal and thus linearly independent.

We prove in what follows.

**(1).** Given $\gamma \in (\mathcal{P}([n]) \times [n])_\sim$, we need to show that $\mathbf{B}_{S,i}^\gamma = \mathbf{B}_{\sigma^{-1}(S), \sigma^{-1}(i)}^\gamma$. Since any $\gamma \in (\mathcal{P}([n]) \times [n])_\sim$ is an orbit in the index space (recall Lemma E.1), and $\mathbf{B}_{S,i}^\gamma$ are indicator vectors of the orbits this always holds.

**(2).** Given a solution $\mathbf{L}$ to Equation (66), it must hold that $\mathbf{L}_{S,i} = \mathbf{L}_{\sigma^{-1}(S), \sigma^{-1}(i)}$. Since the set $\{\gamma^{k^*} : k = 1, \ldots, n; * \in \{+, -\}\}$ corresponds to the orbits in the index space with respect to $S_n$, $\mathbf{L}$ should have the same values over the index space of these orbits. Let's define these values as $\alpha^\gamma$ for each $\gamma \in \{\gamma^{k^*} : k = 1, \ldots, n; * \in \{+, -\}\}$. Thus, we obtain that $\mathbf{L}' = \sum_{\gamma \in (\mathcal{P}([n]) \times [n])_\sim} \alpha^\gamma \cdot \mathbf{B}^\gamma$, since $\mathbf{B}^\gamma$ are simply indicator vectors of the orbits. This completes this step.

**(3).** Once again, since the basis elements are indicator vectors of disjoint orbits we obtain their orthogonality, and thus linearly independent. $\square$

