# OpenReview forum: "A Flexible, Equivariant Framework for Subgraph GNNs via Graph Products and Graph Coarsening"
_NeurIPS.cc/2024/Conference — NeurIPS 2024 poster_

### Official Review · Reviewer_DujM · 2024-07-06

**Soundness:** 3
**Presentation:** 3
**Contribution:** 3
**Rating:** 7
**Confidence:** 2

**Summary:**

This paper proposes CS-GNN, a subgraph GNN approach utilizing graph coarsening and graph Cartesian products. This novel and flexible subgraph GNN can effectively generate and process any desired bag size. The paper also discovers new permutation symmetries in the produced node feature tensor during generalized message passing. Theoretical and empirical analysis validate the efficacy of the proposed method.

**Strengths:**

1. The topic of subgraph GNNs is interesting.
2. The paper is generally well-written with clarity.
3. The proposed method is novel, with theoretical underpinnings.
4. The empirical results seem promising.
5. The implementation code is provided for reproducing the results.

**Weaknesses:**

1. The size of the coarsened graph, though sparse, is still very large with $2^n$ nodes, which might result in high computation complexity for large input graphs.
2. Eq. (3) indicates that the input graph is unweighted and edge weights might not be considered.

**Questions:**

1. The size of the coarsened graph, though sparse, is still very large with $2^n$ nodes, which might result in high computation complexity for large input graphs. Do you have an estimate of the complexity and runtime of the proposed model compared with others?
2. Eq. (3) indicates that the input graph is unweighted and edge weights might not be considered. What if the input graph is weighted, for example, with edge weights 1,2,3, and 4.

**Limitations:**

Yes, the authors have adequately addressed the limitations and, if applicable, potential negative societal impact of their work.

---

> ### Author Rebuttal · Authors · 2024-08-05
>
> We appreciate your positive feedback! We are pleased to hear that you find our method novel with solid theoretical foundations. We're also glad that you find the empirical results promising. Below, we address your specific comments and questions.
>
>  **Q1:** *The size of the coarsened graph, though sparse, is still very large with $2^n$ nodes, which might result in high computation complexity for large input graphs. Do you have an estimate of the complexity and runtime of the proposed model compared with others?*
>
>  **A1:** Let us clarify our approach. In our method, the size of the coarsened graph is always less than or equal to that of the original graph. **To further emphasize, the space complexity of storing this sparse graph is upper bounded by the space complexity of storing the original graph $G$ (we do not store $2^n$ nodes)**. As shown in Figure 1 (left), the coarsened graph $\mathcal{T}(G)$ has fewer nodes than the original graph $G$. Considering the feature dimension to be constant, and given that $V$ is the number of nodes in the original graph, $T$ is the bag size, and $\Delta_{\text{max}}$ is the maximal degree of the original graph, our time complexity is upper-bounded by $\mathcal{O}(T \cdot V \cdot (\Delta_{\text{max}} + T))$. When considering $T=\mathcal{O}(1)$, which is typically the case, the complexity simplifies to $\mathcal{O}(V \cdot \Delta_{\text{max}})$. The space complexity is $\mathcal{O}(T \cdot E + V \cdot T^2)$, where $E$ is the number of edges in the original graph, which reduces to $\mathcal{O}(V +E) $, when $T = \mathcal{O}(1)$. This complexity is comparable to that of other methods in the field [1, 2]. Additionally, we refer the reader to Table 10 in Appendix F.5 (page 28) for a detailed comparison of the runtime of our method with other baselines. Our method's runtime is comparable to theirs.
>
> **Q2:** Eq. (3) indicates that the input graph is unweighted and edge weights might not be considered. What if the input graph is weighted, for example, with edge weights 1,2,3, and 4.
>
> **A2:** Thank you for highlighting this point. Our method is capable of handling weighted graphs. We employ a GINE base encoder [3], which can process edge features when applicable. For further details regarding our implementation, please refer to Appendix F (page 24) -- the update mechanism of the GINE base encoder, which supports edge features, is detailed in Equation (92) on page 24.
>
> **References:**
>
> [1] Bevilacqua et al. Efficient subgraph gnns by learning effective selection policies. ICLR 2023
>
> [2] Kong et al. Mag-gnn: Reinforcement learning boosted graph neural network. NeurIPS 2024
>
> [3] Hu et al. Strategies for pre-training graph neural networks. ICLR 2020

---

> > ### Comment · Reviewer_DujM · 2024-08-10
> > **Thank you**
> >
> > Thank you for your rebuttal. I acknowledge reading it.

---

### Official Review · Reviewer_h2GK · 2024-07-12

**Soundness:** 3
**Presentation:** 1
**Contribution:** 2
**Rating:** 6
**Confidence:** 4

**Summary:**

The authors introduce a novel graph learning method that leverages message passing over product graphs. Specifically, message passing is performed over both the input graph and its product with a coarsened version of itself, which can be derived through techniques such as node clustering. Additionally, the authors propose a concept called symmetry-based updates, where message passing is enhanced by incorporating a graph representing a linear equivariant layer over the product graphs. A basis for such layers is also developed as part of their contribution. Experimental results demonstrate that this method performs comparably to state-of-the-art approaches.

**Strengths:**

**S1 Product construction** The main conceptual contribution of the paper is the use of product graph construction to allow more flexibility and control over certain types of subgraph GNNs. The product construction itself is quite natural, although it introduces some overhead depending on the chosen instantiation.

**S2 Equivariant basis** The key theoretical contribution is the identification of the bases of equivariant layers for such product graphs. This allows for a principled introduction of message passing using symmetries in the product graph.

**S3 Competitive architecture** Experiments validate the proposed methods, showing that they are competitive with state-of-the-art methods.

**Weaknesses:**

**W1 Unclear key definition**
The key idea underlying the paper is commendable, although not extremely innovative. However, one crucial component is the graph coarsening introduced in Section 4.1. I found the definition on page 4 confusing despite multiple readings. What is defined there is an exponentially large fixed graph, with all subsets as nodes and adjacencies defined in equation (3). However, you mention before and after this that various coarsenings can be used, such as spectral clustering. How do these fit into the fixed definition given here? The presentation could be clearer.

**W2 Weak theoretical analysis**
 The significance of Section 5, where the authors use their model for analyzing marking strategies, is not strongly demonstrated, at least based on the main paper. It primarily serves to justify the marking strategy in equation (10) but leaves a more detailed understanding of this strategy open. The section would benefit from a clearer presentation.

**W3 Unclear presentation**
Certain parts, as noted already in **W1** and **W2**, are not clearly described. Another example is Section 4.2.3, which is too succinct, and the visualization of $A_{equiv}$​ is rather cryptic. You mention a characterization and updates, but it is not clear what is meant here. A clear description of the instantiation of the framework used would be helpful. Similarly, the section starting from line 357 about “more than a coarsening function” is unclear, including the “informal” proposition statement.

**W4 Unclear connection with OSANs**
 In the related work section, the authors contrast their proposal with OSANs [30], stating that OSANs use ordered tuples whereas the current paper uses sets of indices. However, unordered subgraphs play a key role in the OSAN paper as well since they showed that they can encompass most subgraph formalisms. A more detailed comparison would benefit the paper.

**W5 Missing experimental details**
- For the peptides experiments, you write “by using effective subsets of subgraphs.” What do you mean by this? What is this effective subset? How did you choose it?
- Which version of OSAN do you compare with in the experiments (Table 4)?
- Why do you use comparisons with different methods across the experiments (Table 2 vs Table 4)?

**W6 Maximally expressiveness: clarification needed** At several points in the paper, you mention maximally expressive subgraph GNNs. It is unclear what is meant by this and this should be detailed, regardless of whether it was shown in [39].  As far as I know, general subgraph GNN do not have a maximally expressive instantiation, so I assume certain restrictions on the class of subgraph GNNs are in place. It would be beneficial to make these explicit to better understand the limitations of the proposed methods.

**W7 Minor comments**

*All figures (especially the font size used in the text in figures) in the paper are too small. Please enlarge them.*

- L.82: What is $\mathcal{X}$?
- L218: The choice of $S_n$​ as a symbol for the symmetric group is not optimal, as subsets are denoted by $S_1$, $S_2$, etc.
- L225: You mention “extension to multiple features is straightforward” and refer to [22]. A bit more explanation would be helpful.

**Questions:**

I would appreciate it if the authors could respond to the weak points **W1**, **W4**, **W5**, and explain what they want to convey with the section on “more than a coarsening function" in Section 5.

**Limitations:**

This has been addressed in a satisfactory way by the authors.

---

> ### Author Rebuttal · Authors · 2024-08-05
>
> Thank you for your positive feedback. We are pleased you appreciated the novelty of our idea and the identification of the basis of equivariant layers for our product graphs. Below, we address each of your points.
>
> **Q1:** *Key definition*
>
> **A1:** We emphasize that the coarsened graph is not exponentially large in practice (see also lines 288-292 in our paper). While nodes of the coarsened graph are in $P([n])$, only a small number are used. Using spectral clustering, we partition the graph $G$ and construct the coarsened graph $\mathcal{T}(G)$ with nodes representing these partitions and edges induced by the original graph's connectivity (see Figure 1, left). This results in a coarsened graph with fewer nodes and edges than $G$. Although we define the graph for mathematical rigor, in practice, we use a much sparser graph by making $A^{\mathcal{T}}$ and $X^\mathcal{T}$ extremely sparse. **The space complexity of this sparse graph is upper bounded by that of the original graph $G$ (we do not store $2^n$ nodes)**. We thank the reviewer for pointing out this confusion and will clarify this in the final version of the paper.
>
> **Q2:** *Theoretical analysis*
>
> **A2:** We have justified the marking strategy from a theoretical perspective, demonstrating its superiority over previously suggested strategies [1,2,3]. While we have only briefly summarized our results in the main body due to space constraints, App. C (p.16) provides a detailed expansion.  We will make efforts to add more information to the main paper.
>
>
> **Q3:** *Presentation*
>
> **A3:** Thank you for highlighting these issues. Due to space limitations, Section 4.2.3 provides a brief summary of our use of the equivariant layers (see Section 4.2.2). For a detailed discussion, refer to App. F.4, p. 26). We will improve presentation and optimize space in the final version.
>
> 'More than a coarsening function’: We wanted to convey the fact that our method does not simply reduce to the joint processing of the graph and its coarsened counterpart. In fact, as we show in App. D.2 in detail, our approach strictly improves over running msg-passing on the graph obtained by combining its original topology with information from the coarsening. We term this new graph the “sum graph” and illustrate it on p. 19. Our method gains in expressiveness over the “sum-graph”, demonstrating the importance of our additional technical contributions, i.e., the use of the graph product and the related equivariant operations.
>
>
> **Q4:**  *Connection with OSAN*
>
> **A4:** We agree with the reviewer and will add a detailed comparison with OSAN in the next revision.
>
> In OSAN, the authors indeed show that ordered subgraphs are more expressive than unordered ones. In our case, although we choose coarsening functions which output unordered sets of nodes, we additionally leverage a set of additional equivariant operations which we formally derive as part of our contribution. Similarly to OSAN, it would be straightforward to show that our approach can indeed recover Subgraph GNNs operating on unordered subgraphs. However, the question whether OSAN fully captures our approach is of a less trivial answer, due to the aforementioned additional operations. Nevertheless, from an empirical perspective, our model demonstrates significantly better results.
>
> Additionally, we note that unordered tuples are practically preferred in OSAN due to computational complexity reasons. Throughout their experimental section, the authors focus on unordered subgraphs and learn how to sample subsets of them. Our method shares this trait of (practically) considering only unordered subgraphs. Contrary to OSAN, however, we choose coarsening and graph products as an alternative to sampling.
>
> We will refer to these observations in the next revision.
>
> **Q5:** *Experimental details*
>
> **A5:** We will answer each of the bullet points below:
> - "Using effective subsets of subgraphs": we refer to how our method naturally selects and uses relevant subgraphs for optimal performance.  To clarify, the initial step of our method involves coarsening the original graph $G$. We then use the graph Cartesian product between the coarsened graph $\mathcal{T}(G)$ and $G$ to create the product graph, as illustrated in Figure 1 on p. 2. Even without explicitly extracting subgraphs, elements in $\mathcal{T}(G) \square G$ can be conceptually associated with them. The "rows" in the product structure (Figure 1, right) represent implicit subgraphs, a common association in previous works [2,3]. We will improve our wording in the final version of the paper.
> - To ensure a fair comparison, we compared with the best-performing version of OSAN, as shown in Table 1(a) of the OSAN paper [4], p. 9.
> - Different papers use various datasets. We used common datasets where possible for comparison. Using different baselines for different datasets aligns with field practices due to the high computational cost of running all baselines on all datasets, as shown, for example, in Tables 2,3 in [2].
>
> **Q6:** *Maximally expressive*
>
> **A6:** We agree with the reviewer's observation, and indeed certain restrictions exist. To clarify, [1] studies the expressiveness of Subgraph GNNs using node-based policies. Thus, "maximal expressiveness" in our paper refers to Subgraph GNNs with node-based subgraph selection [1,3]. We will add an app. section to clarify this point.
>
> **Q7:** *Minor comments*
>
> **A7:**
> - $\mathcal{X}$ is the node feature matrix of the graph $\mathcal{T}(G) \square G$.
> - Thanks for the notes on L218, L225, and the figures. We'll revise them in the final version.
>
> **Refs.:**
>
> [1] Zhang et al. A Complete Expressiveness Hierarchy for Subgraph GNNs via Subgraph Weisfeiler-Lehman Tests. ICML 2023
>
> [2] Bevi. et al. Equivariant Subgraph Aggregation Networks. ICLR 2022
>
> [3] Frasca. et al. Understanding and Extending Subgraph GNNs by Rethinking Their Symmetries. NeurIPS 2022
>
> [4] Qian et al.  Ordered subgraph aggregation networks. NeurIPS 2022

---

> > ### Comment · Reviewer_h2GK · 2024-08-09
> > **Thanks for the rebuttal**
> >
> > Dear authors, thank you for your detailed rebuttal and thoughtful responses to my concerns. Incorporating those comments will definitely improve the paper.

---

### Official Review · Reviewer_hoHT · 2024-07-12

**Soundness:** 1
**Presentation:** 4
**Contribution:** 2
**Rating:** 2
**Confidence:** 3

**Summary:**

The paper is very well written. Existing subgraph GNN methods are mentioned. Then a novel subgraph GNN framework is formulated. The authors discuss the equivariance properties of this new subgraph GNN formulation followed by an experimental evaluation of the proposed method.

**Strengths:**

* Strong mathematical formulation of the developed methods is provided.
* The ideas are communicated well. One can see that the authors put a lot of time and effort into this work.
* A comprehensive appendix is provided explaining the proposed approach in even more detail.

**Weaknesses:**

### The use of few and small datasets

If the goal of this work is to keep computational complexity low while improving the model quality, strong experimental results are expected.  This puts into question the developed theory and methods in the paper. I suggest the authors to find a scenario where their model can achieve SOTA results to justify their ideas. I would recommend including more datasets such as PCQM4Mv2 [1] to the experimental evaluation.

### Limited baselines

A simple comparison against a recent linear runtime complexity graph transformer alternative [2] shows that the experimental results are weak. A select few datasets are chosen and the achieved results are well below the current SOTA on them.

### Weak motivation

If subgraph GNNs address the issue of limited expressiveness of existing work as stated in the introduction of the paper, then it makes one question why are the experimental results not good.

[1] https://ogb.stanford.edu/docs/lsc/pcqm4mv2/

[2] https://openreview.net/pdf?id=duLr8BIzro

**Questions:**

1. Is there a case where the computational gains leads to a SOTA results on a large scale dataset?
2. Could the experimental results improved if the proposed approach was combined with an approach similar to [2]. Improving the expressibility of the proposed model even further by combining with other classes of GNNs to get SOTA results would significantly improve the chances of the paper.

[2] https://openreview.net/pdf?id=duLr8BIzro

**Limitations:**

The authors admit their proposed method has a high computational complexity. Thus naturally one would expect even higher model quality.

---

> ### Author Rebuttal · Authors · 2024-08-05
>
> Thank you for your detailed feedback. We appreciate your positive remarks on our method's novelty, strong mathematical formulation, and clear writing. Below, we address your specific questions.
>
> **Q1:**  *The goal of our work and baselines*
>
> **A1:** Let us clarify. The goal of this work is to reduce the time and space complexity of Subgraph GNNs. As a result, some model degradation is expected. Subgraph GNNs are provably expressive models with theoretical guarantees [3, 4, 5], which may offer certain advantages over MPNNs and graph transformers. However, Subgraph GNNs typically have high computational complexity. Similarly to works like OSAN, PL, and Mag-GNN [6, 7, 8], our objective is to balance the high performance/accuracy of Subgraph GNNs with reduced computational expense. Unlike these other efficient Subgraph GNNs [6, 7, 8], our method offers a much finer control over this trade-off between computational complexity and accuracy in graph learning tasks, this is clearly illustrated in Figure 2, where bag size (x-axis) correlates with computational complexity. Within this family of more efficient Subgraph GNNs, we achieve SOTA results on most of the datasets these baselines are typically benchmarked on (see Tables 1, 4).
>
> **Q2:** *large scale datasets*
>
> **A2:** We refer the reviewer's attention to Table 8 in the appendix (p. 27), which presents results on the larger-scale ZINC-full dataset (250,000 graphs). We achieve top results in the efficient Subgraph GNN regime ($T=4$), outperforming MAG-GNN, the only other efficient subgraph GNN that reported results on this large-scale dataset. Furthermore, when using  our model in the farthest end of the trade-off (Full bag - $T="Full"$), we achieve results comparable to the top-performing traditional "non-efficient" Subgraph GNNs, and *outperform leading Graph Transformers*. For instance, GRIT [9] reported an MAE of 0.023 on ZINC-FULL, while we obtained an MAE of 0.021.
>
> As the reviewer requested, we also conducted a preliminary experiment using the large-scale molpcba dataset (437,929 graphs), which is also considered in the paper mentioned by the reviewer [2]. In particular, we train one instance of our model with a 1M parameter budget and report its performance against the methods in [2], see Table 1 in the attached PDF (in the global response).
>
> Notably, our method achieves comparable results to SOTA on this dataset while using the least number of parameters. Compared to GatedGCN-LSPE, which uses a similar number of parameters, our method demonstrates better performance. This improvement is also evident when compared to the GraphTrans baseline, which utilizes approximately four times more parameters than our approach. We believe that the results over ZINC-FULL, as well as the preliminary results over molpcba, demonstrates the potential of our method to handle large-scale datasets well. If the reviewer considers it important, we will extend our experiments on molpcba and include the results in the final version of the paper.
>
> Due to resource constraints, we couldn't conduct an experiment with the PCQM4Mv2 dataset (approx. 3,500,000 graphs) within the rebuttal period.
>
> **Q3:** *SOTA results on large scale datasets*
>
> **A3:** Recalling **A1**, **A2**, we achieve SOTA results on the large-scale ZINC-FULL dataset (Table 8, appendix, p. 27) in both efficient ($T=4$) and full-bag Subgraph GNN scenarios. Our approach outperforms top transformers like GRIT [9]. Preliminary experiments on the large-scale molpcba also indicate our model's competitive SOTA performance (Table 1 in the pdf).
>
> **Q4:** *Weak motivation*
>
> **A4:** We want to emphasize that our primary motivation, similar to previous works [6, 7, 8], is to reduce the computational cost of Subgraph GNNs. As mentioned in **A1**, we believe that within the family of algorithms that tackle the same setup (OSAN, PL, and MAG-GNN) [6, 7, 8], our method provides significant improvements in most cases – Table 1, 4, 8.
>
> **Q5:** *Could the experimental results improved if the proposed approach was combined with an approach similar to [2]. Improving the expressibility of the proposed model even further by combining with other classes of GNNs to get SOTA results would significantly improve the chances of the paper.*
>
> **A5**: We appreciate the reviewer's suggestion and have incorporated a version of the LCB block from [2] into our framework, which we term `Ours + LCB`. We ran experiments on three datasets (ZINC12k, Molbace, and Molesol) for three different bag sizes ($T=2$, $T=5$, $T="Full"$). The results are in Table 2 of the attached PDF in the global response.
>
> We positively note that, in the small-bag regime, the `Ours + LCB` method improves the performance of our model (which already outperformed most competitors on these datasets), sometimes significantly. We will do our best to extend and include these results in the final version of the paper, should the reviewer find them relevant.
>
> **Summary:** We believe we have thoroughly addressed your concerns about our method's capability with large-scale datasets and its integration with approaches from [2]. If you find our response satisfactory, we kindly request that you consider raising your score.
>
> **References:**
>
> [1] https://ogb.stanford.edu/docs/lsc/pcqm4mv2/
>
> [2] https://openreview.net/pdf?id=duLr8BIzro
>
> [3] Zhang et al. A Complete Expressiveness Hierarchy for Subgraph GNNs via Subgraph Weisfeiler-Lehman Tests. ICML 2023
>
> [4] Bevi. et al. Equivariant Subgraph Aggregation Networks. ICLR 2022
>
> [5] Frasca. et al. Understanding and Extending Subgraph GNNs by Rethinking Their Symmetries. NeurIPS 2022
>
> [6] Qian et al.  Ordered subgraph aggregation networks. NeurIPS 2022
>
> [7] Bevi. et al. Efficient subgraph gnns by learning effective selection policies. ICLR 2023
>
> [8] Kong et al. Mag-gnn: Reinforcement learning boosted graph neural network. NeurIPS 2024
>
> [9] Ma et al. Graph Inductive Biases in Transformers without Message Passing. ICML 2023

---

> ### Author Response · Authors · 2024-08-12
> **Relating to our response**
>
> Dear Reviewer hoHT,
>
> We appreciate your thoughtful review and hope that we have adequately addressed your concerns. We made an effort to address your feedback during the rebuttal period. Specifically, we shared new results on new experiments:
>
> * We conducted new experiments using a large-scale dataset OGBG-MOLPCBA (437K graphs). Additionally, we referred you to another large-scale experiment we already included in the original submission using ZINC FULL (250K graphs). Both experiments show solid results for our method compared to SoTA.
>
> * We successfully incorporated components from [2] to enhance our architecture.
>
> These new results can be found in our original response and PDF.
> We would be grateful if you could review these additions and consider raising your score if they satisfactorily address your concerns.
>
> Best regards,
>
> The Authors

---

> > ### Comment · Reviewer_hoHT · 2024-08-12
> >
> > I would like to thank the authors for their rebuttal.
> >
> > I am comparing Table 2, Peptides-func and peptides-struct results to [2].
> >
> > The reported SOTA results in [2] for these datasets are 0.6975 and 0.2464, both of which significantly are better than what is being reported in Table 2 in this work.
> >
> > molhiv results are subpar as well (79.44 in this paper vs 80.84 reported in for CIN method and 79.80 for GECO both in [2]). Essentially there is no dataset that justifies the added complexity of the proposed methods in this work. To me, it looks like the authors are working on making an approach faster that does not seem to bring any clear benefits.
> >
> > > The goal of this work is to reduce the time and space complexity of Subgraph GNNs. As a result, some model degradation is expected. Subgraph GNNs are provably expressive models with theoretical guarantees [3, 4, 5], which may offer certain advantages over MPNNs and graph transformers.
> >
> > If subgraph GNNs have high computational complexity but don't work well, then this undermines the impact of the work in this paper. Since my score is borderline, I would like to instead change my score and go for a recommendation of rejection. If the authors can motivate subgraph GNNs better by showing that they work well over a class of datasets in a future revision of the paper, that would significantly boost the chances of acceptance of their work. This is due to the fact that datasets with a lot of graphs already are made of small graphs, and non-subgraph classes of methods already work well and are computationally efficient.
> >
> > [2] https://openreview.net/pdf?id=duLr8BIzro

---

> > > ### Author Response · Authors · 2024-08-13
> > > **Concerns Regarding Unjustified Score Reduction**
> > >
> > > Dear Reviewer,
> > >
> > > We appreciate your time and effort in evaluating our work. However, we respectfully disagree with the recent score reduction and would like to raise some concerns:
> > >
> > > 1. Changing the score so abruptly just before the discussion deadline hinders our ability to respond and compromises the fairness of the rebuttal and discussion process.
> > > 2. Our rebuttal comprehensively addressed all your initial concerns, providing clarifications and additional positive experimental results as per your request. Your response does not acknowledge our inputs or new results. Instead, it lowers the score from 4 to 2 based on the same initial comments that originally received a score of 4. This discrepancy is difficult to reconcile.
> > > 3. Your current score of 2 contrasts sharply with both your initial assessment and the positive evaluations from other reviewers (6,7).
> > > 4. We are concerned that our prompt for a response may have indirectly led to this unjustified score reduction.
> > > 5. We would like to emphasize a key contribution that may have been overlooked: the introduction of new symmetries and the characterization of accompanying equivariant layers.
> > >
> > > For a conference of this tier, we expect reviews to be consistent, thorough, and responsive to rebuttals. The dramatic score change without new concerns or acknowledgment of our responses does not align with these expectations. We have also contacted the AC and SAC regarding this matter. We respectfully request that you reassess your recent score reduction.

---

### Author Rebuttal · Authors · 2024-08-05

We are grateful to all reviewers for their feedback and constructive comments and happy to see that our work was positively received in general.

In particular, the reviewers all recognized the novelty and importance of our proposed method:
- “The topic of subgraph GNNs is interesting” (**DujM**)
- “a novel subgraph GNN framework” (**hoHT**)
- “a novel graph learning method” (**h2GK**)
- “The proposed method is novel” (**DujM**)

An essential part of our work involves recognizing that our product graph introduces new permutation symmetries. We demonstrate how to construct networks that respect these symmetries, supported by both theoretical analysis and empirical evidence. This approach has also been positively received:

- “The paper also discovers new permutation symmetries” (**DujM**)
- “Strong mathematical formulation” (**hoHT**)
- “The proposed method is novel, with theoretical underpinnings” (**DujM**)
- “Theoretical and empirical analysis validate the efficacy of the proposed method”. (**DujM**)
- “The key theoretical contribution is the identification of the bases of equivariant layers for such product graphs.” (**h2GK**)
- “Experiments validate the proposed methods, showing that they are competitive with state-of-the-art methods” (**h2GK**)
- “The empirical results seem promising.” (**DujM**)

Finally, we were also very happy to notice that the reviewers have found our paper to be “very well written” (**hoHT**, **DujM**).
At the same time, reviewers shared comments and questions, which we address in the specific rebuttals below. Additionally, we conducted new experiments as requested by (**hoHT**). The results, included in the attached PDF file, demonstrate that our method effectively handles large-scale datasets and benefits from combining our approach with his suggested method.

---

### Decision · Program_Chairs · 2024-09-25

**Decision:**

Accept (poster)

**Comment:**

This paper presents a smart idea of efficient subgraph GNNs by performing message passing on the product graph of the original graph and a coarsend graph, where the coarsening degree controls the complexity of the final model. This contrasts from existing acceleration techniques based on random sampling or learning to select a subset of subgraphs, and has the benefit of training-free acceleration and flexible control of the model complexity. The authors also identify the new symmetries in the product graph and characterizes all the linear equivariant operators through solid theoretical analysis. Experimental results demonstrate that the method compares favorably to random sampling and MAG-GNN (RL-based sampling) in small bag size settings, and approaches SoTA results in full bag size settings.

Overall, considering subgraph GNNs are a promising approach to break the expressivity bottleneck of GNNs/GTs and accelerating subgraph GNNs is of vital importance for their wide adoptations by practioners, I would recommend an acceptance of the paper.